# Reversed Holocene temperature–moisture relationship in the Horn of Africa

A. J. Baxter[1,10 ✉], D. Verschuren[2,10], F. Peterse[1], D. G. Miralles[3], C. M. Martin-Jones[4], A. Maitituerdi[5], T. Van der Meeren[2], M. Van Daele[6], C. S. Lane[4], G. H. Haug[7], D. O. Olago[8] & J. S. Sinninghe Damsté[1,9]

Anthropogenic climate change is predicted to severely impact the global hydrological cycle[1], particularly in tropical regions where agriculture-based economies depend on monsoon rainfall[2]. In the Horn of Africa, more frequent drought conditions in recent decades[3,4] contrast with climate models projecting precipitation to increase with rising temperature[5]. Here we use organic geochemical climate-proxy data from the sediment record of Lake Chala (Kenya and Tanzania) to probe the stability of the link between hydroclimate and temperature over approximately the past 75,000 years, hence encompassing a sufficiently wide range of temperatures to test the 'dry gets drier, wet gets wetter' paradigm[6] of anthropogenic climate change in the time domain. We show that the positive relationship between effective moisture and temperature in easternmost Africa during the cooler last glacial period shifted to negative around the onset of the Holocene 11,700 years ago, when the atmospheric carbon dioxide concentration exceeded 250 parts per million and mean annual temperature approached modern-day values. Thus, at that time, the budget between monsoonal precipitation and continental evaporation[7] crossed a tipping point such that the positive influence of temperature on evaporation became greater than its positive influence on precipitation. Our results imply that under continued anthropogenic warming, the Horn of Africa will probably experience further drying, and they highlight the need for improved simulation of both dynamic and thermodynamic processes in the tropical hydrological cycle.

The incongruence between the common prevalence of severe drought conditions in the Horn of Africa during recent decades[3,4] and climate model simulations projecting rainfall to increase during the twenty-first century[5], termed the 'Eastern African climate paradox'[8,9], confounds the region's climate change adaptation efforts by undermining strategic agricultural planning and water-resource management[9]. Contrary to other dry (sub)tropical regions such as southern Africa, where projections of increasing drought are generally consistent with the instrumental record, projections of increasing rainfall ($P$) and stable effective moisture (precipitation minus evaporation, $P - E$) over easternmost Africa (Fig. 1b) are clearly at odds with the predominantly inverse relationship between annual $P - E$ and temperature ($T$) observed in 42 years of instrumental data (Fig. 1a). The recent historical (and ongoing) drying trend in the Horn of Africa is mainly expressed in delayed onset and earlier cessation of the March–May 'long rains' that support the principal crop-growing season[9]. Whereas observational studies[10] and analyses of model-based climate change projections[11] tend to focus on changes in atmospheric circulation, continental hydrology and water-resources studies place more emphasis on thermodynamic processes such as land–atmosphere feedbacks[12,13]. Palaeoclimate proxy data from high-quality geological archives can help improve projections of future water availability by probing the stability of the relationship between effective moisture and temperature over a range of past temperatures large enough to determine whether the Horn of Africa's semiarid tropical climate regime is more likely to become progressively wetter or drier under future anthropogenic warming.

Despite the growing number of palaeoclimate records from tropical Africa that extend beyond the Last Glacial Maximum (LGM; about 23,000–19,000 years ago, 23–19 kyr ago) into the preceding full glacial period, only a handful of these comprise well resolved reconstructions of both temperature and hydroclimate. Here we present paired high-resolution temperature and hydroclimate proxy data (approximately 200-year interval on average; $n = 373$) for the past 75 kyr, based on the distribution of glycerol dialkyl glycerol tetraethers (GDGTs) in a depositionally continuous[14] sediment sequence from Lake Chala (Kenya/Tanzania; Extended Data Fig. 1) recovered by the International Continental Scientific Drilling Program project DeepCHALLA[15].

[1]Department of Earth Sciences, Faculty of Geosciences, Utrecht University, Utrecht, The Netherlands. [2]Department of Biology, Limnology Unit, Ghent University, Gent, Belgium. [3]Department of Environment, Hydro-Climate Extremes Lab (H-CEL), Ghent University, Gent, Belgium. [4]Department of Geography, University of Cambridge, Cambridge, UK. [5]Dr. Moses Strauss Department of Marine Geosciences, Leon H. Charney School of Marine Sciences, University of Haifa, Mount Carmel, Israel. [6]Renard Centre of Marine Geology, Department of Geology, Ghent University, Gent, Belgium. [7]Department of Climate Geochemistry, Max Planck Institute for Chemistry, Mainz, Germany. [8]Institute for Climate Change and Adaptation, Department of Earth and Climate Science, University of Nairobi, Nairobi, Kenya. [9]Department of Marine Microbiology and Biogeochemistry, NIOZ Royal Netherlands Institute for Sea Research, Den Burg, The Netherlands. [10]These authors contributed equally: A. J. Baxter, D. Verschuren. ✉e-mail: A.J.Baxter@uu.nl

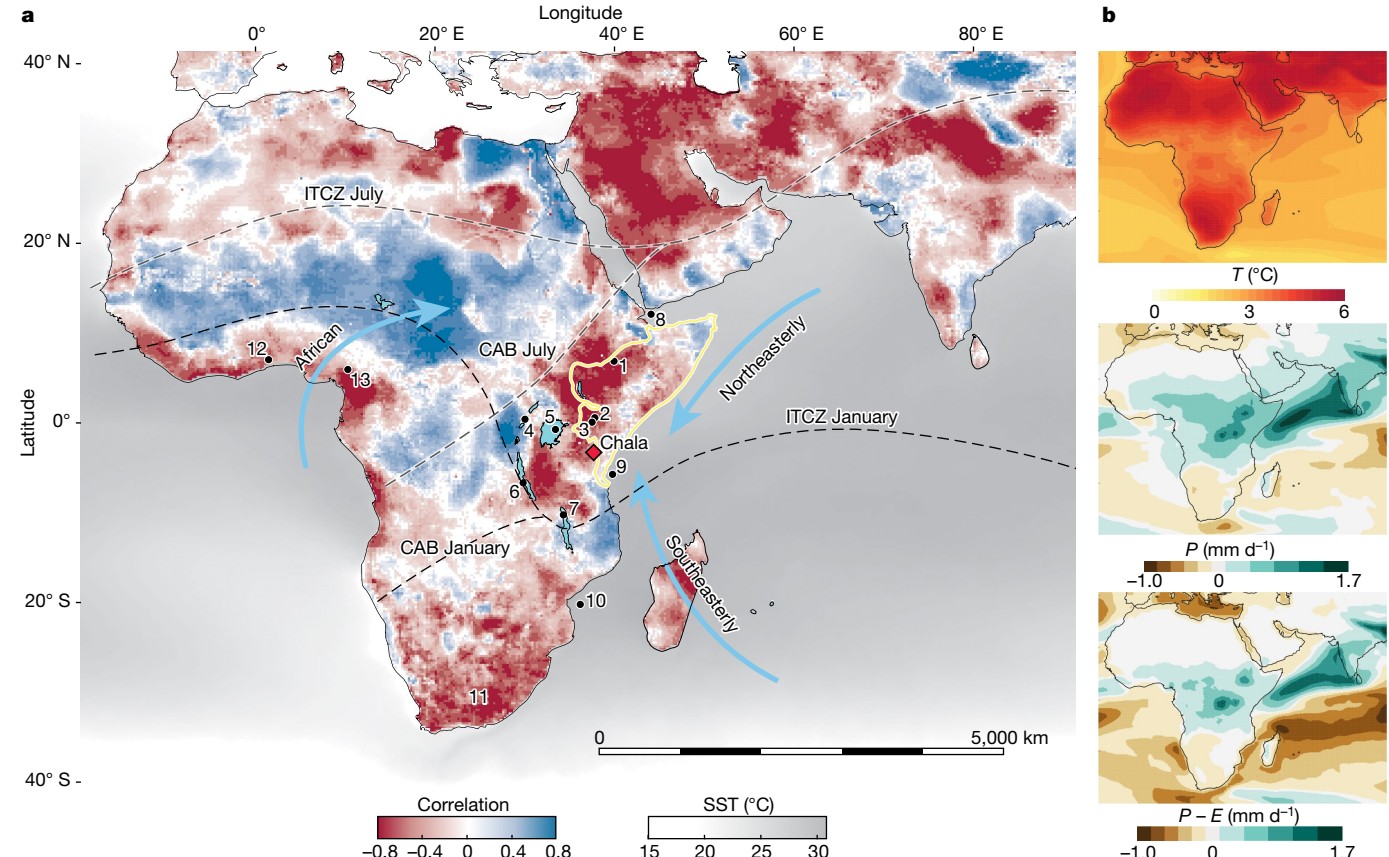

**Fig. 1 | Location of Lake Chala and other palaeoclimate archives in relation to present-day and projected future climatology over the African continent and adjacent oceans. a**, Blue and red hues on the continents show the correlation between annual effective moisture ($P - E$) and annual temperature ($T$) in observational data over the period 1980–2021 (Methods), all downscaled to 0.25° resolution for visualization purposes. Correlations with absolute values exceeding 0.4 are statistically significant ($P < 0.01$). Mean SSTs[50] exceeding 23 °C (grey) delineate the tropics. The black and grey dashed lines represent the approximate positions of the ITCZ and CAB in January and July, respectively, and the blue arrows show the dominant seasonal wind directions associated with the respective monsoon systems. The area in eastern Africa delineated with a yellow line is the Horn of Africa region fully dependent on Indian Ocean moisture[9]. The black dots labelled 1–13 are the locations of lake-based temperature records used to derive a 25-kyr eastern African ensemble reconstruction (1–7), selected SST records from the western Indian Ocean (8–10), a pollen-based temperature record from southeastern Africa (11; all shown in Extended Data Fig. 4) and pollen-based continental moisture records from western tropical Africa (12 and 13)[35,36]. **b**, Changes in $T$ (°C), $P$ (mm day$^{-1}$) and $P - E$ (mm day$^{-1}$) by the end of the twenty-first century (2081–2100 versus 1995–2014) over the African continent and adjacent oceans, as simulated by the CMIP6 model ensemble under the SSP5-8.5 emissions scenario[51].

Located east of the Congo Air Boundary (CAB) year-round (Fig. 1a), past climate dynamics registered there can be considered representative for the easternmost portion of Africa that is fully dependent on Indian Ocean moisture[16]. In modern-day climatology this region is defined as the 'Greater Horn of Africa' (including Somalia, southern Ethiopia, eastern Kenya and northeastern Tanzania) characterized by a bimodal rainfall regime[4,9]. Twice-annual passage of the tropical rainbelt traditionally associated with the Intertropical Convergence Zone (ITCZ)[17] creates two rainy seasons (March–May and October–December) and a distinct dry season during boreal summer (June–September).

## A long paired temperature–hydroclimate record

GDGTs are membrane lipids produced by bacteria and archaea, and widely used in palaeoclimate research[18]. We reconstructed past mean summer temperature (MST) using a calibration based on the distribution of branched GDGTs (brGDGTs) in a set of globally distributed lakes[19], which we consider to best capture temperature variability through time (Methods). We reconstructed hydroclimate variability using the branched versus isoprenoid tetraether (BIT) index[20] based on brGDGTs and the isoprenoid GDGT crenarchaeol (Methods).

In the deep and permanently stratified (meromictic) Lake Chala, brGDGT-producing microbes live in the non-mixing and perennially anoxic lower water column (Extended Data Fig. 2, zones 4–6)[21], whereas crenarchaeol is produced by Thaumarchaeota living in the seasonally mixing upper part of the hypolimnion (Extended Data Fig. 2b), where they thrive mostly in the suboxic zone immediately above the oxycline (Extended Data Fig. 2, zone 2), probably owing to their photosensitivity and the greater nutrient availability at depth[22,23]. Under conditions of strong upper-water-column stratification, the oxycline moves upwards to near the base of the daily mixed layer, eliminating the niche for Thaumarchaeota, and hence the potential for crenarchaeol production (Extended Data Fig. 2a). Thus, Lake Chala sediments with high BIT-index values relate to past conditions of more positive climatic moisture balance, because a greater surplus of $P$ over $E$ enhances both groundwater recharge and overland flow[24], leading to higher lake level (that is, greater lake depth) and more pronounced water-column stratification, in turn affecting the relative niche availability of different GDGT-producing microbes[22] (Methods). Although the DeepCHALLA sequence encompasses about 250 kyr (refs. 14,15), our present analysis is limited to the past 75 kyr because the uniform lithology of this upper section (Methods and Fig. 2) provides confidence that the modern-day limnological setting of Lake Chala persisted throughout, and thus that

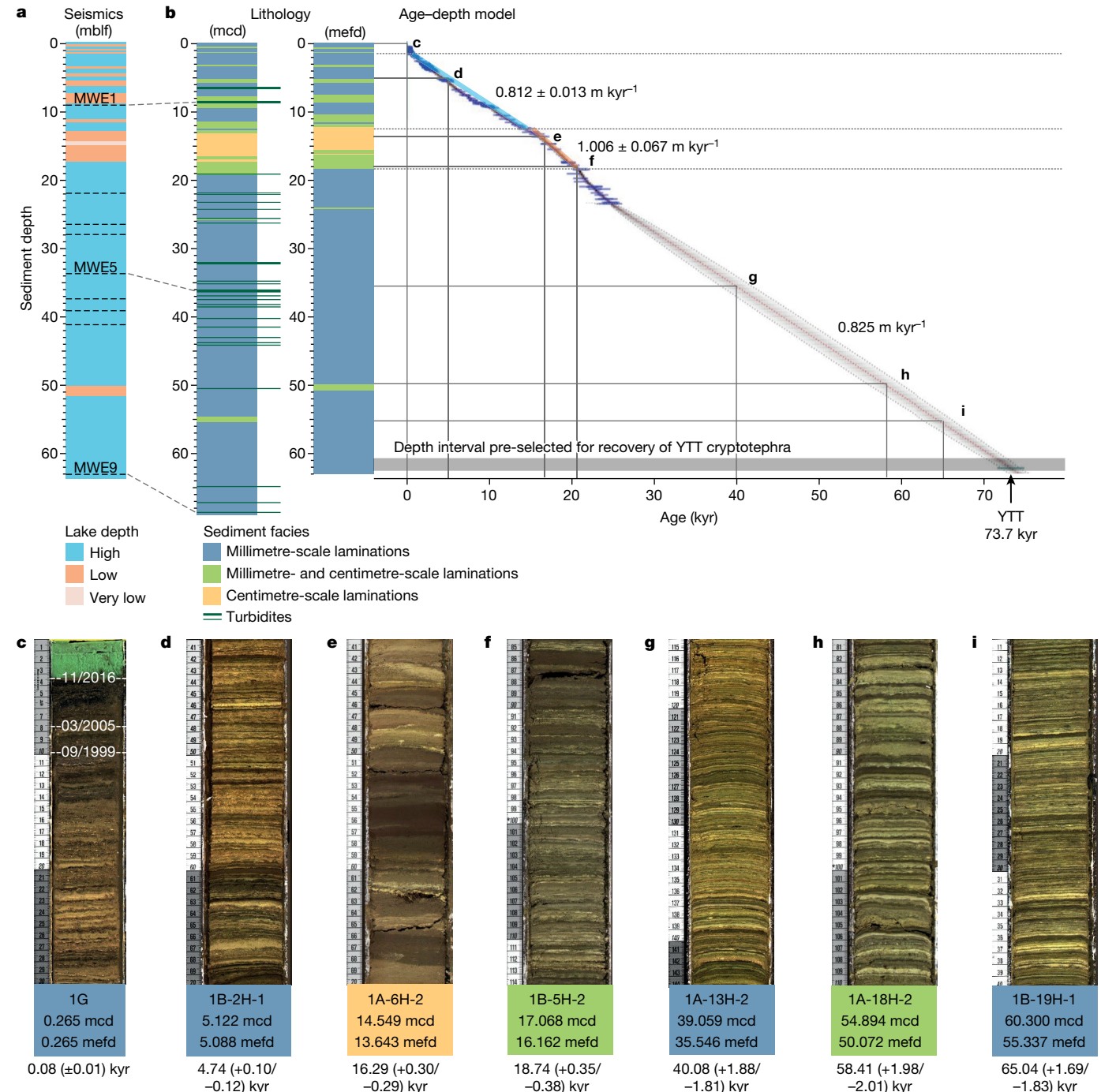

**Fig. 2 | Seismic stratigraphy and lithology of the Lake Chala sediment record, and age–depth model for the approximately 75-kyr GDGT-based temperature and moisture reconstructions. a**, Seismic stratigraphy of lacustrine deposits at the DeepCHALLA drill site[14] in metres below the lake floor (mblf). MWE1, MWE5 and MWE9 are three mass-wasting events (MWEs; thick dashed lines) expressed on seismic profile Challa05 (Extended Data Fig. 1) that can be traced to individual turbidite horizons in the drilled sediment sequence. **b**, Left, lithology of the analysed sequence with distribution of sedimentary facies plotted against composite core depth (mcd; includes turbidite horizons, indicated as brown lines extending to the right) and event-free composite core depth (mefd; with turbidites excised). Right, age–depth plot with 170 absolutely dated depth intervals incorporated in the Bayesian age model (blue horizontal lines) and 2σ age uncertainty envelope (fading grey bordered by stippled line). Also indicated are the 1.78-m (approximately 2,000-year) core section at the base of the sequence pre-selected for recovery of the YTT cryptotephra (grey horizontal bar), the long-term average rates of

sediment accumulation under 'lake high-stand' (bold blue line) and 'lake low-stand' (bold pale brown line) conditions based on [14]C dating of the 25.2–0-kyr interval, and the long-term average rate of sediment accumulation across the 73.7–25.2-kyr interval (Methods). **c–i**, Digital line-scan images of representative core sections (rulers with centimetre scale on left) illustrating the depositional integrity of the Lake Chala sediment record and variation in lithological facies ranging from consistently millimetre-scale (varve-like) lamination (blue; **c,d,g,i**) over a mixture of millimetre-scale lamination and centimetre-scale banding (green; **f,h**) to predominantly centimetre-scale banding (yellow; **e**). Each panel is labelled with its DeepCHALLA (DCH-CHL16) core-section code, its basal depth in mcd and mefd, and its age according to the age model shown in **b**, right. In **c**, the white dashed lines indicate the position of the sediment–water interface at the time of initial site exploration (September 1999) and fieldwork for the Challacea and DeepCHALLA projects (March 2005 and November 2016).

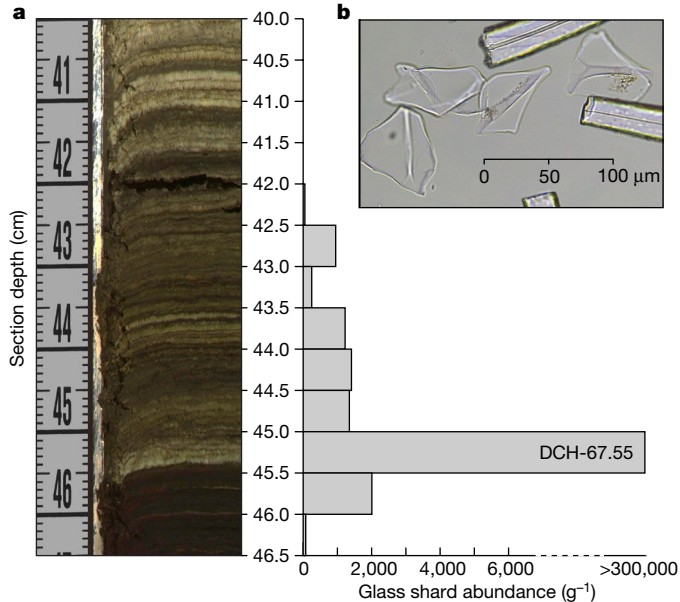

**Fig. 3 | Distribution and morphology of cryptotephra from the 73.7-kyr-old YTT in the Lake Chala sediment sequence. a**, Bar graph alongside partial image of core section DCH-CHL16-1B-21-H-2 shows the vertical distribution of volcanic glass shards, per gram dry sediment in contiguous 0.5-cm-depth increments (Methods). The YTT cryptotephra[25] is identified as DCH-67.55 based on its position at 67.55 mcd (equivalent to 62.36 mefd). As the fine (that is, seasonal scale) sediment lamination in this core section excludes the possibility of post-depositional disturbance, the 2.5-cm-long 'tail' of modest YTT glass shard abundance above their peak concentration of about 300,000 shards per gram dry sediment at 45.5–45.0 cm must represent an approximately 30-year post-eruption period during which initial YTT tephra fall-out over the crater catchment is flushed into the lake, and/or some YTT tephra first settling onto shallower areas of lake bottom is gradually being winnowed towards the DeepCHALLA drill site. **b**, Plane-polarized light photomicrograph of YTT glass shards, which are morphologically distinct from all other volcanic glass shards recovered thus far from Lake Chala sediments[52].

the climate-related drivers of GDGT distribution in its sediments as currently understood[22] can be assumed stable through time.

Excluding turbidites and other event deposits interrupting primary deep-water sedimentation at the DeepCHALLA drill site (Methods), the studied sediment sequence consists almost entirely of millimetre-scale laminated muds (Fig. 2), testifying that deep-water sedimentation has been highly stable through time. The age model supporting our climate-proxy time series is anchored in 170 radiometric age markers ([14]C and [210]Pb) covering the past 25.2 kyr, Younger Toba Tuff (YTT) glass shards from the 73.7-kyr-old Toba super-eruption in Indonesia[25] discovered near the base of the studied sequence (Fig. 3 and Extended Data Fig. 3), and linear interpolation across the section of predominantly varve-like sediments constituting the 73.7–25.2-kyr interval (Fig. 2). Notwithstanding lithostratigraphic constraints on long-term variation in the rate of profundal sediment accumulation in this lake system (Methods), the age ranges cited below of millennial-scale climate events within this age interval should be viewed as approximate.

The temperature history of easternmost Africa over the past 75 kyr (Fig. 4e) shows a clear pattern of variable but relatively cool conditions near the Equator throughout the glacial period (Marine Isotope Stage 4 (MIS4) to MIS2) shifting to the consistently warmer interglacial climate of the Holocene (MIS1). The mildest glacial temperatures are reconstructed for the period between about 56 kyr ago and 26 kyr ago (that is, broadly coincident with MIS3), in agreement with the temperature histories of Greenland[26] and Antarctica[27], and with the reduced volume

of continental ice sheets during that time as inferred from the marine oxygen-isotope record[28] (Fig. 4a,b). The Younger Dryas (YD) stadial (12.9–11.7 kyr ago) is represented as a pause in the postglacial warming trend, as has also been inferred for southern Africa based on pollen data (Extended Data Fig. 4f). Peak Holocene temperatures are inferred to have occurred between 8.5 kyr ago and 5 kyr ago, consistent with the mid-Holocene thermal maximum documented from other lakes in eastern Africa (Extended Data Fig. 4a), southern Africa (Extended Data Fig. 4f) and the global tropics as a whole[29]. Together, these features indicate that the temporal structure of our brGDGT-based reconstruction reflects actual temperature trends through time. However, the magnitude of inferred continental temperature variation (total MST range 16–28 °C; Fig. 4e) is substantially larger than has been inferred from other regional proxy records[30], and can be considered unrealistic. As the focus of this study is on the timing and relative magnitude of past temperature anomalies, we rescaled our brGDGT-based temperature estimates to a regional ensemble reconstruction covering the past 25 kyr based on 7 other currently available temperature records from eastern African lakes (Extended Data Fig. 4a–c and Methods). After rescaling, MST variation amounts to 4.8 °C over the past 75 kyr, with glacial temperatures up to 3.2 °C cooler than today and peak mid-Holocene temperature 1.6 °C warmer than today, consistent with compilations from the global tropics[29,31]. The lowest temperatures during the glacial period are reconstructed to have occurred towards the ends of MIS4 and MIS2, the latter encompassing the LGM and Heinrich Stadial 1 (HS1; 18–15 kyr ago).

Our BIT-index record (Fig. 4d) indicates that during the last glacial period, and especially during MIS3, climate conditions often wetter than today prevailed near the Equator in easternmost Africa, and that sustained dry conditions were limited mostly to the later halves of MIS4 and MIS2. Most extreme drought occurred between about 64 kyr ago and about 57 kyr ago, 20.5–16 kyr ago (that is, considerably offset from the LGM but overlapping with HS1) and during the YD. Our moisture-balance reconstruction also captures the strong resumption of monsoon rainfall at the end of the YD[32] heralding the Holocene portion of the African Humid Period[33]. As highlighted before[34], the Chala BIT-index record indicates that a moist African Humid Period climate regime in easternmost equatorial Africa ended about 9 kyr ago and was followed by a relatively dry mid-Holocene period before conditions became more humid again in the last four millennia.

Our finding of mostly wet conditions during MIS3 (Fig. 4d) contrasts with vegetation and lake-status reconstructions from western equatorial Africa[35,36] indicating that, there, predominantly dry conditions prevailed throughout the last glacial period, with only modest forest recovery (that is, slightly higher effective moisture) during MIS3. It has been suggested that the formation of large ice caps over northern North America and Eurasia induced drying on the Atlantic side of tropical Africa during MIS3, partly because slowdown of the Atlantic Meridional Ocean Circulation shifted the mean position of the tropical rainbelt southwards at least during cold Greenland stadials[37]. It also weakened South Asian monsoon circulation because the atmospheric cooling effect of high ice-sheet albedo reduced evaporation in the Arabian Sea[38]. As eastern Africa lacks a direct oceanic teleconnection to the northern high latitudes, cooling in the western Indian Ocean (Fig. 4e) and further south (Extended Data Fig. 4e) during MIS3 was relatively modest, such that monsoonal moisture delivery to easternmost Africa remained strong (Fig. 4d).

Notably, the coldest episode of the last 75 kyr in easternmost Africa (2.5–3.2 °C colder than today) did not occur during the LGM when global ice volume was greatest[28], but several thousand years later, between 18.3 kyr ago and 14.1 kyr ago. Thus, both the lowest temperatures and the greatest drought occurred broadly during HS1 (18–15 kyr ago), when the Atlantic Meridional Ocean Circulation slowdown[39] prompted a major reorganization of Earth's hydrological cycle[40]. Continental hydroclimate proxy data from around the Indian Ocean, such as the

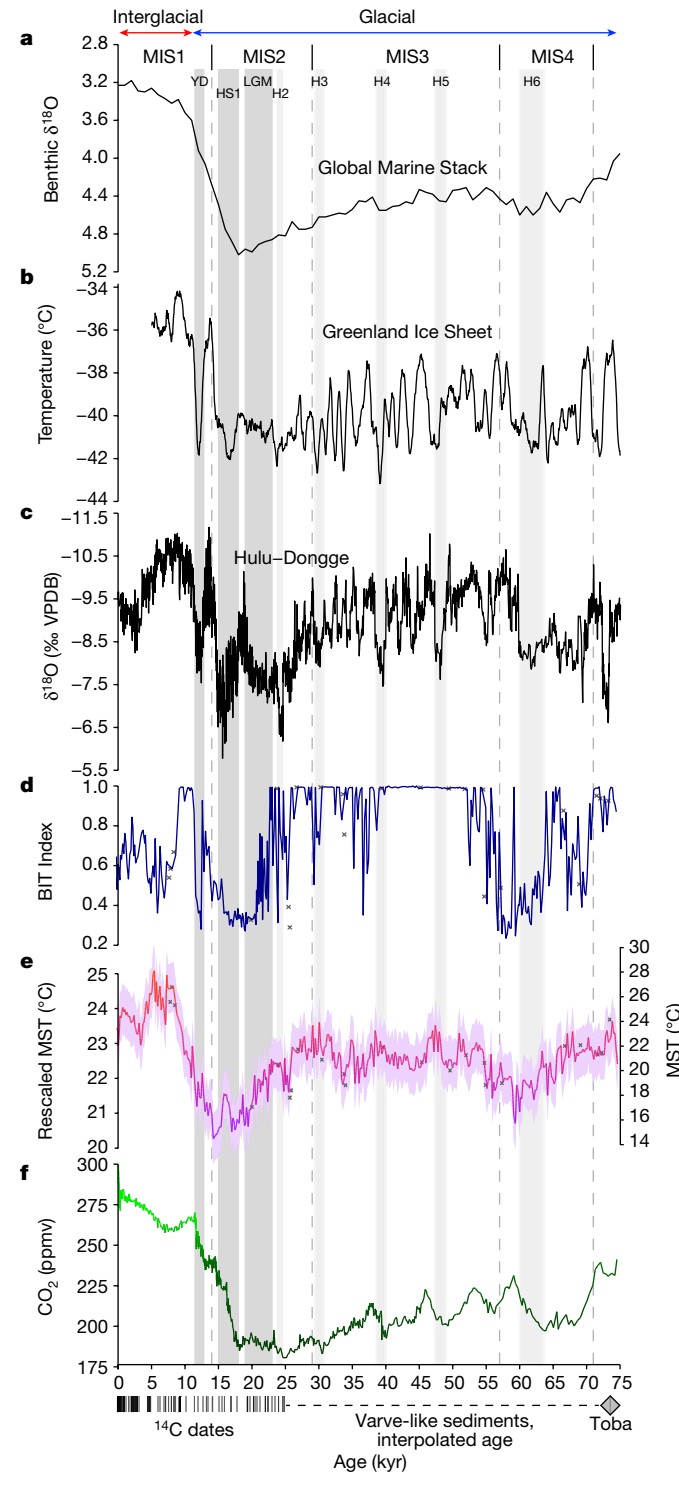

**Fig. 4 | Paired temperature and hydroclimate reconstruction for easternmost Africa spanning approximately the past 75 kyr in relation to globally distributed palaeoclimate reference records. a**, Benthic marine oxygen-isotope stack[28], reflecting global ice volume. **b**, Synthetic Greenland oxygen-isotope record[26], reflecting Arctic temperature variation. **c**, Speleothem composite oxygen-isotope record from Hulu–Dongge[53], reflecting East Asian summer monsoon intensity. **d**, BIT index[20] of sedimentary GDGTs from Lake Chala, reflecting lake water-balance variation. **e**, MST variation based on sedimentary brGDGTs from Lake Chala, calibrated[19] and rescaled (left $y$ axis) using an eastern African ensemble reconstruction (Extended Data Fig. 4 and Methods); the purple shading shows the RMSE uncertainty envelope of this calibration, ±0.7 °C after rescaling. Data from sediment horizons partly consisting of thin turbidites are shown as grey crosses and excluded from interpretation. **f**, Atmospheric $CO_2$ concentration as recorded in Antarctic ice[44], with a shift from dark to light green centred on 250 ppmv. The ages of individual radiocarbon-dated intervals (thin vertical lines) and the 73.7-kyr YTT cryptotephra (grey diamond) used for age-model construction (Fig. 2 and Methods) are plotted along the bottom axis. The timing of the LGM, HS1 and YD are shown for reference. Considering linear age interpolation in the 25.2–73.7-kyr section of our proxy time series (Methods), the timing of Heinrich events H2–H6 should be viewed as approximative (as indicated by gradated light grey shading).

African continent quickly re-invigorated the southeasterly monsoon[34], whereas the South Asian monsoon responded later to this insolation forcing because the Arctic sea-ice expansion that cooled northern Eurasia during HS1 initially weakened monsoonal moisture advection into southern Eurasia[42]. In any case, as the HS1 megadrought impacted both the northern and southern tropics, its principal cause is unlikely to be a southwards shift of the tropical rainbelt triggered by North Atlantic cooling. Rather, the rainbelt's reduced moisture content owing to unusually low Indian Ocean sea surface temperature (SST) may be responsible[41], consistent with our data (Fig. 4d,e).

## From energy- to moisture-limited climate

Focusing on how past hydroclimate at Lake Chala was generally related to changes in temperature, we find a distinctly positive correlation between BIT-index and MST values during glacial time (75–11.7 kyr ago, or broadly encompassing MIS4–MIS2: Spearman's rank correlation coefficient ($\rho$) = 0.71, $P < 0.001$, $n = 311$; Fig. 5a). This strong positive relationship suggests that under a climate regime modestly cooler than today, variation in continental effective moisture was controlled primarily by the amount of local monsoon rainfall, which depends first on the intensity of surface evaporation from the adjacent Indian Ocean, and second on the temperature contrast between ocean and land driving the advection of this moisture onto the continent[43]. Episodes of drought were probably owing to reduced ocean evaporation and weakened monsoon dynamics; accordingly, most extreme drought occurred during the two glacial-era periods with the lowest recorded MST (Fig. 4d,e). Broad similarity between our time series of temperature and effective moisture in easternmost Africa and the Hulu–Dongge record of East Asian monsoon intensity (Fig. 4c) indicates a strong climate-dynamical link, namely, the fact that both eastern African and East Asian monsoon intensity were controlled by temporal variation in the SST of their tropical ocean moisture source.

In marked contrast to the glacial era, BIT-index and MST values are inversely correlated during the Holocene (11.7 kyr to present, or broadly encompassing MIS1: $\rho = -0.79$, $P < 0.001$, $n = 62$; Fig. 5a), meaning that in the current interglacial climate mode, higher temperatures have corresponded to lower continental effective moisture and vice versa. Allowing for the approximately 200-year resolution and modest smoothing of our proxy time series (Methods), the difference between the $\rho$ coefficients for the glacial and interglacial sections of the Chala

Hulu–Dongge speleothem record (Fig. 4c), indicate that during HS1 the entire Afro-Asian monsoon domain experienced its most severe sustained drought episode of the past 50 kyr (ref. 41). Our data confirm this general timeframe: the last previous episode with comparably severe drought conditions occurred between about 64 kyr ago and about 57 kyr ago, that is, towards the end of MIS4, similarly coincident with below-average glacial temperatures (Fig. 4d,e). What is different from many other Indian Ocean monsoon records is that late-glacial drought in eastern equatorial Africa ended about 16.5 kyr ago (Fig. 4d), rather than about 15 kyr ago elsewhere[42] (for example, Fig. 4c). We surmise that this is because increasing summer insolation over the North

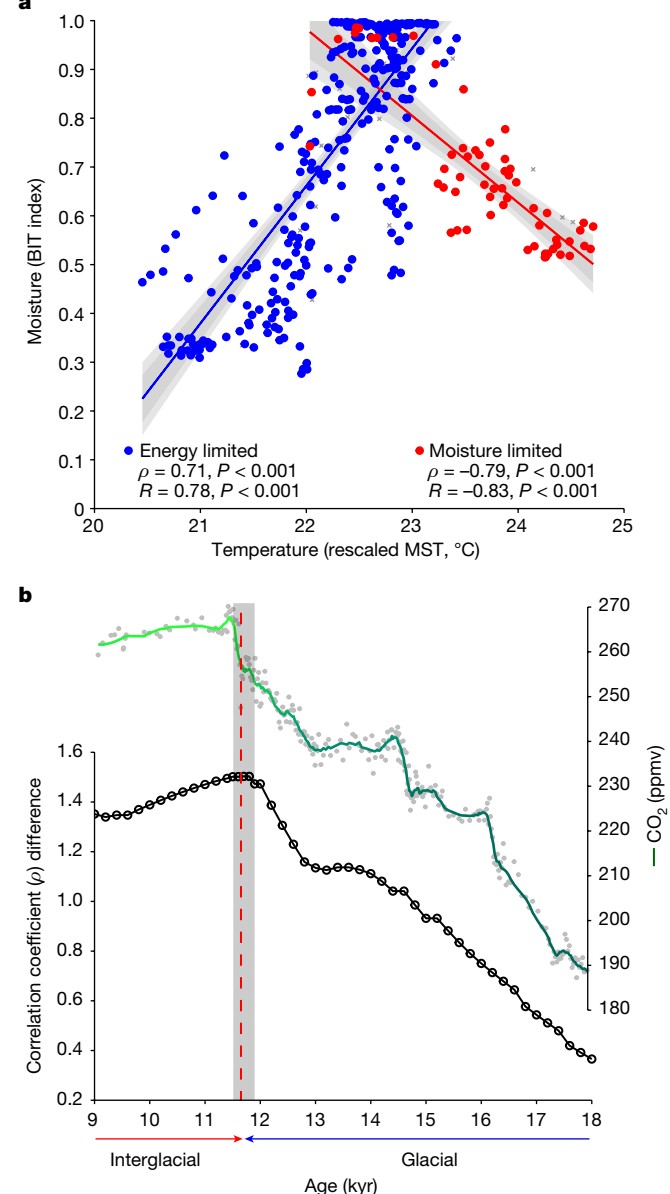

**a**

**b**

**Fig. 5 | Relationships between temperature and moisture-balance variation in easternmost Africa. a**, Linear regressions of BIT index versus rescaled MST in the 75-kyr Lake Chala record with sediment horizons dated to the glacial period (75–11.7 kyr ago) in blue and sediment horizons from the current interglacial period (11.7 kyr to present) in red. Pearson's correlation coefficients (*R*) as well as Spearman's rank correlation coefficients (*ρ*) for the non-parametric relationships are indicated, both with level of statistical significance (*P* values). The proxy time series were smoothed using a five-point rolling mean to accentuate low-frequency variability (Methods). Using the original (that is, unsmoothed) proxy time series (Fig. 4d,e), the corresponding *R* and *ρ* values are 0.66 and 0.62 for the glacial period and −0.67 and −0.59 for the Holocene, respectively, all also statistically significant at *P* < 0.001. The dark and light grey shading encompass two and three standard deviations from the mean, respectively. Sediment horizons partly consisting of thin turbidites (grey crosses) are plotted passively. **b**, Magnitude of the difference between Spearman's coefficients (Δ*ρ*) for glacial and interglacial portions of the paired climate-proxy time series (from **a**) depending on where in time the boundary between them is positioned, in relation to variation in atmospheric $CO_2$ concentration between 18 kyr ago and 9 kyr ago (grey circles)[44] with a ten-point running average (green line).

record is maximized (Δ*ρ* = 1.5) when the separating boundary is positioned between 11.9 kyr ago and 11.5 kyr ago (Fig. 5b), that is, bridging the start of the Holocene. In our data, this timing corresponds to the onset of a rapid temperature increase across the YD–Holocene transition, such that by about 10 kyr ago, regional MST values consistently approached or exceeded today's value of about 23 °C (Fig. 4e). Notably, the onset of the Holocene is also the first time in about 110 kyr (that is, since the end of the previous interglacial period, MIS5) that atmospheric carbon dioxide ($CO_2$) concentrations exceeded 250 ppmv to reach the typical Quaternary interglacial level of 260–280 ppmv (ref. 44; Figs. 4f and 5b).

Comparing our Lake Chala record with paired temperature and hydroclimate reconstructions from five other sites in eastern Africa (Extended Data Table 1), we find that during the glacial period, continental hydroclimate variation was positively related to temperature at all these sites (Fig. 6a). Notably, this relationship has switched to negative during the Holocene only at the three Horn of Africa sites, that is, those located within the part of eastern Africa fully dependent on Indian Ocean moisture (Fig. 6b). Despite noted diversity among the sites in the lipid biomarker proxies used to reconstruct past temperature and hydroclimate variation (Extended Data Table 1), this regional analysis indicates that the climate-proxy record from Lake Chala can be considered representative for late Quaternary climate history in the Horn of Africa. In addition, it reveals that the contrasting relationship in annual *P − E* versus *T* between the wider Horn of Africa region and western and central Africa (that is, areas west of the CAB) evident in historical observational data (Fig. 1a), is also expressed at the longer timescales covered by palaeoclimate reconstructions. Thus, these combined data show that around the onset of the current interglacial period 11.7 kyr ago, the hydroclimate regime of the Horn of Africa experienced a fundamental transition from being primarily energy limited to being primarily water limited. Under the energy-limited regime, which prevailed during the cooler glacial period, positive temperature anomalies enhanced the generation of atmospheric moisture over the western Indian Ocean, as well as monsoonal advection of this moisture onto the continent. Despite this expected positive influence of temperature on monsoonal rainfall[43], above a certain (and probably region specific) threshold temperature that is commonly attained under the present interglacial climate, moisture supply to the atmosphere by terrestrial evaporation no longer meets the exponentially increasing atmospheric demand for water (the Clausius–Clapeyron relation)[45], resulting in a progressive decline of relative humidity and the potential inhibition of precipitation[6]. Increasing temperature may then create even drier, rather than wetter, conditions. Land–atmosphere feedbacks may further reduce the overland recycling of precipitation, locking an already dry tropical climate regime in a drier condition overall[46]. Over time, accumulating rainfall deficits (that is, meteorological drought) can develop into hydrological drought, characterized by declines in groundwater recharge, overland flow and the extent or permanence of surface waters[24]. When persistent, this hydrological drought eventually becomes registered in sedimentary proxies of lake water-balance variation, such as the Chala BIT index (Fig. 4d).

In addition, changes in atmospheric circulation during postglacial warming may have contributed to altering the relationship between effective moisture and temperature in easternmost Africa. Indeed, the recent drying trend in the Horn of Africa[3,4] has been attributed to the Indian Ocean Walker circulation being disrupted by high SSTs in the western Pacific Ocean[4,11,47], or more rapid movement of the ITCZ across the Equator owing to higher summer Indian Ocean SSTs in both hemispheres[9]. Similar atmospheric dynamics may have contributed to on-land precipitation being reduced during the warmer mid-Holocene period relative to the cooler early and late Holocene. Holocene trends in western Indian Ocean SST (Extended Data Fig. 4d) do agree with those of continental MST at nearby Lake Chala (Fig. 4e).

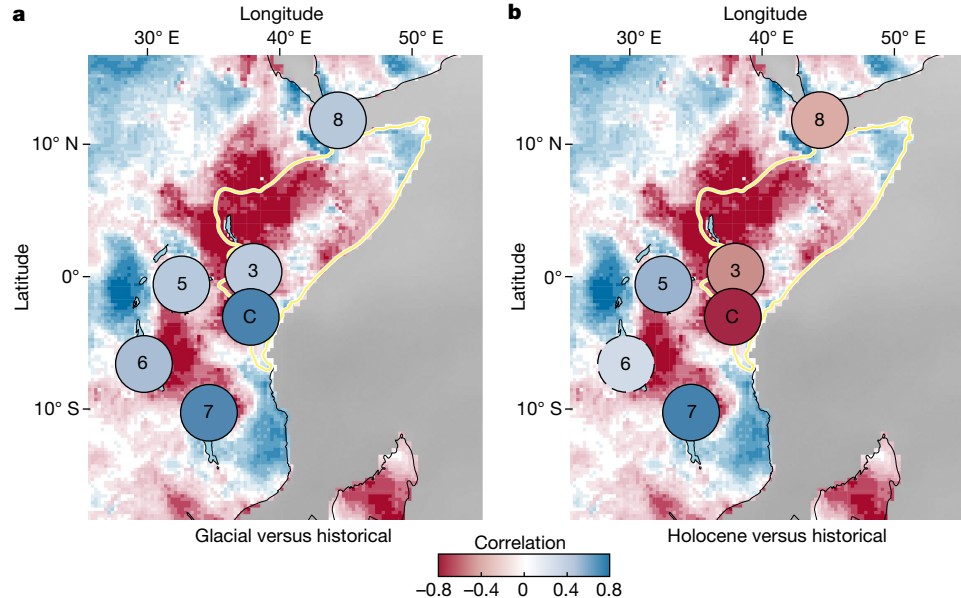

**Fig. 6 | Spatial distribution of the correlation between effective moisture and temperature across eastern Africa based on palaeoclimate reconstructions, compared with that in observational data from 1980 to 2021. a**, Spatial distribution of Spearman's $\rho$ during the last glacial period (coloured circles) based on temperature and hydroclimate proxy time series (Extended Data Table 1) dated to >11.7 kyr from Lake Chala (C, this study), four other eastern African lakes (sites 3 and 5–7 in Fig. 1a) and a deep-sea marine record from the Gulf of Aden (site 8). **b**, Spatial distribution during the Holocene (coloured circles) based on temperature and hydroclimate proxy time series dated to 11.7–0 kyr ago from the same sites. In both panels, circles bordered by a full (dashed) line indicate correlations are (are not) significant at $P < 0.05$ (Extended Data Table 1).

## Future climate in the Horn of Africa

Regardless of the dominant mechanism, the early Holocene transition from energy-limited to water-limited climate conditions in easternmost Africa occurred when the postglacial rise in atmospheric $CO_2$ exceeded about 250 ppmv (Fig. 5b), and mean summer (here closely similar to annual) temperature near Lake Chala approached the modern-day value of 23 °C (Fig. 4e). Referring to the 'dry gets drier, wet gets wetter' paradigm of hydroclimate response to anthropogenic greenhouse gas forcing[6], which has thus far proved difficult to resolve using observational data alone[1,48,49], our results indicate that under the cooler climate regime of the last glacial period the Horn of Africa exhibited a 'wet gets wetter' response to increasing temperature, whereas under the current interglacial temperature regime it has been exhibiting a predominantly 'dry gets drier' response, as is also evidenced by the instrumental record of the past four decades (Fig. 1a). With further increases in anthropogenic greenhouse gas forcing, regional temperatures across the African continent are projected to continue increasing towards the end of the twenty-first century and beyond[1,2] (Fig. 1b). Persistence of a predominantly inverse relationship between effective moisture and temperature would thus imply a continuation of the recent, and ongoing, drying trend in the Horn of Africa, with diminishing water resources and more widespread water scarcity notwithstanding projections of increasing annual rainfall[5]. Our results further suggest that climate models employed to project future climate conditions in tropical dryland regions require better representation of the influence of land–atmosphere interactions on precipitation.

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

## Methods

### Present-day and future regional climatology

Observational data of temperature ($T$), precipitation ($P$) and evaporation ($E$) over the period 1980–2021 were obtained from public data archives. Monthly average daily mean $T$ values are from the Climatic Research Unit archive[54], version 4.06, and based on spatial interpolation of weather station data. In remote regions where stations are sparse (for example, the Congo rainforest) data quality is probably reduced. $P$ data are from the Global Precipitation Climatology Project monthly analysis product[55] version 2.3, and based on integration of multiple satellite datasets over land and ocean with gauge data over land. $E$ data are from the Global Land Evaporation Amsterdam Model[56] version 3.6a, a Priestley and Taylor evaporation model based on satellite and reanalysis data. All three datasets were aggregated to annual scale before computing the Pearson correlation ($R$) between $P − E$ and $T$ (Fig. 1a), and in the case of the Global Precipitation Climatology Project and the Climatic Research Unit, they were linearly interpolated from their original 2.5° and 0.5° resolution to the 0.25° resolution of the Global Land Evaporation Amsterdam Model. End-of-century simulations of changes in $P$, $E$ and $T$ (Fig. 1b) are from the Coupled Model Intercomparison Project phase 6 (CMIP6) ensemble model[51] based on the Shared Socioeconomic Pathway SSP5-8.5 emissions scenario.

### Site context and sediment drilling

Lake Chala (3.3° S, 37.7° E; also written 'Challa' after the nearby village) is located at 880 m above sea level (masl) on the border between Kenya and Tanzania immediately southeast of Mount Kilimanjaro (Extended Data Fig. 1). It fills a steep-sided volcanic caldera basin that probably formed during Mount Kilimanjaro's most recent phase of activity, originally dated to 200–150 kyr ago[57]. Lake Chala has a surface area of 4.2 km$^2$ and a maximum water depth historically varying between 92 m and 98 m (1999–2017), and is maintained against a negative local water balance by subsurface inflow originating from rainfall onto the forested and subalpine slopes of Mount Kilimanjaro[58,59]. Surface inflow is limited to run-off from the inner crater slopes, except that heavy rains can activate a creek breaching the crater rim in its northwestern corner[60]. The lower water column of Lake Chala is permanently stratified (meromictic) and anoxic[60], promoting undisturbed accumulation of fine-grained organic sediments[34,61,62], interrupted by only turbidites and tephra layers[15,63]. The water column of Lake Chala can be divided into six distinct zones, with boundary depths and relative thicknesses mainly determined by the frequency of wind-driven or convective mixing[60] (Extended Data Fig. 2). The thicknesses of zones 1–3 (together forming the mixolimnion, that is, the water mass mixing at least once each year) vary seasonally and interannually and are controlled by wind-driven turbulence, nighttime convection and the rate at which dissolved oxygen is consumed by bacterial decomposition of dead phytoplankton sinking through the water column. By contrast, zones 4–6 (together forming the monimolimnion, mixing at a frequency less than once per year[64]) experience near-constant temperature and chemical stratification.

Analysis of very-high-resolution (3.5 kHz) seismic-reflection data penetrating the entire basin infill, and extrapolation of the [14]C-dated basal age of a 21.70-m sediment sequence collected by the Challacea project in 2003 and 2005[34], indicated that the approximately 210-m-thick deposit of lacustrine sediments in Lake Chala encompasses sedimentation over approximately the past 250 kyr with no or minimal temporal hiatuses[14,65]. In November 2016, the DeepCHALLA[15] project recovered a sediment sequence from the profundal bottom of Lake Chala reaching 214.3 mblf, shortly penetrating the lowermost seismic reflector that is believed to represent the base of the lacustrine deposit[65]. The present study involves the upper portion of this sediment sequence, which consists entirely of soft and fine-grained, organic diatomaceous muds. It was recovered by hydraulic piston coring in two holes (DCH-1A and DCH-1B) with starting depths offset so that overlapping 3-m sections together achieved 100% recovery. Owing to release of hydrostatic pressure during retrieval, the composite core depth (68.39 mcd) exceeds the equivalent drilled depth (mblf) by about 8.2% (Fig. 2). Core splitting and description was done at the US National Lacustrine Core Facility of the University of Minnesota (Minneapolis, USA).

### Lithostratigraphy

The recovered sediments are all laminated either at the millimetre scale (that is, varve-like), at the centimetre scale or alternating between these two facies (Fig. 2), reflecting turbulence-free deposition under oxygen-depleted deep-water conditions throughout. High-resolution images allowed cross-correlation between overlapping core sections with millimetre-scale precision. Excision of all 135 confirmed turbidites (0.5–46.0-cm thick, among which 80 are <2.0 cm)[66] shortens the studied section to 63.19 mefd (Fig. 2). Of these matrix sediments, 52.14 m (82.8%) features lamination that is uniformly millimetre scale, with occasionally thicker light laminae representing an exceptional seasonal diatom bloom[61]. Laminae counting in two multi-millennial (2,609- and 2,510-year) sections of the 25-kyr sequence recovered at the Challacea site (Extended Data Fig. 1) proved that couplets of alternating light and dark laminae (Fig. 2c,d,g,i) often are true varves, that is, they represent one year of deposition[61]. However, owing to muted temperature seasonality at this equatorial location, the deep water-column mixing, which recycles hypolimnetic nutrients for phytoplankton growth, is subject to the relative strength of alternating windy dry seasons and calm rainy seasons[61]. This renders varve structure more complex, so that they can be discriminated unequivocally (and counted accurately) only through microscopic analysis of thin sections at high magnification. Lamination in two intervals of 3.25-m and 0.33-m thick situated between 12.32 mefd and 16.48 mefd (together representing 5.7% of the studied sequence) is predominantly centimetre-scale and often somewhat blurred (Fig. 2e). This suggests that this centimetre-scale banding resulted from post-depositional mixing of originally millimetre-scale lamination. As evidence of bioturbation is lacking in both facies, implying (near-)permanence of bottom-water anoxia, this centimetre-scale banded sediment is interpreted to have been deposited during episodes when the lake occasionally mixed completely with subdecadal frequency, each time creating bottom currents that mixed the uppermost few centimetres of unconsolidated muds. In contrast, millimetre-scale lamination was deposited and preserved intact under a stably stratified lower water column as exists today[60,62]. Finally, 11.5% of the studied section consists of short alternating beds of millimetre-scale lamination and centimetre-scale banding with no obvious threshold thickness to separate both main facies (Fig. 2f,h). In the upper 23 m covered also by the Challacea sequence, this intermediate facies corresponds to intervals with reduced preservation of algal pigments[67], and hence we interpret it to represent a situation where the frequency of complete water-column mixing is on the order of once every few decades. Such instances of complete mixing probably did not inject much oxygen into the lower water column of Lake Chala, because deep-mixing events in tropical lakes are typically short-lived[68], and because the reduced solubility of oxygen in warm water together with intense oxygen consumption by bacterial activity means that deep-water anoxia is typically re-established shortly after the mixing event[64].

### Age markers based on absolute dating

Cross-correlation of millimetre- and centimetre-scale lamination between the Challacea and DeepCHALLA sequences most often allowed precise (subcentimetre scale) transfer of [210]Pb- and [14]C-based age markers from the former[69] to the latter, yielding a well constrained absolute chronology for the DeepCHALLA sequence covering the past 25.2 kyr (ref. 70). This upper portion of our 75-kyr proxy record is supported by 170 absolute age markers, including 162 [14]C-dated horizons, 6 [210]Pb-dated horizons, and the Challacea and DeepCHALLA

sediment–water interfaces (Fig. 2b, right). Twenty-five [14]C-dated horizons from the Challacea sequence were not used, because owing to disturbance caused by hammer-driven piston coring of that sequence[61,63], their corresponding position in the DeepCHALLA sequence (which is perfectly undisturbed: Fig. 2c–i) could not be determined to better than 2-cm accuracy.

Using a preliminary age–depth relationship for the DeepCHALLA drill site[52] based on tracing local seismic stratigraphy back to that at the Challacea coring site[65], the depth interval between 65.90 mcd and 67.80 mcd (60.75–62.53 mefd) was selected for recovery of YTT cryptotephra produced by the super-eruption of the Toba volcano in Indonesia 73.7 kyr ago[25]. First, contiguous 10-cm-long strips of matrix mud (that is, avoiding turbidites) were extracted throughout this approximately 2,000-year interval and processed using a non-destructive tephra extraction protocol[71]. Samples were dried, weighed and sieved to concentrate the >25-μm size fraction, which was then further concentrated using stepped heavy liquid floatation to isolate particles with a density >1.95 g cm[-3]. Extracted residues were mounted on microscope slides for optical analysis, and glass shard abundances were counted under ×100–200 magnification. Core intervals with peaks in glass shard abundance were then resampled at 0.5-cm resolution to determine the exact depth and abundance of glass shards at approximately 10-year resolution. A pronounced peak of about 300,000 shards per gram of dry sediment was identified at 45.0–45.5-cm depth in core section DCH-CHL16-1B-21-H-2 (Fig. 3), and given tephra code DCH-67.55 based on the equivalent composite depth.

DCH-67.55 glass shards were picked manually, set in an epoxy resin mount, ground and polished to expose their cross-section for single-chard analysis of major, minor and trace elements measured using wavelength dispersive spectroscopy on a Cameca SX100 electron probe microanalyser at the University of Cambridge (UK). The instrument was calibrated against a suite of mineral and oxide standards, run with an accelerating voltage of 15 kV, a 6-nA beam current and a defocused 10-μm-diameter beam. Analyses were quantified using Pouchou and Pichoir (PAP) absorption correction[72]. Intermittent measurement of the Max Plank Institute-Dingwell (MPI-DING) standards KL2-G (basalt) and St-Hs6/80-G (andesite)[73,74] and an in-house Lipari obsidian standard (peralkaline rhyolite) were used to test the calibration and to monitor accuracy. The obtained data[75] are visualized in total alkali silica[76] and iron oxide (FeO) versus aluminium oxide ($Al_2O_3$) plots (Extended Data Fig. 3a,b).

Using the same resin mount, shard-specific trace-element compositions were determined using a Thermo Scientific iCapQ laser-absorption inductively coupled plasma mass spectrometer with a Teledyne G2 Eximer laser in the iCRAG laboratory at Trinity College Dublin (Ireland). Analyses used a 40-μm[2] laser spot. The laser was fired at a repetition rate of 5 Hz, with 40-s count times on both sample and gas blank. Concentrations were calculated via calibration against NIST612 and using the concentration of [29]Si, previously determined for each shard via wavelength dispersive spectroscopy electron probe microanalysis (WDS-EPMA), as internal standard. The MPI-DING reference materials GOR132-G (komatiite), St-Hs6/80-G and ATHO-G (rhyolite)[73,74] were analysed to monitor instrument precision and accuracy. Data reduction was performed in Iolite3.4, followed by a secondary matrix correction using calcium[77]. The results[75] are visualized in a biplot of yttrium versus barium concentrations (Extended Data Fig. 3c).

### Age-model construction and validation

The age–depth model of the studied DeepCHALLA sediment sequence (0.00–68.39 mcd, 0.00–63.19 mefd), constructed in rbacon[78], is based on partial varve counting and high-resolution radiometric ([210]Pb and [14]C) dating in the section dated to the past 25.2 kyr (0.00–23.34 mefd); morphological and geochemical identification of 73.7-kyr old[25] YTT cryptotephra recovered at 62.36 mefd; and constraints on long-term variation in the rate of sediment accumulation associated with the

demonstration of depositional continuity[14,65] and overall uniform lithology of soft, either finely laminated or more coarsely banded organic muds (Fig. 2b, left, and c–i). These lithostratigraphical constraints suggest that as a first-order approximation and for the aims of this study, sediment age in the 73.7–25.2-kyr section can be interpolated linearly between the horizons firmly dated to 73.7 kyr ago and 25.2 kyr ago. Average linear sediment accumulation at the DeepCHALLA site during the past 25 kyr is about 12% higher than at the Challacea site[69], with the inter-site difference fairly evenly distributed through time. However, average sediment accumulation in the mostly centimetre-banded interval between 18.4 mefd and 12.4 mefd (1.006 ± 0.067 m kyr[-1]) is 24% higher than in the mostly millimetre-scale laminated interval between 12.4 mefd and 1.5 mefd (0.812 ± 0.013 m kyr[-1]; 1.5 mefd is the approximate base of the sediment compaction zone: Fig. 2b, right, and Extended Data Fig. 5b). This only modestly higher accumulation rate of the mainly centimetre-scale banded section concurs with our interpretation that these centimetre-scale layers are not varves, but originally millimetre-scale lamination impacted by post-depositional disturbance. The slightly higher accumulation rate can be attributed to sediment focusing under the lake low-stand (reduced lake depth) conditions prevailing between 20.5 kyr ago and 14.5 kyr ago[65] (Fig. 2a), the period corresponding with Unit 3 in seismic stratigraphy[14,65] (Extended Data Fig. 5). Lack of a comparable facies change corresponding to inferred low-stands during the YD and the mid-Holocene (Fig. 2a and Extended Data Fig. 5) implies that the latter low-stands were of lesser amplitude than the Unit 3 low-stand. Thus, in terms of sedimentation dynamics, the entire period since 14.5 kyr ago until the present can be treated as representing relative high-stand conditions.

With exception of a 0.85-m-thick interval at 49.81–50.66 mefd displaying alternating millimetre-scale laminated and centimetre-banded sediments (Fig. 2b, left, and 2h), and a similar 10-cm-thick section at 24.11–24.21 mefd, the studied section of the DeepCHALLA sequence below 18.4 mefd (that is, older than 20.5 kyr) is entirely laminated at the millimetre scale, and is often clearly varved (that is, displaying regular light–dark couplets; Fig. 2). Using the well defined age of the YTT (73.7 ± 0.4 kyr ago) to anchor the lower end of this section, the long-term average linear sedimentation rate is 0.825 m kyr[-1], that is, near-identical to that of the section dated to 14.5–1.5 kyr ago in which millimetre-scale lamination (Fig. 2c,d) has been demonstrated to represent varves[61,62]. Moreover, downcore extrapolation of the latter value (0.812 ± 0.013 m kyr[-1]) beyond the [14]C-dated portion of the sequence yields an age estimate of 72.87 kyr ago for the YTT marker layer, that is, only about 800 years younger than its accepted median age of 73.7 ± 0.4 kyr (ref. 25). This small error (1.5%) is at the low end of the commonly accepted range of imprecision in varve counting[79], implying that even if feasible, actual counting of the approximately 53,200 varves in this section of the sequence may not provide substantive additional constraint on its long-term age–depth relationship. We therefore assume a constant rate of accumulation throughout all millimetre-scale lamination between 20.5 kyr ago and the YTT age marker. Accordingly, the Bayesian age model[78] incorporating all absolute age markers (Fig. 2b, right) includes prior settings prescribing an overall average sediment accumulation rate of 1.01 m kyr[-1] between 18.4 mefd and 12.4 mefd, and of 0.81 m kyr[-1] between 23.4 mefd and 18.4 mefd and between 12.4 mefd and the sediment surface. The [14]C-derived age of 25.2 kyr at 23.4 mefd defines the top of the core section where no absolute chronological constraints contribute to the age model besides the YTT at 62.36 mefd. This approach produced an age–depth curve for the studied 75-kyr core section with 2σ uncertainty ranges between ±15 years and ±365 years (on average ±200 years) in the past 12.5 kyr and ±145–380 years (on average ±285 years) in the period 25.2–12.5 kyr ago, that is, comparable to those obtained on the Challacea sequence[69]. Modelled 2σ age uncertainty is substantially larger in the section between 25.2 kyr ago and 73.7 kyr ago, as can be expected

when absolute age markers between the two end-points are lacking. Age uncertainty exceeds ±1,000 years between 28 kyr ago and 75 kyr ago, and peaks at ±2,100 years around 50 kyr ago (Fig. 2b, right). However, the actual age–depth trend in this interval is unlikely to deviate outside the modelled minimum or maximum ages. First, a hypothetical age–depth curve tracing the maximum age solution would imply a sedimentation rate averaging 0.88 m kyr$^{-1}$ (10% higher than the 73.7–25.2-kyr mean) during the approximately 22-kyr period after the Toba eruption, combined with a sedimentation rate averaging 0.74 m kyr$^{-1}$ (9% lower than the 73.7–25.2-kyr mean) during the approximately 27-kyr period leading up to 25.2 kyr ago. Such sustained and opposing changes in sedimentation rate are unlikely to have occurred without a change in lithological facies ever interrupting the millimetre-scale lamination, given the near-complete change to centimetre-scale banding in the 20.5–14.5-kyr interval when the sedimentation rate was on average 24% higher (Fig. 2b, right). Second, any sustained increase (decrease) in sedimentation rate during this 48.7-kyr period would need to have been exactly balanced by a compensating decrease (increase) in sedimentation rate such that the overall mean value of the 73.7–25.2-kyr section (0.825 m kyr$^{-1}$) matched the mean value of the absolutely dated 14.5–1.5-kyr section (0.812 m kyr$^{-1}$) close enough for our successful prediction, based on linear extrapolation[52,65], of the depth at which the (not macroscopically visible) YTT cryptotephra would be found. Thus, our assumption of constant sediment accumulation between 73.7 kyr ago and 25.2 kyr ago as currently modelled (Fig. 2a) probably closely approaches the actual age–depth relationship in this section of predominantly varved-like sediments, at least at the century-scale resolution of the climate-proxy time series presented in this study.

## GDGT analysis and climate proxies

In total, 396 sediment horizons of 2-cm thickness and sampled at regular 16-cm intervals were selected for GDGT analysis at the Royal Netherlands Institute for Sea Research. Of these, 23 sediment horizons were later determined to partly consist of thin turbidites not yet excised from the composite sequence at the time of sampling[66], and were analysed for GDGTs but excluded from the final climate-proxy time series[80]. Freeze-dried and powdered sediments (0.3–1 g dry weight) were extracted using a Dionex accelerated solvent extraction system with a 9:1 (v/v) mixture of dichloromethane (DCM) and methanol. An internal standard (1 µg of synthetic $C_{46}$ glycerol trialkyl glycerol tetraether) was added to the total lipid extract[81]. The total lipid extracts were dissolved in DCM, passed through a sodium sulfate column and dried under nitrogen gas. They were then subjected to chromatography using an $Al_2O_3$ column with eluents of hexane/DCM (9:1, v/v), hexane/DCM (1:1, v/v) and DCM/methanol (1:1, v/v) to obtain apolar, ketone and polar fractions, respectively, which were dried under nitrogen gas. The GDGT-containing polar fractions were redissolved in hexane/isopropanol (99:1, v/v) and filtered using a polytetrafluoroethylene 0.45-µm filter before analysis using an Agilent 1260 Infinity ultrahigh-performance liquid chromatography (UHPLC) system coupled to an Agilent 6130 mass spectrometer[82]. GDGTs were identified by detecting the $[M + H]^+$ ions in selected ion monitoring mode for the relevant $m/z$ values. Peak area integration was done using Agilent Masshunter software and a peak area of $3 \times 10^3$ units was applied as the detection threshold. The fractional abundances of the individual brGDGTs were calculated relative to the sum of all 15 brGDGTs (roman numerals refer to structures defined elsewhere[83]) and are expressed in proxy equations using square brackets. The variability in the fractional abundance of brGDGTs in the sediment record was compared with that in suspended particulate matter (SPM) from the Lake Chala water column, sampled at 13 depth intervals and collected monthly for a period of 17 months (from September 2013 to January 2015; $n = 141$)[21] using principal component analysis in the R statistical package FactoMineR[84].

## Chala BIT index as moisture-balance proxy

Changes in hydrological moisture balance, or effective moisture, in eastern equatorial Africa were inferred from the BIT index[20], expressed as the ratio between the summed abundances of the five brGDGTs not containing cyclopentane moieties and that of crenarchaeol (cren):

$$\text{BIT index} = ((\text{Ia}) + (\text{IIa}) + (\text{IIa}') + (\text{IIIa}) + (\text{IIIa}'))/((\text{cren}) + (\text{Ia}) + (\text{IIa}) + (\text{IIa}') + (\text{IIIa}) + (\text{IIIa}')) \quad (1)$$

where GDGTs in brackets refer to peak areas derived from UHPLC–mass spectrometry (MS) analysis.

On the short timescale of modern-system studies[21–23,60,85], the relative proportion of these GDGTs is controlled by the seasonal succession of strong upper-water-column stratification (oxycline at about 10–15 m) and deep mixing (oxycline at about 45–50 m)[61]. This seasonal cycle is timed by the latitudinal migration of low-latitude convective activity[17], such that high BIT-index values reflect episodes of strong stratification and shallow oxycline during the rainy seasons with slack winds, and low BIT-index values reflect episodes of deep mixing and a depressed oxycline during the windy principal dry season[22] (Extended Data Fig. 2a,b). At longer timescales, periods with a relatively dry or wet climate regime will also promote a low or high BIT-index signature, respectively, to be exported to the sediments. Moreover, variation in climatic moisture balance at these longer timescales causes changes in surface run-off and groundwater recharge[17], which, when of sufficient magnitude and duration, affect the annual water balance of Lake Chala and cause the major fluctuations in lake surface level (and thus lake depth) evident in seismic stratigraphy[14,34,65] (Fig. 2a and Extended Data Fig. 5). Importantly, this lake-level fluctuation also affects the relative sizes of the oxygenated and anoxic parts of the water column, such that greater lake depth will increase the proportion of the anoxic zone where brGDGTs are produced relative to the upper mixed water layer, resulting in higher-BIT-index values. Conversely, during lake low-stands, the niche for brGDGT producers is contracting, resulting in lower-BIT-index values (Extended Data Fig. 2c–e). Hence these two mechanisms, namely, (1) changes in lake mixing as a reflection of varying monsoonal rainfall, and (2) changes in overall water-column structure in relation to lake depth, exercise a synergistic control on BIT-index variation in Lake Chala. This latter mechanism is probably becoming more important when moving away from the seasonal scale to the timeframe of palaeoclimate reconstruction. Hence, the BIT index in Lake Chala can be used as a semi-quantitative hydroclimate proxy reflecting changes in continental effective moisture at the timescale of the present study. However, this application of the BIT index cannot be directly generalized to the sediment records of other lakes as it would require detailed study of the response of the local microbial communities to seasonal and longer-term changes in water-column stratification.

During long periods of reconstructed very wet climatic conditions (for example, about 52–40 kyr ago; Fig. 4d), the abovementioned mechanisms relating BIT-index variation in Lake Chala sediments to long-term hydroclimate variation appear to be affected by limited sensitivity of the BIT-index proxy. Predominance of strong upper-water-column stratification during such periods may have resulted in near-complete absence of Thaumarchaeota in Lake Chala over extended intervals of time, and hence a lack of crenarchaeol production resulting in sustained maximum BIT-index values approaching 1.0 (ref. 22). Thus, in these circumstances of unusually wet regional climate conditions (relative to the long-term mean; Fig. 4d), the Chala BIT index becomes a categorical hydroclimate proxy such as is the case with seismic stratigraphy[65] and lithology (Fig. 2a and 2b, left). As a means of reconstructing temporal variation in effective moisture, the Chala BIT index is complementary to organic biomarker proxies thought to more directly reflect past variation in the amount of continental rainfall[86] (Extended Data Table 1).

However, in context of the present study, the former is arguably the more pertinent climate variable to assess, especially when considering the wider ecosystem and societal impacts of water scarcity associated with hydrological and agricultural drought[87].

## BrGDGT-based temperature reconstruction

Both soil[88] and lake[89] datasets have revealed that the degree of methylation of brGDGTs is largely determined by temperature, such that warmer climates generally give rise to a higher abundance of less methylated brGDGTs. This degree of methylation (expressed by the MBT index) probably impacts the fluidity of the bacterial membrane and modification of the distribution of brGDGT membrane lipids and therefore reflect a physiological adaptation to temperature[88].

$$MBT = [Ia] + [Ib] + [Ic] \quad (2)$$

The application of well established temperature calibrations for soils[83,88] to lake sediments results, however, in poor estimations[90] because these GDGTs are not derived from soil erosion but produced predominantly in situ in most lakes[21,85,90–94] and respond differently to temperature. Hence, alternative temperature calibrations were developed that are based on brGDGT distributions in a global collection of surface sediments from lakes, and these have also found application[30,95]. Specifically, best subset regression has been used to select those brGDGTs that predict MST with the smallest error[19]:

$$MST = 20.9 + 98.1 \times [Ib] - 12 \times ([IIa] + [IIa']) \\ - 20.5 \times ([IIIa] + [IIIa']) \quad (3)$$

($R^2 = 0.88$; root mean squared error (RMSE) = 2.0 °C).

Three other calibrations derive mean annual air temperature (MAAT) from brGDGT distributions in surface sediments from 111 East African lakes[95], using correlations between the fractional abundances of different sets of brGDGTs and MAAT:

$$MAAT = 36.90 - 50.14 \times ([IIIa] + [IIIa']) - 35.52 \times ([IIa] + [IIa']) \\ - 0.96 \times [Ia] \quad (4)$$

($R^2 = 0.88$; RMSE = 2.7 °C).

$$MAAT = 2.54 + 45.28 \times MBT - 5.02 \\ \times (-\log_{10}(([Ib] + [IIb] + [IIb'])/([Ia] + [IIa] + [IIa']))) \quad (5)$$

($R^2 = 0.87$; RMSE = 2.8 °C).

$$MAAT = 22.7 - 33.58 \times ([IIIa] + [IIIa']) - 12.88 \times ([IIa] + [IIa']) \\ - 418.53 \times ([IIc] + [IIc']) + 86.43 \times [Ib] \quad (6)$$

($R^2 = 0.94$; RMSE = 1.9 °C).

Subsequent improved chromatographic separation of brGDGTs[82] revealed that the penta- and hexamethylated brGDGTs have two prominent alternative positions of additional methyl groups, either at the fifth or sixth carbon positions (referred to as the 5-Me and 6-Me brGDGTs)[83,96]. Reassessment of the relationship of brGDGT distributions and environmental parameters established that the methylation of the 5-Me brGDGTs most strongly relates to temperature in soils[83] and lake sediments[97–99]. Consequently, a revised MBT' index, the $MBT'_{5Me}$ index[83], was defined, which captures the degree of methylation of only the 5-Me brGDGTs:

$$MBT'_{5Me} = ([Ia] + [Ib] + [Ic])/([Ia] + [Ib] + [Ic] + [IIa] + [IIb] + [IIc] + [IIIa]) \quad (7)$$

Reanalysis of surface sediments from 70 East African lakes[97] resulted in two new calibration models. The first model predicts MAAT directly from the $MBT'_{5Me}$ –MAAT correlation:

$$MAAT = -1.21 + 32.42 \times MBT'_{5Me} \quad (8)$$

($R^2 = 0.94$; RMSE = 2.14 °C).

The second (based on stepwise forwards selection) applies the fractional abundance of a subset of brGDGTs:

$$MAAT = 23.81 - 31.02 \times [IIIa] - 41.91 \times [IIb] - 51.59 \times [IIb'] \\ - 24.70 \times [IIa] + 68.80 \times [Ib] \quad (9)$$

($R^2 = 0.92$; RMSE = 2.43 °C).

Most recently, the $MBT'_{5Me}$ index of brGDGTs in surface sediments from 272 globally distributed lakes has been used to develop a new Bayesian calibration ($R^2 = 0.82$; RMSE = 2.9 °C) to reconstruct mean air temperature during the months above freezing (MAF)[98].

However, application of these more up-to-date calibrations of the brGDGT palaeothermometer to the 75-kyr DeepCHALLA record to reconstruct either MAAT or MAF did not show obvious resemblance to known global temperature trends over glacial–interglacial time-scales: all three records show an anomalously warm late-glacial period and YD (about 17–11.7 kyr ago), and the lowest temperatures of the last glacial period are recorded around 50 kyr ago (Extended Data Fig. 6). These ambiguous results prompted further consideration of the available calibration models. Long-term monitoring of the Lake Chala water column has revealed vital information about the sources and behaviour of brGDGTs[21,22,60,85]. An extensive study of SPM from 13 depth intervals in the 90-m water column collected monthly for a period of 17 months confirmed that brGDGTs are produced abundantly in the anoxic lower part of the water column, and are strongly impacted by seasonal shifts in the depth of the oxycline associated with episodes of stratification and deep mixing of the upper water column[21] (Extended Data Fig. 2, zones 2 and 3). Whereas the 6-Me brGDGTs (IIa', IIb' and IIIa') are most abundant in the permanently stratified lowermost anoxic water layers (zones 4–6), the 5-Me brGDGTs (IIa and IIb) most abundantly occur in zone 3 (the seasonally anoxic zone), but their abundance is strongly reduced during seasonal deep mixing[21] (Extended Data Fig. 2b). This implies that the 5-Me and 6-Me producers occupy distinct ecological niches, probably related to oxygen-controlled chemical gradients varying with depth. Considering the climatic sensitivity of Lake Chala[14,65], significant declines in lake level at decadal to millennial time-scales probably had an effect analogous to that of seasonal deep-mixing events[21,22], thus annihilating zone 3 and thereby critically impacting the habitat of 5-Me brGDGTs producers (Extended Data Fig. 2d). As events of deeper mixing and of reduced lake level both restrict the habitat of 5-Me brGDGT producers, calibrations based on the distribution of only 5-Me brGDGTs are probably less effective at estimating temperature accurately during such episodes. Indeed, during the late-glacial period (about 20–15 kyr ago) of strongly reduced lake depth[65], temperatures estimated by several $MBT'_{5Me}$ based calibrations are unexpectedly high (Extended Data Fig. 6c).

Importantly in this context is that, albeit not as strong as for the 5-Me brGDGTs, the degree of methylation of 6-Me brGDGTs also shows moderate correlation with temperature[98,99]. In both Lake Chala SPM and sediments the fractional abundance of 6-Me brGDGTs contributes significantly to the variance in the distributions of brGDGTs (Extended Data Fig. 7). This is in clear contrast with lake surface-sediment calibration datasets, such as the recently expanded global lakes dataset[98], in which the acyclic 5-Me brGDGTs play the dominant role in determining the variance of the distributions of brGDGTs. Considering the potential temperature response of the 6-Me brGDGTs and their relative importance in Lake Chala, the potential of earlier brGDGT-based temperature calibrations for lakes, which include the fractional abundances of 6-Me brGDGTs[19,95], were considered. Application of the East African lake calibrations[95] (equations (4–6)) yields temperature reconstructions that are all fairly flat throughout the past 75 kyr, except for two periods of (much) colder climate conditions towards the ends of MIS4 and

MIS2, and a more modest temperature depression centred around 35 kyr (Extended Data Fig. 6b). By contrast, the global lakes calibration[19] (equation (3)) yields a temperature reconstruction with similar timing of the lowest recorded temperatures, but also clear differentiation between a cooler glacial climate and a warmer Holocene climate, and a feasible timing of the glacial–interglacial transition (Extended Data Fig. 6a). Therefore, we deemed this calibration as best suited for reconstructing past temperature variation at Lake Chala. We surmise that this global lakes calibration dataset encompasses a range of climate states, as well as niches for aquatic brGDGT producers, that are not present in East Africa today but are required for reliable reconstruction of past temperature variation using brGDGT distributions.

Although perhaps counterintuitive, there are proper arguments for why the East African lakes calibrations[95] may be less suitable for application at Lake Chala. Of the 111 lakes included, the majority are shallow and mix completely at least once per year[100]; only 36 lakes (32%) are deeper than 20 m, and overall average depth is about 24 m. Also, 50% are mid-elevation (2,000–3,000 masl) or high-elevation (>3,000 masl) lakes, resulting in an overall average elevation of about 2,460 masl. Therefore, their suitability for temperature reconstruction at a deep (90 m) and permanently stratified tropical lowland lake such as Lake Chala (880 masl) is not necessarily greater than the global lake temperature calibration[19], which also includes shallow lakes but with elevations ranging from 2.5 masl to 2,260 masl and, perhaps more importantly, also includes deeper permanently stratified lakes in cold-temperate climate regimes.

The brGDGTs in DeepCHALLA sediments were analysed using an UHPLC–MS technique capable of separating the 5-Me and 6-Me isomers[82]. To apply a calibration developed before this separation was achieved[19], the fractional abundances of the 5-Me and 6-Me isomers of penta- and hexamethylated brGDGTs were summed (as expressed in equation (3)). In principle, the summed concentration of the individually quantified 5-Me and 6-Me isomers should equal that of the co-eluting 5-Me and 6-Me isomers using the earlier HPLC–MS method. However, this exercise may be complicated by the presence of minor isomers and/or partial co-elution of the 5-Me and 6-Me isomers. To test this, we applied the global lakes calibration[19] to brGDGT data from the 25-kyr Challacea sequence[101], which was obtained using an earlier analytical method that combined the quantification of the co-eluting the 5-Me and 6-Me isomers (that is, with the same analytical technique used to develop this calibration). The resulting Challacea MST record is nearly identical to that obtained from the DeepCHALLA sequence (Extended Data Fig. 8), with small differences in MST between them relating only to the amplitude, not to the timing or trend of the temperature change. This excellent match is convincing, especially considering that these records derive from two different sediment cores recovered from sites located about 650 m apart (Extended Data Fig. 1), not simply duplicate analysis of the same sediment samples.

Notwithstanding confidence that our 75-kyr Lake Chala MST reconstruction reflects the actual temporal trends of past temperature change in easternmost Africa, the inferred amplitude of temperature change (12 °C, range 16–28 °C) is unrealistically large. For comparison, cooling of the global tropics during the LGM is generally accepted to have been on the order of 2–3 °C at sea level relative to today[30,31], and even in high-latitude regions the LGM to Holocene transition involved 'only' about 7 °C of warming[26]. This overestimation of the absolute range of past temperature change in easternmost Africa is not unique to the chosen global temperature calibration (equation (3)), as also other calibrations including 6-Me brGDGTs infer temperature ranges between about 9 °C (MBT'$_{5Me}$ calibration, equation (8))[97] and about 22 °C (stepwise forwards selection calibration, equation (9))[95] (Extended Data Fig. 6). Considering that seasonal limnological transitions in Lake Chala already cause notable shifts in its aquatic microbial community[21,22], and hence in sedimentary brGDGT distributions, the unknown but almost certainly substantial shifts in the microbial community during the glacial period may not be well represented in brGDGT-based temperature calibrations using only recently deposited interglacial sediments. To compensate for the overestimation, we scaled temperature change across the glacial to Holocene transition in our 75-kyr MST time series to that reconstructed at 7 other eastern African lakes for which GDGT-based temperature reconstructions exist: Garba Guracha[102], Sacred[95], Rutundu[30], Mahoma[103], Victoria[104], Tanganyika[105] and Malawi[106] (Fig. 1a and Extended Data Fig. 4). To correct for differences in elevation, these reconstructions were first normalized by shifting the mean temperature at each site to zero. Subsequently temporal temperature anomalies were averaged among all records per 100-year interval and then smoothed using 5-point running means to create a regional ensemble reconstruction for eastern tropical Africa (Extended Data Fig. 4b). The minimum (−2.8 °C) and maximum (+2.0 °C) temperature anomalies averaged over 3,000-year intervals covering the cold early late-glacial period (21–18 kyr ago) and warm mid-Holocene period (7–4 kyr ago) infer a temperature change of 4.8 °C across the glacial–Holocene transition. The 75-kyr Lake Chala MST record was rescaled to this range, while preserving the original MST estimate of 23.4 °C for the youngest sediment horizon analysed (sample mid-depth dated to −0.04 kyr (ref. 80), encompassing the period of about AD 1980–2000). This value matches the present-day MAAT in the immediate vicinity of Lake Chala (23.7 °C during the period 2007–2010; ref. 60) within calibration error (±2 °C, rescaled to ±0.7 °C). Modest uncertainty on the accuracy of this rescaling does not affect the principal outcomes of the present study, which focuses on climate-dynamical implications of the temporal variation in temperature trends, their correlation with temporal variation in the hydroclimate proxy (that is, BIT index) and the timing of changes relative to those observed in global reference records.

## Regional assessment of paired hydroclimate–temperature records

We assessed how hydroclimate has been related to temperature during the Holocene (the past 11.7 kyr) and during the last glacial period (that is, before 11.7 kyr) at five other sites in eastern Africa besides Lake Chala from where paired temperature and hydroclimate reconstructions of sufficient temporal reach are available: the Gulf of Aden[107,108] and lakes Rutundu[30,86], Victoria[104], Tanganyika[105,109] and Malawi[110,111] (Extended Data Table 1). The proxy time series from lakes Rutundu, Tanganyika and Malawi were interpolated before correlation analysis to avoid having to remove data points that lack paired temperature–hydroclimate proxy values. For a proper comparison, the Lake Chala proxy time series were smoothed using a five-point rolling mean to mimic the time integration of individual data points in the other records, thereby accentuating the longer-term trends (compare with Fig. 5). In the unsmoothed Chala BIT-index and MST time series, each data point integrates over about 25 years of sedimentation at approximately 200-year intervals, and thus retains a high degree of the short-term variability expressed in a proxy record with approximately 25-year temporal resolution sampled contiguously (compare with Extended Data Fig. 5a). In proxy records from bioturbated or physically mixed sediments (Gulf of Aden, Rutundu and Victoria) this short-term variability is erased (that is, the proxy time series are smoothed in situ), and when data points integrate over longer periods of time owing to lower sedimentation rates (Tanganyika and Malawi), the smoothing is imposed at the sampling stage.

## Data availability

The three principal sets of data presented in this paper are publicly accessible via Zenodo[70,75,80].

## Code availability

No specific code has been used. Statistical data treatment has been performed with commercially available software, as mentioned in Methods.

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

**Acknowledgements** This research was co-financed by NESSC Gravitation Grant 024.002.001 from the Dutch Ministry of Education, Culture and Science (OCW) to J.S.S.D.; by Ghent University Collaborative Research Operation grant BOF13/GOA/023, BRAIN-be project BR-121-A2 from the Belgian Science Policy Office (BelSPO), Hercules infrastructure grant AUGE/15/14-G0H2916N from the Research Foundation of Flanders, and a Francqui research professor mandate to D.V.; by UK Natural Environment Research Council standard grant NE/P011969/1 to C.S.L.; by the Max Planck Society to G.H.H.; and by the International Continental Scientific Drilling Program through the DeepCHALLA project (https://www.icdp-online.org/projects/world/africa/lake-challa/). D.G.M. acknowledges support from the European Research Council under grant agreement 101088405 'HEAT' and the EU Horizon 2020 project 869550 'DOWN2EARTH'. A.M. was supported by a PhD scholarship from the Graduate Studies Authority and Department of Marine Geosciences at the University of Haifa. Recovery of the Lake Chala sediment record was facilitated by the government of Kenya through permit P/16/7890/10400 from the National Commission for Science, Technology and Innovation (NACOSTI), license EIA/PSL/3851 from the National Environmental Management Authority (NEMA), and research passes for foreign nationals issued by the Department of Immigration; and by the government of Tanzania through permits NA-2016-67 (270–285) and NA-2016-201 (277–292) from the Tanzania Commission for Science and Technology (COSTECH), permit EIA/10/0143/V.I/04 from the National Environmental Management Council, and resident permits issued by the Immigration Department. The lake-drilling operation was subject to environmental impact assessments conducted by Kamfor (Nairobi, Kenya) and Tansheq (Dar es Salaam, Tanzania), and permission from the Lands and Settlement Office of Taita-Taveta County (Kenya) to use government land as staging area. We thank all DeepCHALLA partners not directly involved in this work for project facilitation; C. M. Oluseno, the 'Air Force One' team, the Kamba and Taveta communities of Lake Chala area, and all field scientists for assisting in the lake-drilling operations; and the National Lacustrine Core Facility (LacCore) at the University of Minnesota (USA) for organizing the splitting, logging and initial processing of core samples. We further thank A. Mets and J. Riekenberg (NIOZ) for analysis of sedimentary biomarkers, I. Buisman (University of Cambridge) and E. Tomlinson (Trinity College Dublin) for tephra geochemical analyses, G. De Cort (Ghent University) for help with coding in R, I. Petrova (Ghent University) for providing CMIP6 climate model output, T. Markus (Utrecht University) for help with map drafting, and D. McLeod (Cardiff University) for feedback on the interpretation of hydroclimatic projections.

**Author contributions** D.V., J.S.S.D. and G.H.H. devised the project and acquired funding. F.P., J.S.S.D. and D.V. developed principal methodology and supervised analyses. Formal analyses were executed by A.J.B., D.G.M., A.M., C.M.M.-J., T.v.d.M., M.V.D. and D.V. A.J.B. and D.V. led the data interpretation, using data processing and visualization provided by A.J.B. and D.G.M. A.J.B. and D.V. wrote the paper with substantial contributions by F.P., D.G.M. and J.S.S.D. All other authors contributed to paper review.

**Competing interests** The authors declare no competing interests.

**Additional information**
**Correspondence and requests for materials** should be addressed to A. J. Baxter.

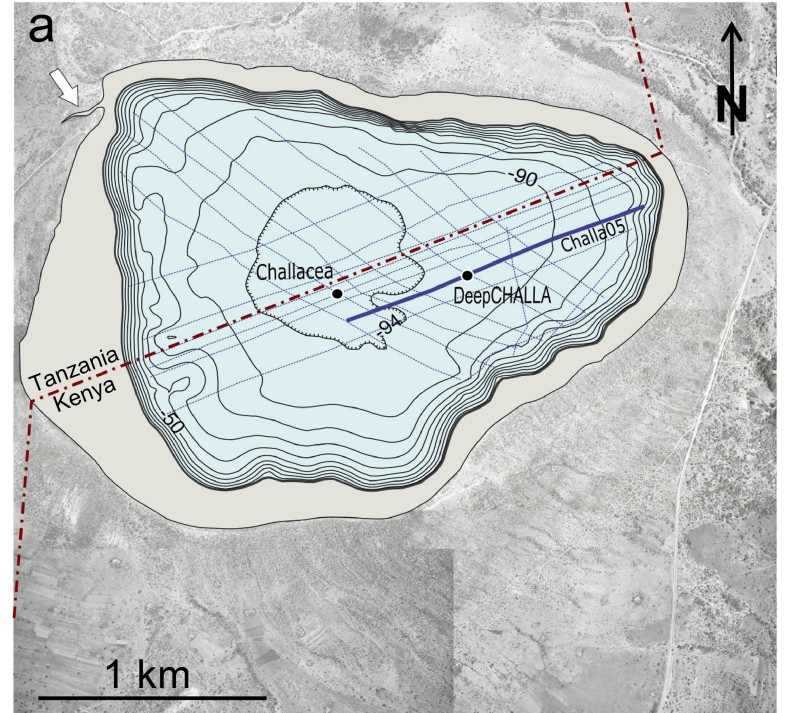

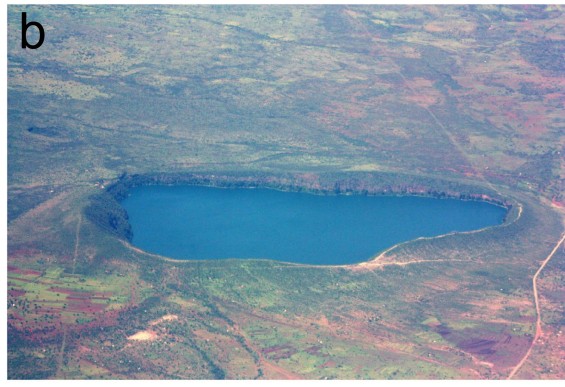

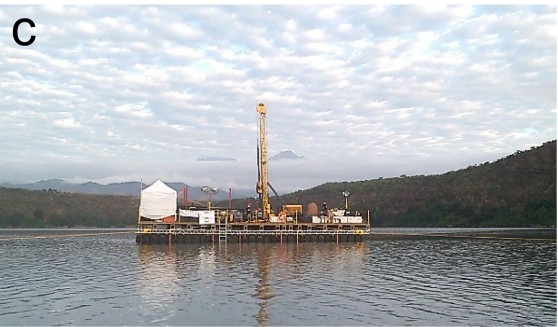

**Extended Data Fig. 1 | Local setting of the study site. a**, Lake Chala and its catchment bridging the Kenya/Tanzania international border and surrounded by a partly cultural landscape covered with dry scrub savannah vegetation. Lake bathymetry (depth contours at 10-m intervals) is based on the seismic survey grid completed in 2003 (thin stippled blue lines)[65], and locations of the Challacea[34] and DeepCHALLA[15] drill sites (closed circles) are shown in relation to the Challa05 seismic profile[14] (Extended Data Fig. 5). White arrow points to an ephemeral creek breaching the north-western corner of the crater rim, which when active extends the catchment modestly beyond Chala crater. Background are collated aerial photographs from March 1993 taken by Photomap (Kenya) Ltd. **b**, Oblique aerial view of Lake Chala within its steep-sided crater basin, viewed from the south (Wikimedia Commons). **c**, Drilling platform of the DeepCHALLA project on Lake Chala, with the Kibo and Mawenzi summits of Mt. Kilimanjaro peeking through morning clouds (photo credit: D. Verschuren).

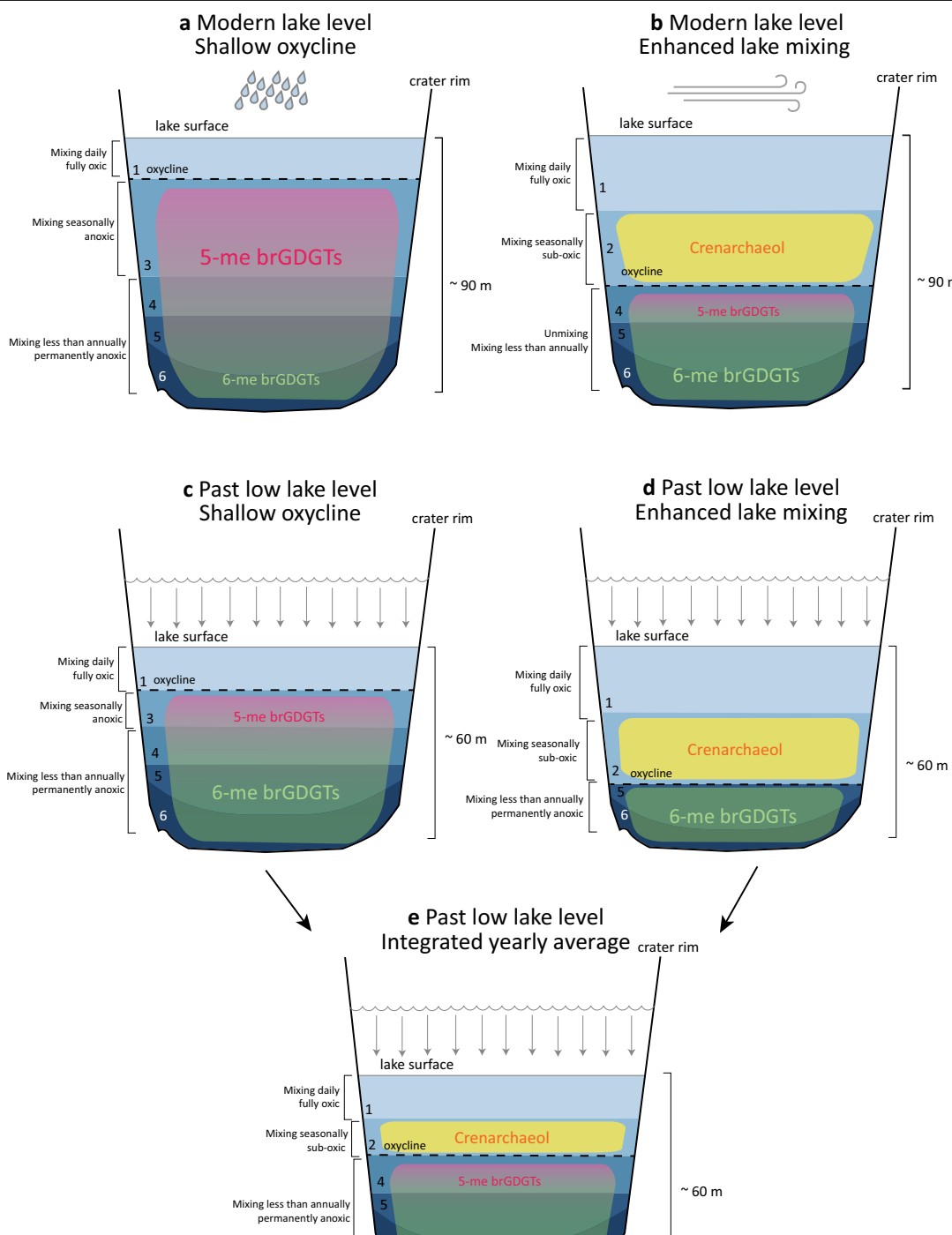

**Extended Data Fig. 2 | Schematic representation of GDGT distribution in the water column of Lake Chala under different conditions of mixing and lake depth.** Light blue to dark blue shading indicates the six depth zones between the daily mixed layer (1) and the near-bottom environment (6), with decreasing frequency of mixing and oxygenation[60], that delimit spatially distinct niches for the microbes producing crenarchaeol (yellow shading) and 5- or 6-Me brGDGT isomers (purple to green shading). **a**, Approximate positions of depth zone boundaries and microbial habitats under conditions of strong upper water-column stratification (shallow oxycline), as occurs today during rainy seasons. **b**, Approximate position of depth zone boundaries and microbial habitats during deep mixing (and deepening oxycline), as occurs today in the main dry season during boreal Summer. **c**, Hypothesised position of depth zone boundaries and microbial habitats during the rainy season under past lake low-stand conditions. **d**, Hypothesised position of depth zone boundaries and microbial habitats during the main dry season under past lake low-stand conditions. **e**, Hypothesised mean-annual position of depth zone boundaries and microbial habitats during past lake low-stands, producing the GDGT distributions reflecting a climate regime substantially drier than today.

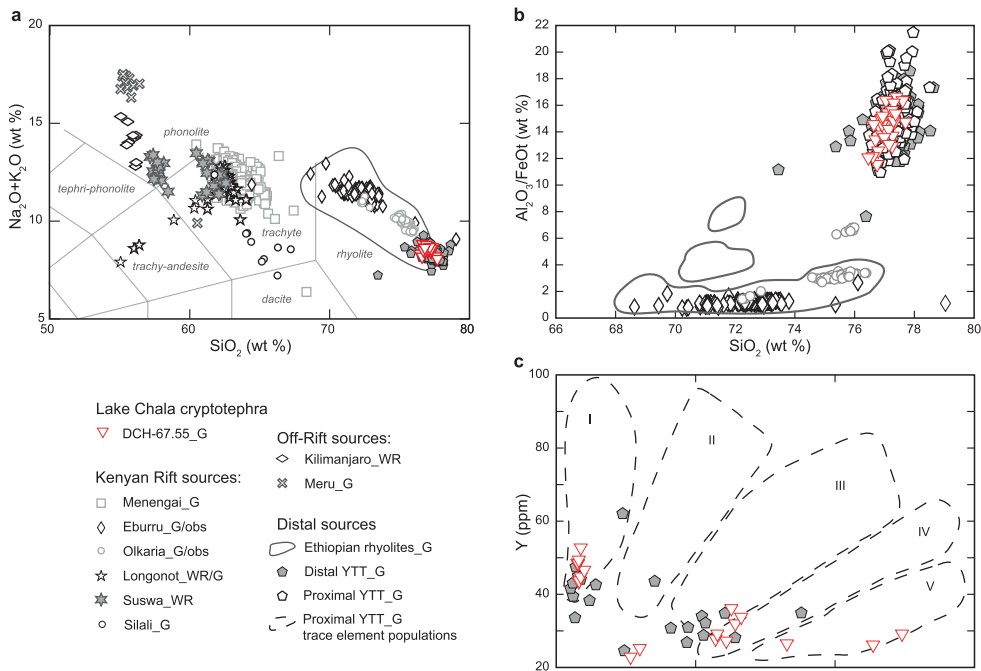

**Extended Data Fig. 3 | Geochemical correlation of cryptotephra DCH-67.55 to the 73.7-kyr YTT[25] based on major, minor and trace-element glass shard composition. a**, TAS plot[76] comparing major element composition of DCH-67.55 to published compositions of YTT[112–114] and of evolved (phonolitic, trachytic and rhyolitic) volcanic products from Middle to Late Pleistocene eruptions of Central Kenyan Rift[115–123], Northern Tanzanian Off-Rift[52] and Main Ethiopian Rift[124] volcanoes. Reference compositional data represent tephra glass shards (G) where available, supplemented by obsidian (obs) and whole rock (WR) samples where data are limited. DCH67.55 is discriminated from all eastern African tephra sources in the TAS plot except for Ethiopian rhyolites, which can be separated from YTT using FeO and Al₂O₃ compositions (**b**). Comparative Y and Ba compositions (**c**) confirm agreement of DCH-67.55 to YTT sub-populations I-IV, as recognized in YTT samples from Malaysia and the Indian Peninsula[125].

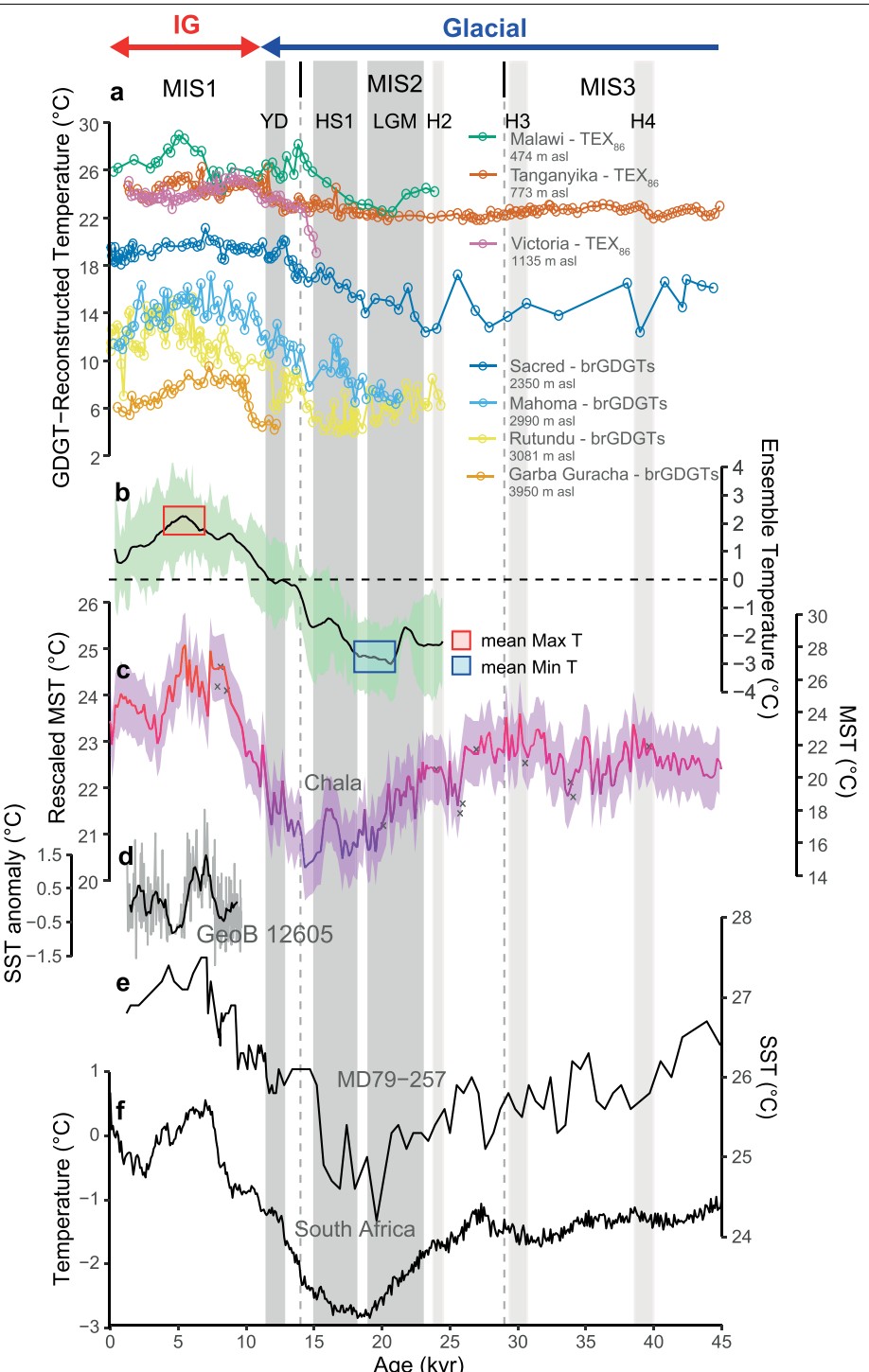

**Extended Data Fig. 4 | Temperature reconstructions from eastern African lakes used to rescale the *ca* 75-kyr MST record from Lake Chala, and from other reference sites. a**, Lake-based temperature reconstructions from eastern African lakes (sites 1–7 in Fig. 1a): Garba Guracha[102], Sacred[95], Rutundu[30], Mahoma[103] (all derived from branched GDGTs), Victoria[104], Tanganyika[105] and Malawi[106] (from the TEX$_{86}$ index based on the distribution of isoprenoidal GDGTs; Extended Data Table 1). **b**, Ensemble regional temperature reconstruction for eastern Africa during the last 25 kyr, calculated by averaging normalized temperature trends from the seven reconstructions shown in **a**. The blue and red frames cover 3000-year intervals of the cold early late-glacial period (21-18 kyr ago; average temperature anomaly −2.8 °C) and warm mid-Holocene (7-4 kyr ago; average temperature anomaly +2.0 °C) which together encompass a total amplitude of 4.8 °C regional temperature change across the glacial-Holocene transition. **c**, Lake Chala MST time series for the last 45 kyr, estimated using BSR of branched GDGT distributions in a global lakes calibration dataset[19]

and rescaled to match the amplitude of the regional ensemble reconstruction (Methods). Purple shading shows the ±2 °C RMSE envelope of this calibration, similarly rescaled to ±0.7 °C. Data from samples partly consisting of thin turbidites are indicated as grey crosses and excluded from interpretation. **d**, Sea surface temperature (SST) anomaly record for the equatorial western Indian Ocean (site 8 in Fig. 1a) based on the magnesium to calcium (Mg/Ca) ratio of planktonic foraminifera in deep-ocean sediments off northern Tanzania (grey line) with 400-year rolling mean (black line)[126]. **e**, Alkenone-based SST record for the Mozambique Channel[127] (site 9 in Fig. 1a). **f**, Regional temperature reconstruction for southeastern Africa based on fossil pollen assemblages[128] (site 10 in Fig. 1a). The timing of the Younger Dryas (YD), Heinrich-1 stadial (HS1), the Last Glacial Maximum (LGM) and marine isotope stages 1–3 (MIS1-MIS3) are indicated for reference. Considering linear age interpolation in the >25.2-kyr section of the proxy time series shown in **c** (Methods), the timing of Heinrich events H2–H4 is approximative only (as indicated by gradated light grey shading).

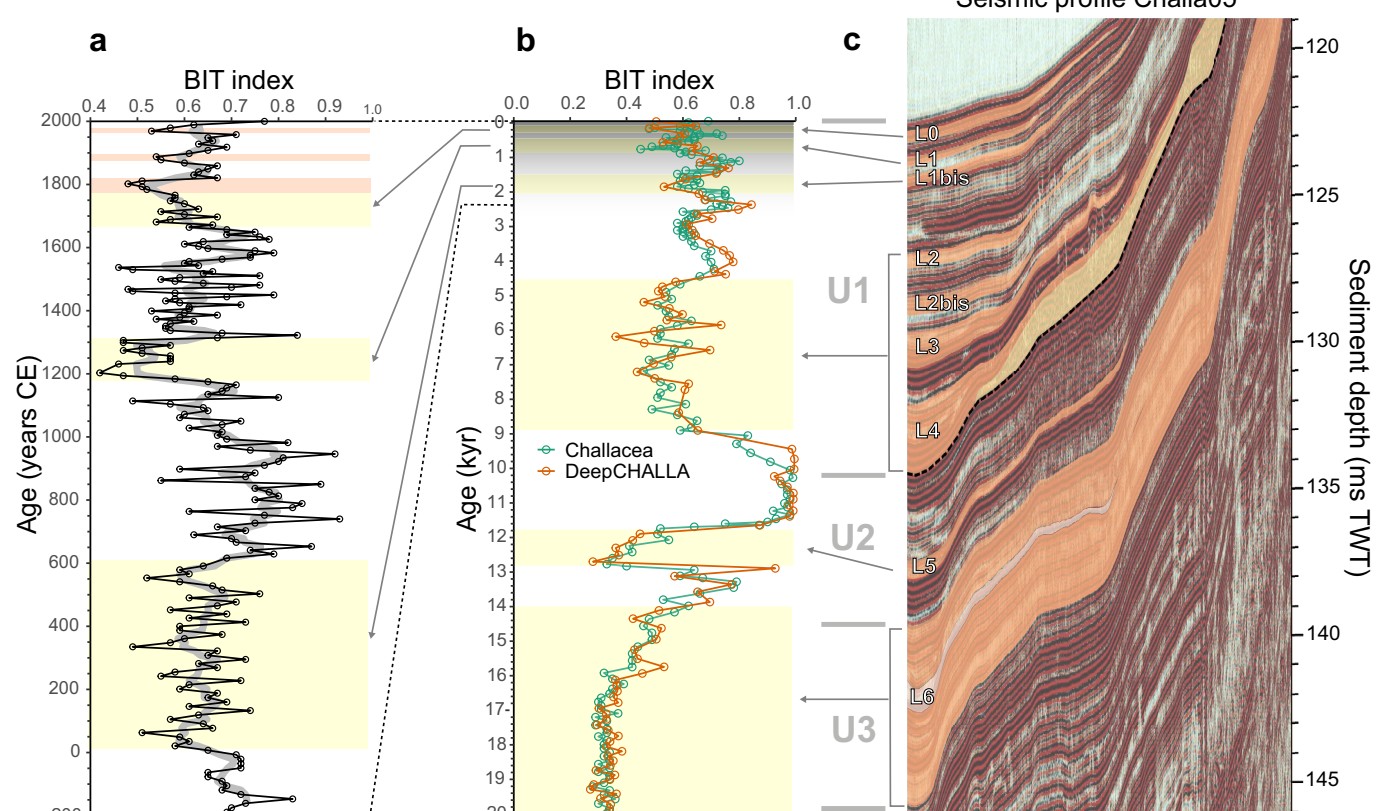

**Extended Data Fig. 5 | Coherence between the BIT-index record and seismic stratigraphy of past lake-depth fluctuation in Lake Chala, two independent proxies of past variation in climatic moisture balance.**
**a**, BIT-index record of the last 2150 years at decade-scale resolution (Challacea data[129]), with three episodes of relative drought (majority of BIT-index values <0.65) highlighted in yellow. These periods correlate to lenses L0, L1 and L1bis of basin-focused (i.e., lake low-stand) sedimentation in the uppermost ~2 m (122–125 milliseconds two-way travel time of the seismic-reflection signal, ms TWT) of seismic-reflection stratigraphy at the DeepCHALLA drill site (orange highlights in **c**; modified from ref. 14). The three short-lived BIT-index minima highlighted in pink are coeval with three historically documented episodes of widespread regional drought within the last 200 years[130]. **b**, BIT-index records of the last 20 kyr at century-scale resolution from the Challacea (green[34]) and DeepCHALLA (orange; this study) sites, with prolonged episodes of relative

drought highlighted in yellow and correlated to lenses L0-L5 and Unit 3 (containing L6) of basin-focused sedimentation in the uppermost ~23 m (122–146 ms TWT) of seismic-reflection stratigraphy (**c**). Note that the TWT scale is not a truly linear depth scale, as it is influenced by the relative strength of successive reflectors. The excellent match between the two 20-kyr BIT-index records from drill sites ~650 m apart (Extended Data Fig. 1) demonstrates i) successful transfer of [210]Pb/[14]C-based chronology from the Challacea sequence[69] to the equivalent section of the DeepCHALLA sequence through visual cross-correlation of finely laminated sediments shared by both; and ii) considering that the DeepCHALLA site is closer to shore than the Challacea site, and can thus be expected to receive a greater influx of GDGTs produced in catchment soils, the near-identical BIT-index values at both sites indicate that all brGDGTs used in the calculation of the Chala BIT index must result from microbial production within the water column of Lake Chala[21].

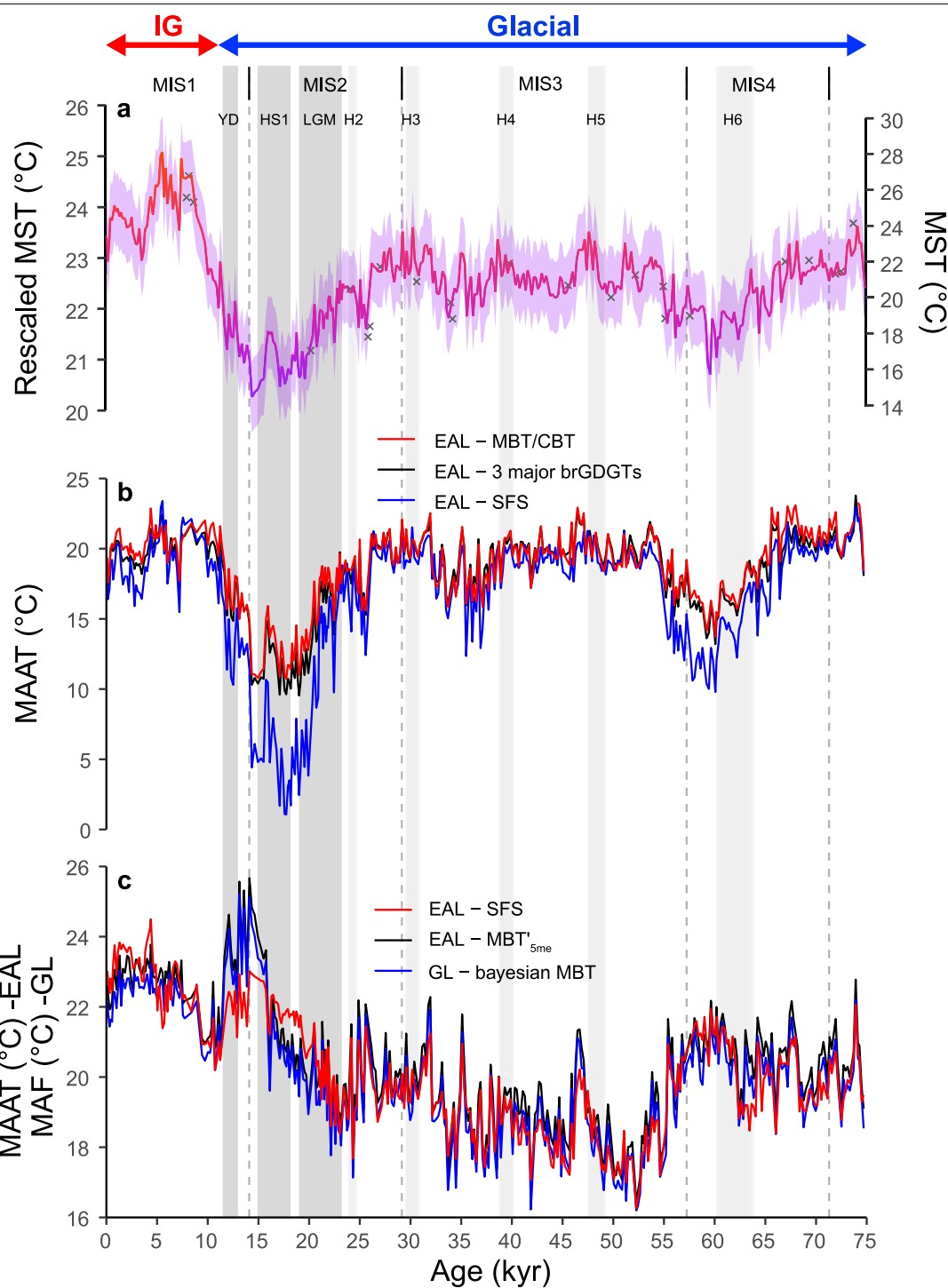

**Extended Data Fig. 6 | Application of different brGDGT-based temperature calibrations to the time series of brGDGT distributions in the *ca* 75-kyr DeepCHALLA record. a**, MST reconstruction based on the calibration of brGDGT distributions (using both 5-Me and 6-Me isomers) in surface sediments of 90 globally distributed lakes developed using BSR[19], and absolute values rescaled (left y-axis) to match the temperature range of the eastern African ensemble reconstruction (Extended Data Fig. 4b). Purple shading represents the root RMSE of the original regression model, ±0.7 °C after rescaling. **b**, MAAT reconstructions based on calibration of brGDGT distributions (including both 5- and 6-Me isomers) in surface sediments of 111 East African lakes (EAL), using correlations between the fractional abundances of three different sets of brGDGTs and MAAT[95]. **c**, MAAT and MAF reconstructions based on updated calibrations of brGDGT distribution (separating 5- and 6-Me isomers) in surface sediments of East African lakes (EAL; red and black, Eqs. 8–9)[97] and of globally distributed lakes (GL; blue)[98]. The timing of the Younger Dryas (YD), Heinrich-1 stadial (HS1) and Last Glacial Maximum (LGM) and marine isotope stages 1–4 (MIS1-MIS4) are indicated for reference. Considering linear age interpolation in the 25.2–73.7-kyr section of the proxy time series (Methods), the timing of Heinrich events H2–H6 should be viewed as approximative (as indicated by gradated light grey shading).

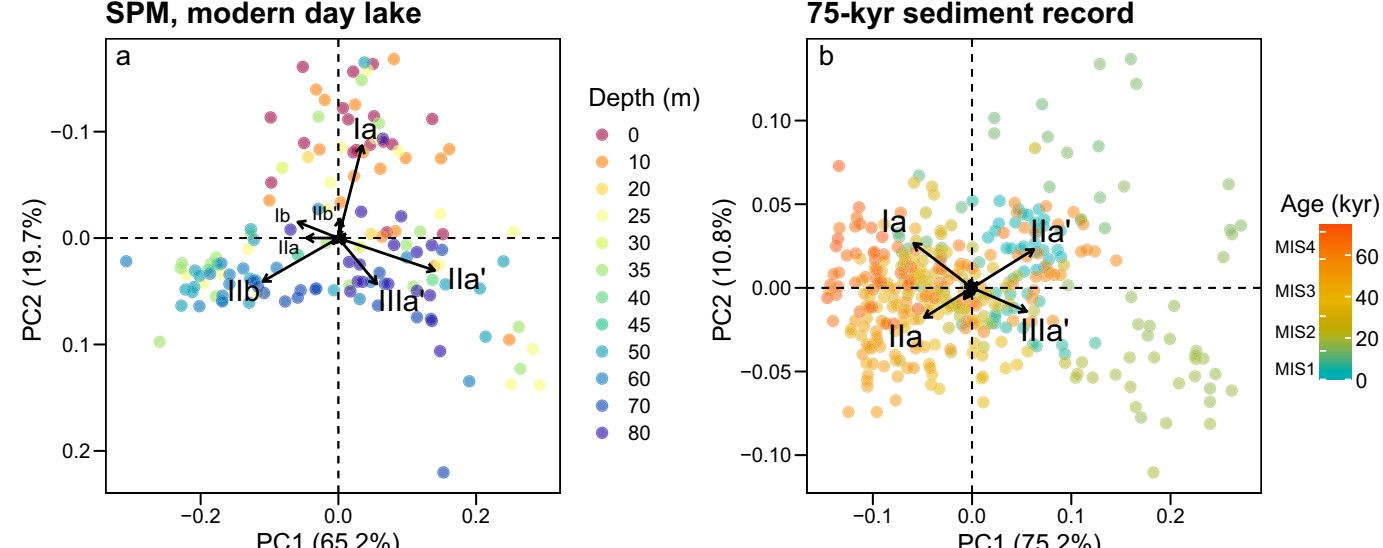

**Extended Data Fig. 7 | Comparison of GDGT distributions in suspended particulate matter (SPM) of modern-day Lake Chala and in sediments covering the last *ca* 75 kyr. a**, Principal component analysis (PCA) biplot showing PC1 and PC2 of the fractional abundances of brGDGTs in SPM collected monthly between September 2013 and January 2015 from surface water and 11 discrete depth intervals between 10 m and 80 m. Data from previously published study[21], with PC scores of individual samples coloured according to water depth. **b**, PC1 and PC2 resulting from PCA of the fractional abundances of brGDGTs in the studied portion of the DeepCHALLA sediment sequence, excluding 23 samples which partly consist of thin turbidites (Methods), and with individual samples coloured according to age, using the Marine Isotope Stage (MIS) boundaries as reference.

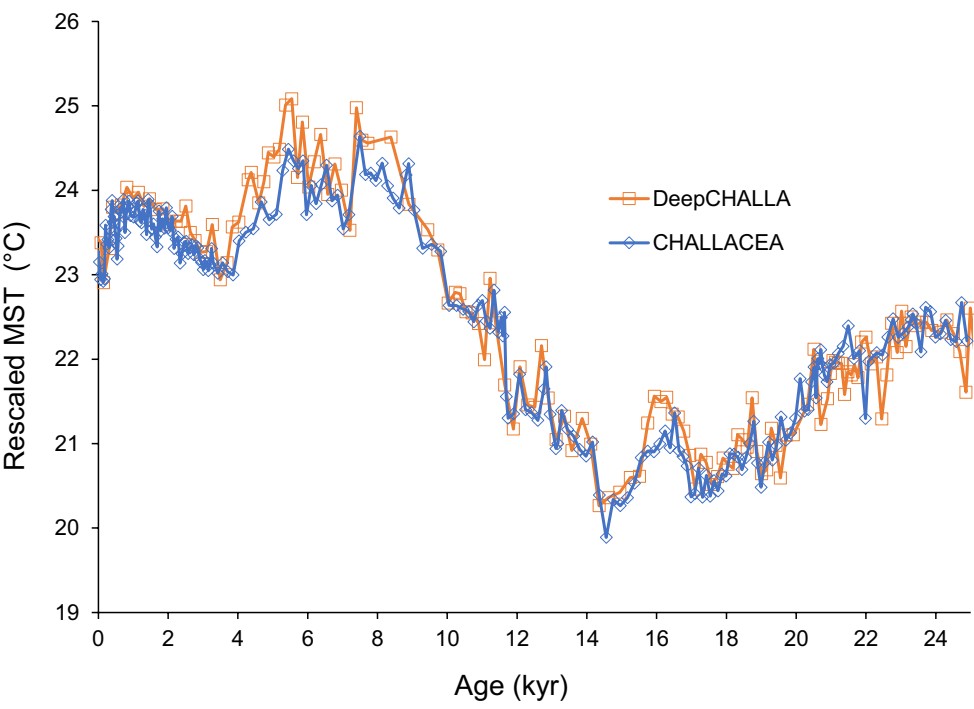

**Extended Data Fig. 8 | Comparison of MST reconstructions for the last 25 kyr from Lake Chala.** Reconstructions are based on brGDGT distributions in sediment sequences from respectively the Challacea[101] (blue) and DeepCHALLA (orange, this study) sites in Lake Chala (Extended Data Fig. 1), and both generated using the global lakes calibration[19].

**Extended Data Table 1 | Paired hydroclimate-temperature reconstructions from eastern Africa used to assess the relationship between these climate variables during the Holocene and the last Glacial period at the regional scale**

| Proxy records | | | | Holocene (11.7–0 kyr) | | | Glacial (>11.7 kyr) | | | Reference |
|---|---|---|---|---|---|---|---|---|---|---|
| Site | Period covered | Temperature proxy (biomarker used) | Hydroclimate proxy (biomarker used) | Spearman's ρ Pearson's R | Significance | # Data points | Spearman's ρ Pearson's R | Significance | # Data points | |
| **Lake Chala** | 75-0 kyr | MST (branched GDGTs) | BIT index (GDGTs) | $\rho = -0.79$ $R = -0.83$ | $p < 0.001$ $p < 0.001$ | 62 | $\rho = 0.71$ $R = 0.78$ | $p < 0.001$ $p < 0.001$ | 311 | This study |
| **Gulf of Aden** | 39-0 kyr | SST-$U^{K'}_{37}$ (alkenones) | $\delta D_{wax}$ ($C_{30}$ fatty acid) | $\rho = 0.42$ $R = 0.35$ | $p = 0.003$ $p = 0.013$ | 48 | $\rho = -0.31$ $R = -0.32$ | $p < 0.039$ $p < 0.031$ | 45 | 106-107 |
| **Lake Rutundu** | 24-0 kyr | MAAT (brGDGTs) | $\delta D_{wax}$ ($C_{29}$ alkane) | $\rho = 0.50$ $R = 0.46$ | $p < 0.001$ $p < 0.001$ | 59 | $\rho = -0.29$ $R = -0.33$ | $p < 0.021$ $p < 0.007$ | 63 | 30,85 |
| **Lake Victoria** | 15-0 kyr | LST-$TEX_{86}$ (isoprenoid GDGTs) | $\delta D_{wax}$ ($C_{28}$ fatty acid) | $\rho = -0.49$ $R = -0.48$ | $p = 0.022$ $p = 0.024$ | 22 | $\rho = -0.29$ $R = -0.41$ | $p = 0.493$ $p = 0.308$ | 8 | 103 |
| **Lake Tanganyika** | 59-0 kyr | LST-$TEX_{86}$ (isoprenoid GDGTs) | $\delta D_{wax}$ ($C_{28}$ fatty acid) | $\rho = -0.11$ $R = -0.21$ | $p = 0.535$ $p = 0.219$ | 35 | $\rho = -0.41$ $R = -0.33$ | $p < 0.001$ $p < 0.001$ | 159 | 104,108 |
| **Lake Malawi** | 22-0 kyr | LST-$TEX_{86}$ (isoprenoid GDGTs) | $\delta^{13}C_{wax}$ ($C_{29}$-$C_{33}$ alkanes) | $\rho = -0.74$ $R = -0.79$ | $p < 0.001$ $p < 0.001$ | 23 | $\rho = -0.69$ $R = -0.73$ | $p < 0.001$ $p < 0.001$ | 21 | 109-110 |

Including this study the respective temperature reconstructions[30,104,105,107,110] use three different lipid biomarkers (isoprenoid GDGTs, branched GDGTs, alkenones) and five different calibrations. The paired Gulf of Aden reconstruction combines a spatially integrated view of terrestrial hydroclimate[108] with local SST[107]. In the four studies employing $\delta D_{wax}$ as hydroclimate proxy[86,104,105,108] this biomarker is considered to primarily reflect the amount of precipitation, while in other lake studies it has been interpreted to reflect past variation in moisture source[131] or effective moisture[132], or to contain an imprint of past changes in relative humidity[133]. Period covered refers to the interval of paired data in each record. Pearson's coefficient (R) expresses linear correlation, Spearman's rank correlation coefficient (ρ) indicates the strength of non-parametric regression, and the statistical significance of all relationships is indicated with p values. Note that more negative $\delta D_{wax}$ and $\delta^{13}C_{wax}$ values reflect wetter climate conditions, hence negative R and ρ values indicate a positive relationship between hydroclimate and temperature.