## [Peer Review File · Nature]

Manuscript Title: Reversed Holocene temperature–moisture relationship in the Horn of Africa

Editor Notes: Author responses to points raised or summarized by the editor have been removed, therefore the numbering of author replies is not continuous throughout this file

Reviewer Comments & Author Rebuttals

Reviewer Reports on the Initial Version:

Referee's comments:

Referee #1 (Remarks to the Author):

Baxter et al. provide important geological time-scale context for ongoing debates on the causative mechanisms for modern and future changes in temperature and effective moisture in eastern Africa. They use sedimentary records from scientific drill cores obtained from a unique environmental and depositional setting, Lake Chala, adjacent to Mount Kilimanjaro and on the border of Kenya and Tanzania. Such sampling projects present extraordinary logistical and funding challenges, and require an expansive network of collaborators in different fields to achieve successful outcomes. The Chala site is particularly valuable because its restricted watershed and high aspect-ratio morphology faithfully record environmental conditions over geologically meaningful time periods, which cross multiple glacial-interglacial intervals. Moreover, its position east of the Congo Air Boundary means Lake Chala is influenced principally by regional Indian Ocean climatological conditions, simplifying the palaeoclimatological record and allowing robust comparisons to climate modeling efforts.

The scientific achievements of the Chala team are exceptional and the community has long looked forward to seeing its important outcomes. The work presented herein is part of the forthcoming story and is a thoughtful and rigorous analysis. With the current manuscript I have questions and concerns about the age model as presented, the rather limited proxy analyses included herein, and aspects of the synthesis.

The age model presented relies on an exquisite record of 162 ^{14}C ages over the last 25000 years, shallow ^{210}Pb ages, and the presence of the Younger Toba Tuff (73.7 ka). Notably, the ^{14}C ages were transferred into the DeepChalla project records from an earlier study (Challacea project); accordingly there is a gap in absolute ages beyond 25,000 years. The authors argue that there is extreme uniformity in sedimentation rates over the 75 kyr records presented herein, and that additional ages are not necessary.

The presence of the Youngest Toba Tuff is presented as confirmed in the Chala record, and the authors refer to shard density and geochemical fingerprinting results as definitive. However the actual data behind this postulation are not represented, even in the Supplemental, and work characterizing the YTT and other tephra analyses are cited as 'in prep'. Given the vast distances of airborne transport to eastern Africa for this singular event, the potential for confusion with other east African tephtras, and its importance for this study, primary data on the YTT and/or definitive citations are seemingly required here.

The authors argue that additional age dating in the interval 25-73 ka is not required, yet it is

incongruous that they would present such a detailed and exceptional stratigraphy for the last 25,000 years from their previous work, and not include new observations from these newer drill cores, which were acquired with extraordinary effort. The authors proceed to interpret their proxies (see below) with an exceptional level of confidence and granularity. They may be correct about the veracity of the deeper age model, but given the extent of the interpretative effort, further verification of the age-depth relations between 73 ka and 25 ka is warranted.

In the published work of Martin-Jones (2020) there is reference to several other tephras identified in the Chala cores over this depth range. Could any of these other tephras represent regional (East African) eruptive events as time markers? Is there not the possibility of incorporating tephra stratigraphy in the age model?

Given the intensity of the analytical work on the shorter cores from previous projects, it is surprising that there are no additional radiocarbon results from parts of the core that are older than 25 ka. Are there any other absolute ages possible or available between 25 ka and YTT? For instance with the marked advances in luminescence dating in recent years are there any results to include here? Is it possible to employ varve counting to further constrain the age model? My sense is that there must be a more robust age model forthcoming that may better justify the highly granular level of thoughtful interpretations and climate forcings considered in this paper.

The authors analyze lipids from prokaryotes to assess past temperatures and hydroclimates in the Lake Chala sediment cores (specifically the distribution of the GDGT's). I will let other experts evaluate their analyses of branched GDGTs for summer mean temperatures. The authors utilize the so-called BIT index as a measure of effective moisture. When initially applied to paleoclimate studies, the BIT index was characteristically used to assess the relative quantity of soil-derived Archaea washed into lake basins, and thus was used to validate other biomarker methods (e.g. TEX86) that were employed to assess lake surface paleotemperatures. Here in Chala however, the authors assert that the BIT index is a direct moisture indicator, as a consequence of several ancillary studies of the water column and sediments deposited over the past 250 years. The authors may indeed be spot-on here, however it is easy to conceptualize other scenarios from deep-time when this proxy may not have been produced under identical circumstances. Moreover, the BIT values presented in this record are clipped at a maximum for extended periods, thus clearly limiting its sensitivity to evaluate maximally wet environments. It is surprising that the authors have not considered employing a multiproxy approach to assessing past hydroclimate conditions in Chala. For instance there are numerous other measurements that must be available as part of their broad suite of analyses that could contribute to quantifying past hydroclimate, and perhaps be integrated using a basic PCA analysis. For instance turbidite frequency or data derived from scanning XRF measurements (Ti and other elements) are commonly employed to indicate terrigenous inputs into lacustrine and marine environments. Accordingly I am baffled why such important work is not utilizing other proxy techniques to better validate the environmental forcings.

I find the conclusions very thoughtful and informative, especially with respect to the later Quaternary interval. I remain suspicious however of the correlations to high latitude intervals prior to 25 ka because of the lack of absolute age dating in the deeper section, as discussed above. The evolutive spectra I found to be of limited value as there is little change in the primary forcing over

the duration of the record. There is some change in power of the precessional forcing of MST over the length of the record, but not much change in the signal of the BIT proxy. This may be in part due to the limited sensitivity of this measurement as a moisture proxy. The half-precessional signal seems comparatively weak, and in the case of BIT, the half-precessional signal barely rises above background.

In summary this is an exceptional project, but expect there must be additional data in-hand (or close to in-hand) that would alleviate my unease about the authors' sole reliance on BIT for a moisture proxy, and on a purely modeled age for the interval 26-73 ka. I hope they will consider including such data in any manuscript revision.

Line specific comments:

44-60 The authors generalize their study site as representative of "East Africa"; however it is well known that there are many sub-regional influences on climate in this part of the continent, including impacts of Atlantic moisture, vagaries of elevation, variable orographic effects on different parts of the monsoon cycle, etc. Whereas possible they should use the term "equatorial East Africa" rather than "East Africa", so as not to overrepresent or geographically overextend the importance of their results.

Manuscript-specific comments by line number:

61only A few....

71 of EQUATORIAL East Africa

119-120 Without additional absolute ages in the 25-75 ka range, I question this statement.

151-151 This is most important, but prefer to see it framed by additional age constraints.

169-177 Run-on statement. Please break this up and clarify.

197-199 Important point!

250-254 Suggest the authors expand on this in the context of the recent multiyear interval of extraordinary high lake levels across the central and northern Kenya rift valley.

Referee #2 (Remarks to the Author):

In this manuscript, Baxter et al. provide high-resolution GDGT records from well-dated sediment cores from Lake Chala, East Africa. The results show the variabilities of temperature and hydroclimate in East Africa during the past 75,000 years. The authors argue that the correlation between temperature and hydroclimate are positive before 11,700 years ago but suddenly shifted to negative, which is linked to a tipping point that the pCO₂ reached 250 ppmv. This is a well-written paper, providing some important and interesting insights into the African climate in the past and future. However, I have some concerns about the proxy interpretations.

For the temperature reconstructions, the authors select the Pearson et al. (2011) calibration that was established prior to the new method that could differentiate the 5- and 6- methyl brGDGTs. In the Supplementary Discussion, the authors provide some arguments, but I still have some questions:

1. Does the sum of 5- and 6-methyl actually equal the intensity of using the old method? There seems to be some systemic bias when doing a such calculation. For example, Fig. 4 in De Jonge et al. (2014) and Fig. 2 in Russell et al. (2018).

2. The East Africa calibrations in Loomis et al. (2012) are excluded because the authors argue that there are not many lakes in the dataset that are similar to Lake Chala (SI Lines 382-392). However, only 16 African lakes are used in Pearson et al. (2011) calibration, which contains 90 lakes in total. Moreover, most of the lakes are way shallower than Lake Chala (Table 1, Pearson et al. (2011)). Therefore, the implication of 6-methyl may not be well constrained in this calibration, given the 6-methyl brGDGTs tend to be produced more in the anoxia deep water (Weber et al., 2018). To date, we've known that the 5- and 6-methyl are mainly produced at different depths, as the authors show in Fig. S2a. It is certainly believable that 6-methyl brGDGTs could play a role in reflecting water temperatures. However, we still do not have enough data to quantitatively support this hypothesis. In the meanwhile, the SFS calibration in Russell et al. (2018) seems to work well for the Lake Chala SPM samples (van Bree et al., 2020).

3. The records using EAL calibrations (Russell et al., 2018) (Fig. S1c) make sense in a way that the temperature started to increase around the end of LGM which agrees with the onset of Rwenzori glacier deglaciation (Jackson et al., 2019). In fact, the ensembled temperatures in Extended Data Figure 3b also show this feature, which contrasts with the Lake Chala MST.

4. I'm also curious about the temperature rescaling process. It seems like melt-water input from the retreating Mt. Kilimanjaro is a plausible reason for the Lake Chala cooling phase between 20 and 13 ka BP (Sinninghe Damsté et al., 2012), which is also apparently the coolest period in the 75-kyr MST records. How would this potentially affect the rescaling since it's hard to estimate the contribution from the meltwater?

I also have some thoughts on the interpretation of the BIT index. First of all, the representative of BIT is rather vague. It's not very clear to me whether is this a proxy for precipitation amount, P-E, lake level, or other stuff. Moreover, the schematic mechanism in Figure S2 is a good hypothesis but needs more testification. For example, Fig. c Past low lake level. Even if the lake level was drastically lower than today, the depths of upper zones should remain the same, while the bottom water might be the only part that became 'shallower'. In addition, the authors argue that '... low BIT index values reflect periods of enhanced mixing and a depressed oxycline due to dry and windy conditions when the ITCZ is further afield.' (SI Lines 457-458); while this seems to have some inconsistency with the Lake Chala SPM results (Baxter et al., 2021). In Figure 6F in Baxter et al. (2021), the BIT indexes tend to be lower during the upper water-column stratification. Collectively, I believe this manuscript would benefit from a better clarification of interpreting the BIT index.

In the manuscript, the authors imply that the inverse relationship happened when the pCO₂ reached a tipping point. I feel that this part needs a more thorough discussion. Some of the hypotheses were provided on Lines 226-244 but none of them seems related to pCO₂.

Minor comments:

Lines 53-55 This sentence seems irrelevant at this spot and interrupts the coherence.

SI Lines 352-354 Raberg et al. (2021) should not be cited here. It seems true that the 6-methyl within the Meth set is somewhat correlated to MAF (Fig. 7 in Raberg et al. (2021)); however, the methylation of 6-me brGDGTs shows very poor correlations (Fig. 3 in Raberg et al. (2021)). It is likely because all those 6-methyl brGDGTs show negative correlations with temperature (Fig. 7 in Raberg et al. (2021)). While for the 5-methyl brGDGTs, the brGDGT-Ia is always correlated to temperature positively while the IIIa has a negative correlation. It also should be noted that the 6-methyl has moderate correlations with a variety of environmental factors, based on the Martínez-Sosa et al. (2021), such as pH, precipitation, and conductivity.

Fig 1. The topography does not really provide useful information here. Also, please double-check the elevation data. I think the Tibet plateau shouldn't be that tall.

References:

- Baxter, A.J., van Bree, L.G.J., Peterse, F., Hopmans, E.C., Villanueva, L., Verschuren, D., Sinninghe Damsté, J.S., 2021. Seasonal and multi-annual variation in the abundance of isoprenoid GDGT membrane lipids and their producers in the water column of a meromictic equatorial crater lake (Lake Chala, East Africa). *Quaternary Science Reviews* 273, 107263.
- De Jonge, C., Hopmans, E.C., Zell, C.I., Kim, J.H., Schouten, S., Sinninghe Damsté, J.S., 2014. Occurrence and abundance of 6-methyl branched glycerol dialkyl glycerol tetraethers in soils: Implications for palaeoclimate reconstruction. *Geochimica et Cosmochimica Acta* 141, 97–112.
- Jackson, M.S., Kelly, M.A., Russell, J.M., Doughty, A.M., Howley, J.A., Chipman, J.W., Cavagnaro, D., Nakileza, B., Zimmerman, S.R.H., 2019. High-latitude warming initiated the onset of the last deglaciation in the tropics. *Science Advances* 5, 1–9.
- Loomis, S.E., Russell, J.M., Ladd, B., Street-Perrott, F.A., Sinninghe Damsté, J.S., 2012. Calibration and application of the branched GDGT temperature proxy on East African lake sediments. *Earth and Planetary Science Letters* 357–358, 277–288.
- Martínez-Sosa, P., Tierney, J.E., Stefanescu, I.C., Dearing Crampton-Flood, E., Shuman, B.N., Routson, C., 2021. A global Bayesian temperature calibration for lacustrine brGDGTs. *Geochimica et Cosmochimica Acta* 305, 87–105.
- Pearson, E.J., Juggins, S., Talbot, H.M., Weckström, J., Rosén, P., Ryves, D.B., Roberts, S.J., Schmidt, R., 2011. A lacustrine GDGT-temperature calibration from the Scandinavian Arctic to Antarctic: Renewed potential for the application of GDGT-paleothermometry in lakes. *Geochimica et Cosmochimica Acta* 75, 6225–6238.
- Raberg, J.H., Harning, D.J., Crump, S.E., de Wet, G., Blumm, A., Kopf, S., Geirsdóttir, Á., Miller, G.H., Sepúlveda, J., 2021. Revised fractional abundances and warm-season temperatures substantially improve brGDGT calibrations in lake sediments. *Biogeosciences* 18, 3579–3603.

Russell, J.M., Hopmans, E.C., Loomis, S.E., Liang, J., Sinninghe Damsté, J.S., 2018. Distributions of 5- and 6-methyl branched glycerol dialkyl glycerol tetraethers (brGDGTs) in East African lake sediment: Effects of temperature, pH, and new lacustrine paleotemperature calibrations. *Organic Geochemistry* 117, 56–69.

Sinninghe Damsté, J.S., Ossebaar, J., Schouten, S., Verschuren, D., 2012. Distribution of tetraether lipids in the 25-ka sedimentary record of Lake Challa: Extracting reliable TEX 86 and MBT/CBT palaeotemperatures from an equatorial African lake. *Quaternary Science Reviews* 50, 43–54.

van Bree, L.G.J., Peterse, F., Baxter, A.J., De Crop, W., van Grinsven, S., Villanueva, L., Verschuren, D., Sinninghe Damsté, J.S., 2020. Seasonal variability and sources of in situ brGDGT production in a permanently stratified African crater lake. *Biogeosciences* 17, 5443–5463.

Weber, Y., Sinninghe Damsté, J.S., Zopfi, J., De Jonge, C., Gilli, A., Schubert, C.J., Lepori, F., Lehmann, M.F., Niemann, H., 2018. Redox-dependent niche differentiation provides evidence for multiple bacterial sources of glycerol tetraether lipids in lakes. *Proceedings of the National Academy of Sciences* 115, 10926–10931.

Referee #3 (Remarks to the Author):

Baxter et al., present a new GDGT biomarker record, interpreted as mean summer temperature and moisture balance, from Lake Challa in eastern Africa over the last 75 kyr. The authors find that prior to the Holocene, temperature and moisture balance were correlated, placing importance on certain mechanisms within the climate system. Then, coinciding with the deglacial increase in global CO₂, the two parameters become anti-correlated. This suggests that the increasing global temperatures due to anthropogenic climate change will drive further aridification in the region, contradicting climate models that predict more extreme rainfall. The new reconstructions are of unprecedented temporal resolution for GDGT biomarkers, and the age model is fantastic in the top third of the record, where much of the conclusions are based. One thing I really like about this paper is how well-motivated the study is. Quantifying the responses of this region to future warming is of utmost importance, and I think there could be more at the back end of the manuscript circling back to this original motivation. However, there are some major concerns relating to some supporting evidence of the conclusions. Please find my major, minor, and technical comments and corrections below.

Major Comments:

1. Literature

I want to first off mention that I firmly disagree with the concept of journal-imposed reference limits, as I think that this inhibits the documentation of thorough literature review and support of claims. That being said, there are statements within the manuscript such as beginning on line 61 (“only few well-resolved climate records ...”), that I feel go beyond the simple omission of citations,

and need to be revised. There has been a whole body of work in eastern Africa (e.g., Foerster et al. 2018) that falls into these categories and should be cited and explored further.

2. Frequency analysis

With a 75 kyr-long record, the spectral and wavelet analyses and their respective conclusions need to be revised. It is clear in Figure ED 2 that the cone of influence does not even touch the 41 kyr cycle. That is because you need at least 3 cycles, preferably more, within the length of a record to document any significance. It isn't that the 41 kyr cycle is not there, it is that this particular record (although with wonderful resolution!) is simply not long enough to comment on obliquity forcing. I recommend any mention of cycles above precession be removed from the text. I also recommend updating the figures to not even show these cycles that aren't resolvable, and perhaps the color scale will change in a way that highlights the half-precession cycle or something else better.

3. Mechanisms

The mechanisms that are discussed as drivers of the variability and relationships between moisture balance and MST are thorough, yet I feel like 1, the data are over-interpreted and 2, the way that this section is written is too matter-of-fact. Starting with over-interpretation, there appear to be multiple steps between a highly-resolved coupled summer temperature / BIT index record and major global climate system features. Does the increase of CO₂ drive the change in correlation at 11 ka? Without forcing experiments and a fuller study, I warn against mistaking correlation as causation. I do think it is good to discuss all of the potential processes, which are there, but the language needs to be toned down to indicate that these are possibilities based on the literature and what this new dataset supports, rather than a firm statement like in lines 169-177 (which needs to be broken up in multiple sentences) suggesting that the new record is definitive proof of a fairly convoluted series of steps. The new data does not confirm that Indian Ocean SST was unusually low (lines 198-199), but it could be consistent with that finding. Some reworking of the text in the latter half of the paper on the discussion of possible forcing mechanisms needs to be done.

Further, the study is motivated (very well!) by the issue that CO₂ is rising and we are unsure of how EA will respond. This is mentioned again at the end, and the authors highlight the CO₂ threshold, but then temperature itself is invoked as the driver of the shift in correlation. This is a bit confusing, and goes back to my original point that perhaps a regional record of BIT index and GDGT summer temp over one g-ig cycle with no modeling efforts is not robust enough to make these big, impactful conclusions. There is also at times some circular logic, like lines 204-206, that needs to be explained in more detail.

Two more moderate issues are the 1, the BIT index values of 1 and 0 should be removed from any sort of correlation analysis, and 2, summer temp and moisture balance are not annual temp and rainfall, and so more discussion into how powerful these are for interpreting the climate cycle should be included.

Minor Comments:

- Please refer to the study region as eastern Africa, as East Africa is a political designation and eastern Africa is a geographic one.
- I've never heard it be called the east African climate paradox, and this isn't mentioned anywhere

else in the paper, so I think it's fine to remove

- More citations for climate dynamics intro line 69-74
- Please be more up front about the age model, which does have many constraints, but they are 99% concentrated in the top 25 kyr of the record. Make this clear.
- Explain use of moisture balance – P-E? how does this relate to BIT? Lines 95-96.
- Calling temp interpretation MST but moisture-balance BIT-index is perhaps misleading (One is the climate parameter and one is the index). BIT index has been interpreted other ways, so maybe abbreviate to MB?
- Line 116: what climate proxy? Be more specific and perhaps use a different term than “climate-proxy” when meaning climate reconstruction or proxy records?
- If GDGT temp is defined and supported as MST, continue calling it MST and not temperature
- Line 151 and beyond: drought refers to shorter events with certain impacts – perhaps change the language to extreme aridity
- Line 165: it's tricky to compare proxies for different things from different regions – tone down these conclusions here about forest contraction with a BIT-index record
- Line 182: I don't think it's necessary to recall this as a mystery interval
- Line 241-244: this kind of background info on drought should be introduced earlier
- Line 249-250: this is an important point that needs to be expanded on more and earlier – that this is how it is today, but paleoclimate records x, y, and z show otherwise going further back in time

Technical corrections:

- Line 30: which allows... us? for the?
- Last line of abstract: make it clear this is about future climate change
- Line 67: change to depositionally-continuous and permanently-stratified
- Line 75: varve refers to annual no matter the depth scale
- Line 76: not sure what is meant by quiet
- Line 82-83: grammar seems off here, perhaps rephrase
- Line 90: change to globally-distributed
- Line 104: change to can be assumed to be stable...
- Line 149: move the word “often”, perhaps before prevailed
- Line 153: remove “is known to have”
- Line 155: change to BIT-index (although other times it's referred to as BIT index – be consistent), but again, perhaps it would be best to call this moisture balance or change MST to brGDGT calibration
- Line 248: conducive of

Figures:

- Fig 1: the colors and arrows everywhere is not very effective at introducing the study location. I don't think it's necessary to include the Hulu-Dongge cave site. Make sure it is color-blind appropriate.
- Fig 2: it's very difficult to see the error region of MST – and is it just a simple 2 degrees? Please use error propagation from the respective calibration
- Fig extended 2: I commented on this above – please redo these figures to eliminate periodicities that are not resolved by the dataset. Add (kyr) to the Period labels.

B. Author response to referee comments

Referee #1

5. Baxter et al. provide important geological time-scale context for ongoing debates on the causative mechanisms for modern and future changes in temperature and effective moisture in eastern Africa. They use sedimentary records from scientific drill cores obtained from a unique environmental and depositional setting, Lake Chala, adjacent to Mount Kilimanjaro and on the border of Kenya and Tanzania. Such sampling projects present extraordinary logistical and funding challenges, and require an expansive network of collaborators in different fields to achieve successful outcomes. The Chala site is particularly valuable because its restricted watershed and high aspect-ratio morphology faithfully record environmental conditions over geologically meaningful time periods, which cross multiple glacial-interglacial intervals. Moreover, its position east of the Congo Air Boundary means Lake Chala is influenced principally by regional Indian Ocean climatological conditions, simplifying the palaeoclimatological record and allowing robust comparisons to climate modeling efforts.

The scientific achievements of the Chala team are exceptional and the community has long looked forward to seeing its important outcomes. The work presented herein is part of the forthcoming story and is a thoughtful and rigorous analysis.

R5. We thank referee #1 for the positive assessment of this work, and for recognition of the cumulative achievement that it represents, as indeed our paleoenvironmental research at Lake Chala, including rigorous study of the modern lake system and the surrounding landscape for calibration and validation of sedimentary climate proxies, has by now spanned almost two decades.

6. With the current manuscript I have questions and concerns about the age model as presented, the rather limited proxy analyses included herein, and aspects of the synthesis.

R6a. In view of several detailed comments by referee #1 about the age model and 'limited' proxy analysis that our study represents, we would like to start this rebuttal by pointing out critical differences between the principal aim and research strategy of the DeepCHALLA project (18; <https://www.icdp-online.org/projects/world/africa/lake-challa/>), of which we present a first set of comprehensive results here, and other large geoscience projects which for the purpose of climate and/or paleoenvironmental reconstruction have drilled the sedimentary records of extant lakes, dry lake beds or buried ancient lake deposits.

R6b. The principal goal of most other large continental drilling projects to date, many of them made possible through co-financing by the International Continental Scientific Drilling Program (ICDP; <https://www.icdp-online.org/projects/world>), has been to target and recover geological archives of a considerably longer temporal reach than is the case for DeepCHALLA. Those projects typically cover a sizable number of astronomical cycles (cyclic variation in Earth's orbit around the Sun, which affects the seasonal and latitudinal distribution of solar insolation), firstly because it is of logical research interest to reconstruct regional climate and ecosystem dynamics in response to this external climate forcing, and secondly because the often strong signatures of that response can provide highly valuable chronological constraints to complement the results of absolute dating, especially in the common situation of those sedimentary records being discontinuous. For continental drilling projects situated in eastern Africa (e.g., Lake Malawi, HSPDP including Chew Bahir, Olorgesailie), an additional reason to aim for the longest possible records is their role as paleoenvironmental reference frame for the origin and evolution of *Homo sapiens* and its hominin ancestors. In this research context, a drilled sediment record is considered continuous when all astronomical cycles it covers are well-represented, and ideally also sub-Milankovitch (i.e., millennial-scale) climate variability can be discerned. The general reasoning is that hiatuses longer than a few millennia, if not constrained by absolute dating, can always be recognized as distinct unconformities in lithostratigraphy. In any case, the 'high-resolution' multi-channel seismic profiling that is often employed to assess the temporal continuity of targeted (ancient) lake beds before drilling have a best-possible vertical resolution on the order of 1-

2 meter, thus typically representing several millennia of lacustrine sedimentation. The unavoidable trade-off between the temporal reach and depositional continuity of these 'deep-time' archives complicates direct comparison of the paleoenvironmental data they produce with the important reference records of global climate history, as the latter either derive from demonstrably continuous climate archives (e.g., the polar ice-core records of past greenhouse gas concentrations and high-latitude temperature; deep-sea marine-sediment records of sea-surface temperature and total continental ice volume), or are so well dated that they can be assembled into a single continuous archive (the cave speleothem records of tropical monsoon intensity). But typically this issue does not undermine those projects' main research goals, because their main focus is either on the long-term climate trends, on how long-term periodicities in the data reveal the dominant process of regional climate forcing, or on contrasting climate conditions during different time 'windows' in the past.

R6c. In contrast, the principal objective of the ICDP DeepCHALLA project is to produce climate and paleoenvironmental reconstructions from near the equator (as opposed to the northern or southern hemisphere tropics) with demonstrated continuity at the shortest-possible time scale, so that the full range of (temporally nested) climate variability and forcing mechanisms can be investigated: from the seasonal and inter-annual (e.g., ENSO) over century-scale (e.g., solar radiation output) and millennial to orbital scale, and this ideally over at least one complete glacial-interglacial cycle (i.e., the last ~130,000 years, comparable to the Greenland ice-core record). So in terms of temporal reach, DeepCHALLA is decidedly a modest ICDP project. However, demonstrating temporal continuity in lake-sediment records is a tall order, so much so that paleolimnologists have been stressing already for 30 years (e.g., Larsen & McDonald 1993; Dearing 1999; Verschuren 1999a) the importance of considering various factors of catchment size and hydrology, lake morphometry and water-column mixing regime before selecting individual lake sites for paleoecological (and by extension, paleoclimate) studies with high temporal resolution. The high aspect ratio (i.e., high depth to diameter ratio) of Lake Chala referred to by referee #1 is a major bonus, because it reduces the probability of sediment resuspension due to wave-induced bottom turbulence (Håkanson & Jansson 1983; Hilton 1985; Fig. R1), and thus the risk of cryptic hiatuses in sediment accumulation during past low-stands (Verschuren 1999a). Moreover, rare events of extreme wind stress may generate wave-induced turbulence that disturbs or interrupts the accumulation of fine-grained sediments at water depths significantly greater than the so-called mud deposition boundary depth (Rowan et al. 1992) predicted by more typical maximum annual wind speeds (Dearing 1999). Indeed, even at 68 m water depth in the middle of Lake Victoria, a ~25-year cryptic hiatus occurs in seemingly undisturbed sediments dated to the early 20th century (Verschuren et al. 1998). Continuous sediment accumulation can only be guaranteed in lakes with aspect ratio high enough to render it unlikely that wave-induced turbulence ever reached the lake bottom throughout the period of interest (Larsen & McDonald 1993); and a sufficiently large area of undisturbed fine-grained sediment accumulation so that mass-wasting events in steeply sloping bottom areas along the basin periphery are no longer erosive when reaching the drill site (cf. 19,33).

R6d. Given the ambitious objectives of the DeepCHALLA project as presented to ICDP, the promise of continuous sediment accumulation in Lake Chala over long time scales was credible for four reasons:

- i) the lake's high aspect ratio (relative depth $Z_r = 3.9\%$; compare with data in ref. 126);
- ii) very-high-resolution seismic-reflection profiling (3.5 kHz, with vertical resolution of 10-15 cm) had already demonstrated absence of temporal hiatuses longer than a few centuries in the sediment record back to ~130,000 years ago (33);
- iii) the last 25,000 years of the sediment record as recovered by the Chalacea project consists entirely of laminated and fine-grained (clay to fine silt) lake muds with high organic content, implying a depositional setting with (near-)permanent low turbulence and anoxic conditions (20,75); and
- iv) reconstruction of past lake-level fluctuations based on seismic-reflection stratigraphy had indicated that the amplitude of changes in water-column depth prior to 25,000 years ago was not significantly greater than that within the last 25,000 years (33).

And indeed we find that the 62.96-m long section of the DeepCHALLA record presented in this study (68.39 m including turbidites) consists entirely of fine-grained organic lake muds (line 89 and Extended Data Fig. 1), implying both continuity of sediment accumulation and temporal uniformity of any taphonomical processes that may in some way affect the incorporation of climate proxies in the bottom sediments. In other words, when compared to the more variable lithologies (and thus more eventful depositional history) of lake records from other African ICDP sites (e.g. Chew Bahir, cf. **R41**), the 75-kyr portion of Lake Chala sediments analysed in this study can be treated as a single lithostratigraphic unit. The three facies that we do distinguish (Extended Data Fig. 1) only differ from each other in the scale of lamination: mm-scale (varved-like), cm-scale (banded) or a rapidly alternating mixture of the two.

R6e. This combination of demonstrated depositional continuity and taphonomical constancy is rare among long tropical lake records drilled to date (only lakes Malawi and Bosumtwi are comparable, at least with regard to their record of the last 75 ka); and in our view this depositional continuity and taphonomical constancy has been instrumental in our successful production of climate-proxy time series that are, from a climate-dynamical perspective, fully consistent with the world's major climate reference records mentioned above (Fig. 2).

Fig. R1. Principal mechanisms of sediment focusing and redistribution in lakes, in relation to lake surface area, maximum depth and local bottom slope (modified from Hilton 1985), with superimposed the range of maximum depth and surface area realised at Lake Chala between high-stand (~modern-day) and extreme past low-stand conditions as inferred from seismic stratigraphy (19). Lakes in which peripheral wave activity and/or intermittent complete water-column mixing are the dominant mechanism of sediment redistribution have well-defined mud deposition boundary depths (Rowan et al. 1992) and are hence best suited for preserving consistent sedimentary signatures of past changes in lake water balance (Dearing 1999). At long time scales, most climate-sensitive lakes go through low-stand phases when random sediment redistribution is the dominant mechanism, creating (cryptic) hiatuses in their sediment record (or phases of complete desiccation, evidently also creating hiatuses in sediment accumulation). Due to its strongly concave crater-basin morphology (Supplementary Fig. 1), Lake Chala has maintained a large central area of flat lake bottom throughout the past 75 kyr, such that slope instabilities were always restricted to the steeper-sloping basin periphery (33) and turbiditic flows from past mass-wasting events (92) were no longer erosive by the time they reached the offshore DeepCHALLA drill site.

7a. The age model presented relies on an exquisite record of 162 ^{14}C ages over the last 25000 years, shallow ^{210}Pb ages, and the presence of the Younger Toba Tuff (73.7 ka). Notably, the ^{14}C ages

were transferred into the DeepChalla project records from an earlier study (Challacea project); accordingly there is a gap in absolute ages beyond 25,000 years. The authors argue that there is extreme uniformity in sedimentation rates over the 75 kyr records presented herein, and that additional ages are not necessary.

R7a. Correct; we refer to our detailed argumentation on lines S42-56 and S157-208, summarised in **R8** below.

7b. *The presence of the Youngest Toba Tuff is presented as confirmed in the Chala record, and the authors refer to shard density and geochemical fingerprinting results as definitive. However the actual data behind this postulation are not represented, even in the Supplemental, and work characterizing the YTT and other tephra analyses are cited as 'in prep'. Given the vast distances of airborne transport to eastern Africa for this singular event, the potential for confusion with other east African tephras, and its importance for this study, primary data on the YTT and/or definitive citations are seemingly required here.*

R7b. This is a fair comment, which we have accommodated by now providing in the Methods, Supplementary Methods and Supplementary Tables all data and analysis related to identification of the 73.7-ka YTT cryptotephra near the base of our 75-kyr DeepCHALLA record, and visualised these results in the new Extended Data Figs 2-3. Further we now include Dr. Catherine Martin-Jones and Dr. Christine Lane (University of Cambridge, UK), who generated these data, as co-authors of this study.

8. *The authors argue that additional age dating in the interval 25-73 ka is not required, yet it is incongruous that they would present such a detailed and exceptional stratigraphy for the last 25,000 years from their previous work, and not include new observations from these newer drill cores, which were acquired with extraordinary effort. The authors proceed to interpret their proxies (see below) with an exceptional level of confidence and granularity. They may be correct about the veracity of the deeper age model, but given the extent of the interpretative effort, further verification of the age-depth relations between 73 ka and 25 ka is warranted.*

R8a. First, we would like to point out that similar concern about the age model is not expressed by referees #2 and #3, indicating that they accept our argumentation about the constancy of sediment accumulation (SAR) between 73.7 ka and 25 ka, such that additional absolute dating would not significantly alter the ages presently inferred for data intervals between those anchoring points. Below we summarise this argument in five steps:

1) In the section between 12.4 and 1.5 meter event-free depth (mefd), which consists largely of varve-like sediments (Extended Data Fig 1b: core segment panels 1-3), SAR is demonstrated by 96 ¹⁴C dates (corrected for old-carbon age offset; 72) to be highly stable (Extended Data Fig. 1a) with a long-term mean value of 0.812 ± 0.013 m/kyr between 14.5 ka and 1.0 ka (we do not consider the uppermost ~1.5-m thick section of uncompacted sediments; lines S144-145).

2) In the section between 18.4 and 12.4 mefd, which consists mostly of cm-scale banded sediments (Extended data Fig. 1b: core segment 4) or alternating mm-scale lamination and cm-scale banding (Extended data Fig. 1b: core segment 5), SAR is demonstrated by 17 ¹⁴C dates to be similarly stable (Extended Data Fig. 1a) with a long-term mean of 1.006 ± 0.067 m/kyr between 20.5 ka and 14.5 ka.

3) The latter section was deposited during the most extreme low-stand of Lake Chala in the past 75 kyr (Extended Data Fig. 1a: seismic stratigraphy column, from 19) when sediment focusing restricted undisturbed deposition of soft sediments to a smaller central area of lake bottom (33). Considering that this most 'extreme' focusing event of the last 75 kyr increased SAR by only 24%, evidently SAR must have varied much less than that in the section dated to between 73.7 and 20.5 ka, given that it consists almost entirely of varve-like sediments (Extended data Fig. 1b: core segments 6-10).

4) The actual long-term mean SAR in the 73.7-20.5 ka core section is 0.825 m/kyr, i.e. near-identical to that in the 14.5-1.0 ka section, consistent with the highly similar appearance of the predominantly varve-like sediments in both sections (Extended data Fig. 1b: core segments 6-10 versus 1-3).

5) Because the 73.7ka YTT horizon which anchors the base of the studied 75-kyr sequence is not a visible tephra layer but a crypto-tephra, its characteristic glass shards (Extended Data Fig. 2) had to be discovered using labour-intensive microscopic analyses (Supplementary Methods, lines S95-103). Based on linear interpolation between seismic signatures of past low-stands (33) implying a long-term mean SAR of ~0.8 m/kyr throughout the DeepCHALLA sequence, the 1.78-m (~2000-year) long core section between 60.75 and 62.53 mefd was selected for a 'first attempt' to discover the YTT cryptotephra (lines S92-95), and that is where it was found (Extended data Figs. 1-2). Evidently this immediate success would not have been achieved if SAR had varied significantly, or if its mean value had been significantly different from 0.8 m/kyr. For comparison, based on the then-available absolute age model for the ICDP-drilled sediment record from Lake Malawi (based on ¹⁴C dating and paleomagnetic intensity signatures; Scholz et al. 2007), in that record a 20-m long core section (covering ~40,000 years) was scanned microscopically in search of the YTT cryptotephra, and its eventual discovery at 28.08-28.10 m depth (in core MAL05-1C) required adjustment of the absolute age model upward by ~14,000 years at that level (80).

R8b. In summary, we believe that our demonstration of depositional continuity (**R6d**) and highly uniform lithology of the upper 62.96 mefd of the DeepCHALLA sequence (Extended data Fig. 1) provides a strong lithostratigraphic constraint on the stability and rate of sediment accumulation in the pre-25 ka section of our climate-proxy record, a belief strengthened by our recovery of the 73.7-kyr YTT cryptotephra, which anchors our 75-kyr record at its base, very near the depth where it was predicted to occur based on that lithostratigraphic constraint. Moreover, the main result presented in our paper, namely that the relationship between effective moisture and temperature in eastern equatorial Africa switched ~11,700 years ago, i.e. between the glacial and interglacial sections of our paired 75-kyr climate-proxy time series, actually does not depend on the exact timing of each paired data point in the glacial section of the record. Thus, even if future addition of absolute age markers between 73.7 ka and 25 ka were to result in minor age-model adjustments, it would not affect our main conclusion based on the data presented here.

9. *In the published work of Martin-Jones (2020) there is reference to several other tephras identified in the Chala cores over this depth range. Could any of these other tephras represent regional (East African) eruptive events as time markers? Is there not the possibility of incorporating tephra stratigraphy in the age model?*

R9. Martin-Jones et al. (31) present local/regional volcanic eruptions recorded as visible tephra layers in the Chalacea and DeepCHALLA sequences, using an age model for the DeepCHALLA sequence anchored in the seismic-stratigraphy interpretation of ref. 33 and of which the 0-75 ka portion is near-identical to the age model presented here. The aim of that study is to place those regional eruptions (which are mostly undated, or indirectly dated at proximal sites with often large age uncertainty) in a single chronological framework, as provided by the continuous and high-resolution Lake Chala record, not vice versa. Conversely, of the few East African eruptions that have been well-dated elsewhere, such as the 36-ka Menengai Tuff (99) found widely in the Lake Victoria region, none have so far been discovered in the sediment sequence of Lake Chala.

10. *Given the intensity of the analytical work on the shorter cores from previous projects, it is surprising that there are no additional radiocarbon results from parts of the core that are older than 25 ka. Are there any other absolute ages possible or available between 25 ka and YTT? For instance with the marked advances in luminescence dating in recent years are there any results to include here? Is it possible to employ varve counting to further constrain the age model?*

R10a. With respect, referee #1 may have overlooked our mention (lines S51-56) of the varve counting that we performed on the Chalacea-project sequence from Lake Chala, namely one ~2510-year

section from the glacial period (21-18.5 ka; in the DeepCHALLA sequence recovered from slightly lower water depth than the mid-lake Challacea location, part of the equivalent section has alternating mm- and cm-scale lamination) and the last ~2609 years of the Holocene (74). As argued above, there is no reason whatsoever to suspect that the varve-like lamination of the 25-73.7 ka section has a rate of accumulation that is either significantly fluctuating and/or significantly different from that during the Holocene (Extended data Fig. 1b: core segments 6-8 and 10 versus 1-3).

R10b. With respect to the varve counting, we also note that, due to more subdued seasonality in the deposition of successive laminae composing one varve year at this tropical site compared to varved lake sequences from north-temperate lakes, varve counting must be done through microscopic visual scanning of resin-impregnated and thin-sectioned sediment blocks (74; lines S54-56). Counting the two above-mentioned multi-millennial varved sections in the Challacea record was a multi-year effort, and to our best knowledge, no one has ever counted ~53,000 varves in a single lake record using this method. Of course, we do aim to eventually count this entire section (as well as varve-like sections in the >75-ka portion of the DeepCHALLA record), but this will require innovative combination of microscopic thin-section analysis with μ XRF scanning (which is no trivial enterprise on a total ~150 m of varve-like sediments), perhaps aided by hyperspectral analysis of sediment colour (but which will need to be validated before application to these varves composed of diatomaceous organic mud).

R10c. As regards the possible application of other dating techniques we refer to **R11b**.

11. *My sense is that there must be a more robust age model forthcoming that may better justify the highly granular level of thoughtful interpretations and climate forcings considered in this paper.*

R11a. With respect, our impression is that referee #1 undervalues the chronological constraint provided by the lithostratigraphy (cf. **R8**), in particular the limited facies variation (mm-scale or cm-scale lamination, but otherwise identical sediment composition) combined with our calibration of the mean sedimentation rates of both facies (Supplementary Fig. 1 and lines S141-147), which correctly predicted at what depth the 73.7-ka YTT cryptotephra would be discovered.

R11b. Evidently, we are pursuing diverse absolute dating approaches on the DeepCHALLA sequence, including ^{14}C , Ar/Ar and paleomagnetic dating. However, the reality is that all these techniques (as well as thermoluminescence dating) have their limitations, which become evident when the results are compared directly (as experienced in other long lake records: Shanahan et al. 2013; Valero-Garcés et al. 2019; Roberts et al. 2021) and confronted with the lithostratigraphic constraints and the YTT age marker presented here. Consequently, a final integrated age model for the DeepCHALLA sequence, with appropriate weighing of the different types of absolute dating data, is not achievable in the near future. However, we are confident that the last 75-kyr section of that final age model will be very close to the age model presented here.

12. *The authors analyze lipids from prokaryotes to assess past temperatures and hydroclimates in the Lake Chala sediment cores (specifically the distribution of the GDGT's). I will let other experts evaluate their analyses of branched GDGTs for summer mean temperatures. The authors utilize the so-called BIT index as a measure of effective moisture. When initially applied to paleoclimate studies, the BIT index was characteristically used to assess the relative quantity of soil-derived Archaea washed into lake basins, and thus was used to validate other biomarker methods (e.g. TEX86) that were employed to assess lake surface paleotemperatures. Here in Chala however, the authors assert that the BIT index is a direct moisture indicator, as a consequence of several ancillary studies of the water column and sediments deposited over the past 250 years. The authors may indeed be spot-on here, however it is easy to conceptualize other scenarios from deep-time when this proxy may not have been produced under identical circumstances.*

R12a. Indeed the BIT index of branched and isoprenoid GDGTs was initially developed (Hopmans et al. 2004; our ref. 25) as a proxy for terrestrial input in marine and lacustrine sediments, and this is also how we initially applied it in the sediment record of Lake Chala (20), surmising that the influx of

terrestrial soil material was proportional to the amount of precipitation over the crater catchment. However, since the discovery of branched GDGT production in the water column of Lake Chala²⁶ and elsewhere (e.g., 119), our understanding of the BIT index as hydroclimate proxy in Lake Chala has evolved as our studies of the modern lake system produced increasingly more detailed and extensive data. Rather than ‘ancillary studies’, our current understanding of what the BIT index represents, specifically at Lake Chala (28), is based on ~15 years of targeted research including analyses of soils from within and outside the crater catchment, recently deposited sediments at >20 locations covering all areas of the lake, suspended particulate matter (SPM) at up to 13 depth intervals over 17 consecutive months, a sediment trap series covering 53 consecutive months, and validation of a BIT-index-based hydroclimate reconstruction against historical documentary data of regional drought episodes within the last ~200 years (27-29,91). With regard to source attribution and temporal variation of lipid biomarker proxies, this little lake on the border of Kenya and Tanzania is among the best studied lake systems worldwide, and indeed the only African lake where such studies have been conducted. Thus, it seems fair to state that while paleoclimatic applications of lipid biomarker proxies at other African sites rely on plausible but locally untested ‘scenarios’ (even for reconstructions in the not-so-distant past), our interpretation of Lake Chala sedimentary data are grounded in the actual mechanisms by which short- and longer-term regional climate variability influences lipid biomarker distributions. At the same time we are the first to advise caution in using the BIT index as proxy for past hydroclimate variation at other sites, unless GDGT distributions in the sediments deposited there is properly understood, based on studies of the local modern system and short sediment records such as we have conducted on Lake Chala (lines S555-558).

R12b. We fully agree with referee #1 that there may have been environmental settings in the distant past (‘deep time’) when the relationship between the habitat of GDGT-producing microbiota in Lake Chala and regional hydroclimate may have been different than currently understood. This is the exact reason why the present study is limited to the last 75 kyr, the period during which we are confident that 1) the basic relationship between the Chala BIT-index proxy and effective moisture has been the same as elucidated in the modern lake system; and 2) the depositional environment of Lake Chala has been sufficiently stable to validly assume that the incorporation and preservation of that relationship into the sediment record has been constant through time (lines S555-561). Thus, we consciously excluded situations where alternative ‘deep-time’ scenarios might need to be considered.

13. *Moreover, the BIT values presented in this record are clipped at a maximum for extended periods, thus clearly limiting its sensitivity to evaluate maximally wet environments.*

R13. As described in the Supplementary Discussion (lines S540-549), we agree that the Chala BIT index has reduced sensitivity to very wet climate conditions, as translated in a water column lacking a niche suitable for crenarchaeol producers and thus leading to a maximum BIT-index values approaching 1. However, in all but three analysed samples crenarchaeol was still above detection limit, meaning that the BIT index is equal to 1 in less than 1% of all sediment horizons. Regardless, using a ratio (e.g., $\Sigma\text{brGDGTs}/\text{crenarchaeol}$) instead of an index ($\text{BIT index} = \Sigma\text{brGDGTs}/(\text{crenarchaeol} + \text{brGDGTs})$), and thus allowing unconstrained outcomes, yields similar results. Secondly, this reduced sensitivity under very wet climate conditions does not compromise the proxy value of the Chala BIT index more than that palynology would be compromised because all lowland regions in Africa with annual rainfall exceeding ~1400 mm and lacking a distinct dry season are characterized by closed-canopy rainforest vegetation (while some of those regions receive >5000 mm annually). What we deem more important is that the Chala BIT index is sensitive both to relatively modest hydroclimate variation within the historical period (29; Supplementary Fig. 7) and to the full range of documented African hydroclimate variation between some of the driest (the Heinrich 1 Stadial: 42) and wettest (the early Holocene: 43-44 and Bergner et al. 2003) episodes of the late Quaternary (Fig. 2d). Under the very wet conditions which appear to have prevailed in eastern equatorial Africa between 52 ka and 42 ka (relative to the region’s long-term mean climate regime; not necessarily extremely wet in absolute terms) the BIT

index becomes a categorical rather than a quantitative (cf. **R54**) hydroclimate proxy, but it remains trustworthy in its Lake Chala application.

14. *It is surprising that the authors have not considered employing a multiproxy approach to assessing past hydroclimate conditions in Chala. For instance there are numerous other measurements that must be available as part of their broad suite of analyses that could contribute to quantifying past hydroclimate, and perhaps be integrated using a basic PCA analysis. For instance turbidite frequency or data derived from scanning XRF measurements (Ti and other elements) are commonly employed to indicate terrigenous inputs into lacustrine and marine environments. Accordingly I am baffled why such important work is not utilizing other proxy techniques to better validate the environmental forcings.*

R14a. Referring to our introductory statement (**R6**), using a multi-proxy approach as a way to validate climate reconstructions is certainly advisable at sites where proxy validation in the modern system has not, or cannot, be carried out (such as is the case for paleolakes or transient lake basins). In such geologically-oriented research strategy, which is best adapted to climate reconstruction in the distant past with little guidance from the local modern system, it is assumed that while no single proxy can be considered robust or free of ambiguity, the climate-change signature common to all applied proxies (and resolvable using multivariate statistical methods such as PCA) must have a high probability of reflecting past reality. However, the unique qualities of the Lake Chala sediment record allowed us to pursue a research strategy more akin to that applied on climate archives that are demonstrably continuous in time and with uniform proxy-climate relationship throughout the period of interest starting from their present-day setting, such as δD or $\delta^{18}O$ variation in polar ice-core records as temperature proxy, or $\delta^{18}O$ in the calcite of tropical speleothems as proxy of monsoon intensity. And indeed, publication of paleoclimate records based on ice-core or speleothem data does not typically require support from other local temperature or hydroclimate proxies. Moreover, our published climate reconstructions for the last 25 kyr have shown that different environmental proxies contained in the Chala sediment record reflect different aspects of past hydroclimate regimes. For example, $\delta^{18}O$ variation in fossil diatoms reflects the intensity of seasonal drought rather than annual rainfall or annually integrated moisture balance (Barker et al. 2011). For optimal exploitation of the Lake Chala climate archive, our overarching research strategy is to validate each hydroclimate proxy in its own right, so that our reconstructions can discriminate between different aspects of past hydroclimate. If successful, this will create added value for elucidation of the climatic drivers causing variation in those individual proxies. Evidently this discriminating power would be lost if those proxies were assumed to reflect 'more or less' the same climate variable and lumped together in a multi-proxy approach.

R14b. Secondly, we want to stress again that the Chala BIT-index proxy has been mechanistically unravelled in the modern system (summarized in 28), as well as historically validated (29): three intervals of low BIT-index values within the last 200 years align convincingly with three historically documented episodes of severe regional drought (panel a in the new Supplementary Fig. 7 reproduced below). To our knowledge, no other lipid-biomarker-based hydroclimate proxy from any published site worldwide can claim this level of performance; and no other lake-based hydroclimate proxy (such as turbidite frequency or XRF-based elemental proxies of terrigenous input) from any published African site is as well constrained both in the modern system and in the historical time domain. In Supplementary Fig. 7 we also directly compare our published high-resolution BIT-index time series for the past 2150 years (29; panel a), and published (20) and new (this study) BIT-index records for the last 20 kyr (panel b), with signatures of past lake-level fluctuation in the seismic stratigraphy of profundal sediments at the DeepCHALLA drill site (19; panel c). The excellent agreement between these two hydroclimate proxies should be especially persuasive because both reflect variation in the balance between continental precipitation and evaporation (P-E; see **R31**), and they are fully independent from one another given that the seismic evidence is not extracted from the drilled sediment sequence.

R14c. Referring specifically to the referee’s recommendation to validate our BIT-index inferences through use of XRF-based proxies of terrigenous mineral input to Lake Chala (assumed proportional to the amount or intensity of precipitation), we deliberately refrained from exploring such data as hydroclimate proxy in this study, because the local combination of a small crater catchment (Supplementary Fig. 1) with a semi-arid tropical climate regime means that terrigenous mineral inputs via wind-blown dust (enhanced during drought episodes) are potentially of similar magnitude as the input via overland flow (enhanced by high rainfall), and thus prone to creating conflicting hydroclimate signatures in the sediment record. Clearly, an independent reconstruction of past effective moisture must be established before the relative influence of fluvial and airborne mineral input on XRF spectra in the Lake Chala record can be discerned.

Supplementary Fig. 7: Coherence between the Chala BIT index and seismic stratigraphy of past lake-depth fluctuation in Lake Chala, two independent proxies of past variation in climatic moisture balance. a) BIT-index record of the last 2150 years at decade-scale resolution (Challacea data: 29), with three episodes of relative drought (majority of BIT index values <0.65) highlighted in yellow. These periods correlate to lenses L0, L1 and L1bis of basin-focused (i.e., lake low-stand) sedimentation in the uppermost ~2 m (122-125 ms TWT) of seismic-reflection stratigraphy at the DeepCHALLA drill site (orange highlights in panel c; modified from 19). The three short-lived BIT-index minima highlighted in pink are coeval with three historically documented episodes of widespread regional drought within the last 200 years (29; see also the review by Nash et al. 2016). b) BIT-index records of the last 20 kyr at century-scale resolution from the Challacea (green; 20) and DeepCHALLA (orange; this study) sites, with prolonged episodes of relative drought highlighted in yellow and correlated to lenses L0-L5 and Unit 3 (containing L6) of basin-focused sedimentation in the uppermost ~23 m (122-146 ms TWT, covering Units 1-3) of seismic-reflection stratigraphy (19) (panel c). Note that the TWT scale (two-way travel time of the seismic-reflection signal) is not a truly linear depth scale, as it is influenced by the relative strength of successive reflectors. The excellent match between the two 20-kyr BIT-index records from drill sites ~650 m apart (Supplementary Fig. 1) further demonstrate i) successful transfer of the $^{210}\text{Pb}/^{14}\text{C}$ -based chronology from the Challacea sequence (72) to the equivalent section of the DeepCHALLA sequence through visual cross-correlation of finely laminated sediments shared by both (this study); and ii) considering that the DeepCHALLA site is closer to shore than the Challacea site, and can thus be expected to receive a greater influx of GDGTs produced in catchment soils, the near-identical BIT-index values at both sites indicate that all brGDGTs used in calculation of the Chala BIT index must result from microbial production within the water column of Lake Chala (27).

R14d. In addition to the direct comparison between the BIT-index time series and seismic stratigraphy provided by Supplementary Fig. 7, a high level of coherence can also be observed between the record of past lake-level fluctuation expressed in the seismic data and the distribution of mm-scale versus

cm-scale lamination in the lithology column (Extended Data Fig. 1a). This is not surprising, considering the dominant control of changes in lake depth on water-column stability, and in turn on the potential for undisturbed deposition of fine-grained soft lake muds (Håkanson & Jansson 1983; Dearing 1999). As documented specifically in African crater lakes (Verschuren 1999b, 2001), greater lake depth (associated with episodes of wetter weather/climate) promotes stable stratification and preservation of mm-scale (i.e., seasonally distinct) sediment lamination; and it tends to slow down sediment accumulation because terrigenous inputs are distributed over a larger area of undisturbed deposition. Conversely, reduced lake depth (associated with drier weather/climate) reduces water-column stability such that recently deposited sediments are more vulnerable to disturbance by near-bottom turbulence (cf. Fig. R1); and it tends to increase the rate of sediment accumulation because terrigenous inputs are focused into a smaller central area of undisturbed deposition. We refrain from detailed presentation of the latter two sedimentary hydroclimate proxies in this paper, firstly because it would require a significant length of text that we deem is better used to elaborate on the climatic implications of our results, and secondly because transitions in both seismic units and lithological facies are threshold-dependent (33; Verschuren 1999b) and thus they are categorical rather than quantitative hydroclimate proxies. The Chala BIT index is at least semi-quantitative (see **R54**) and thus better reflects the rate and magnitude of past hydroclimate changes through time.

15. I find the conclusions very thoughtful and informative, especially with respect to the later Quaternary interval. I remain suspicious however of the correlations to high latitude intervals prior to 25 ka because of the lack of absolute age dating in the deeper section, as discussed above.

R15. We hope that **R8-R11** cover this comment to the referee's satisfaction.

16. The evolutive spectra I found to be of limited value as there is little change in the primary forcing over the duration of the record. There is some change in power of the precessional forcing of MST over the length of the record, but not much change in the signal of the BIT proxy. This may be in part due to the limited sensitivity of this measurement as a moisture proxy. The half-precessional signal seems comparatively weak, and in the case of BIT, the half-precessional signal barely rises above background.

R16a. We respectfully disagree with referee #1 concerning the 'limited sensitivity' of the BIT-index hydroclimate proxy (see **R13**). We also disagree with their apparent suggestion that the relatively weak half-precessional signal is contrary to expectation for this site near the equator. Given the influence of northern hemisphere glaciation on the mean position and seasonal range of the ITCZ (e.g., 59), there is no *a priori* reason why, at any specific location near the equator, half-precessional periodicity in insolation forcing (32) should have caused an equal measure of moisture-balance variation during glacial and interglacial periods.

R16b. We do agree with referee #1 that our wavelet analysis of the BIT-index and MST proxies as presented in the original submission is unconvincing. We followed up on this critique by replacing the wavelet spectrum with band pass filtering (Extended Data Fig. 4, revised), using bandwidths that capture the significant orbital frequencies revealed by Lomb-Scargle spectral analysis. This has the added benefit of providing information about the phasing of the cycles in BIT-index and MST variation, and this now highlights the clear correspondence between the 42.8-kyr frequency band in the data and the timing of marine isotope stages, which are a reflection of changes in global ice volume known to be controlled largely by variation in Earth's tilt (36). Namely, negative phasing of the obliquity-length band pass corresponds to the most extreme glacial stages (MIS4 and MIS2) while positive phasing corresponds to warmer/milder stages (MIS3 and MIS1). Thus, even though the presented 75-kyr proxy record covers barely two obliquity cycles, the phase relationship between the obliquity-scale cyclicity in our proxies and climate changes at high northern latitudes fits with generally accepted understanding of global climate teleconnections during the late Quaternary. Fully aware of the relatively short length of our proxy records in relation to orbital insolation cycles, we deliberately refrained from assessing the nature of orbital insolation forcing on temperature and moisture balance

variation in eastern equatorial Africa because this requires proxy records extending preferably to 130 ka, while we cannot vouch for the robustness of the MST and BIT-index proxies in Lake Chala sediments older than 75 ka.

17. *In summary this is an exceptional project, but expect there must be additional data in-hand (or close to in-hand) that would alleviate my unease about the authors' sole reliance on BIT for a moisture proxy, and on a purely modeled age for the interval 26-73 ka. I hope they will consider including such data in any manuscript revision.*

R17. As explained above (**R8-R11, R12-R14**), we surmise that the unease of referee #1 about the age model and BIT-index proxy may be inspired by a personal reference frame comprised of continental climate archives with less continuity and taphonomical uniformity than that of Lake Chala (**R1, R6**). Nevertheless, we thank the referee for the helpful comments on several aspects of our results and hope that the added primary data and figures (on the 73.7-ka YTT and on validation of the BIT-index proxy), supplemented by our detailed explanations of the age model, alleviate this unease.

Manuscript-specific comments by line number:

18. *44-60 The authors generalize their study site as representative of "East Africa"; however it is well known that there are many sub-regional influences on climate in this part of the continent, including impacts of Atlantic moisture, vagaries of elevation, variable orographic effects on different parts of the monsoon cycle, etc. Whereas possible they should use the term "equatorial East Africa" rather than East Africa", so as not to overrepresent or geographically overextend the importance of their results.*

R18. Given precedence in the pages of this journal (e.g., 20, 21) we generalized our study region as 'East Africa' only in the title to allow expressing the principal result of our study within the two-line limit (~90 characters). Also in the meteorological-climatological literature (e.g., 10, 12-13) we see common use of the term 'East Africa' in studies that do not deal with this sub-continent in its entirety. However, sensitive to the referee's comment we now replaced 'East Africa' in the title by 'the Horn of Africa' as short-hand for the region to which our results apply; and in the introductory paragraphs (lines 81-89) we specify our study region explicitly as the part of eastern equatorial Africa situated east of the Congo Air Boundary, and thus experiencing bi-modal (not unimodal or tri-modal) rainfall seasonality with moisture entirely derived from the Indian Ocean (11,20-21). Politically this region corresponds to the eastern 'Greater Horn of Africa' (lines 84-86). Throughout the text we use shorthand versions of this regional specification for succinctness, but we hope never in a way that would suggest that we are trying to overextend our results.

19. *61only A few....*

R19. OK, now line 72.

20. *71 of EQUATORIAL East Africa*

R20. Changed to "eastern equatorial Africa" (cf. **R18**), now line 80.

21. *119-120 Without additional absolute ages in the 25-75 ka range, I question this statement.*

R21. Text modified, because our earlier orbital periodicity analysis was defective (cf. **R16**).

22. *151-151 This is most important, but prefer to see it framed by additional age constraints.*

R22. We deleted "around the time of H6" (now line 172) to avoid the impression that we would tune our data to match this high-latitude climate event.

23. *169-177 Run-on statement. Please break this up and clarify.*

R23. OK, this sentence is now split in two without adding too many words (now lines 189-198).

24. *197-199 Important point!*

R24. We thank the referee for concurring with this statement (now lines 218-220).

25. 250-254 *Suggest the authors expand on this in the context of the recent multiyear interval of extraordinary high lake levels across the central and northern Kenya rift valley.*

R25. The Kenya rift valley lakes are, to a varying extent, so-called amplifier lakes (Olaka et al. 2010), characterized by a large catchment to lake-area ratio, headwater regions substantially wetter than the rift valley itself, and therefore river inflow representing an important contribution to lake water balance. In this setting, the rising lake levels can be attributed at least partly to the greater river input (and less groundwater recharge) following widespread deforestation of upstream catchment areas for agriculture (Odongo et al. 2015). Exactly how much is at present difficult to assess on a lake-by-lake basis, because of large water extractions from Lake Naivasha since the 1980s by industrial-scale horticulture, redirection of Naivasha headwater streams to Lake Nakuru since the late 1990s, and recent hydrographic changes in the Baringo-Loboi-Bogoria basin. Research into this question requires a long-term perspective, but this is complicated by the fragmentary and/or unreliable historical water-level data and the inundation of many official staff gauges during the lake transgressions in 2011-2012 and 2020. Although difficult to quantify at this time, the portion of the recent lake-level rises that can be attributed uniquely to climate change appears associated with increased westerly moisture advection between June and September, bringing rain to the central Kenya rift valley, and more importantly, reducing lake-surface evaporation during this normally long dry season. However, in the context of our results and conclusion, the essence is that Kenya's central rift valley lakes are not situated eastward of the Congo Air Boundary throughout the year (refs. 11, 21 and Fig. 1) and thus outside of the eastern Greater Horn of Africa region where we think our results apply.

Referee #2

26. *In this manuscript, Baxter et al. provide high-resolution GDGT records from well-dated sediment cores from Lake Chala, East Africa. The results show the variabilities of temperature and hydroclimate in East Africa during the past 75,000 years. The authors argue that the correlation between temperature and hydroclimate are positive before 11,700 years ago but suddenly shifted to negative, which is linked to a tipping point that the pCO₂ reached 250 ppmv. This is a well-written paper, providing some important and interesting insights into the African climate in the past and future.*

R26. We thank referee #2 for their recognition of the quality and importance of our work.

27. *However, I have some concerns about the proxy interpretations. For the temperature reconstructions, the authors select the Pearson et al. (2011) calibration that was established prior to the new method that could differentiate the 5- and 6- methyl brGDGTs. In the Supplementary Discussion, the authors provide some arguments, but I still have some questions:*

1. Does the sum of 5- and 6-methyl actually equal the intensity of using the old method? There seems to be some systemic bias when doing a such calculation. For example, Fig. 4 in De Jonge et al. (2014) and Fig. 2 in Russell et al. (2018).

R27a. Of course, we used the most up-to-date HPLC-MS technique that is capable of separating the 5- and 6-Me isomers (76), as this technique was developed in our lab (NIOZ Royal Netherlands Institute for Sea Research, lines 23-24), and this would indeed allow application of the more recently developed temperature calibrations. Nevertheless, after careful assessment of the options available as described in Supplementary Discussion (lines S303-461), we chose to apply the global lake calibration developed before this separation was achieved (Pearson et al. 2011; ref. 24). To this end we summed the peak area of the 5- and 6- isomers for the penta- and hexamethylated brGDGTs before making the temperature calculation (equation 2 in Methods, line 615). Theoretically, the sum of the individually quantified 5- and 6-Me isomers should be the same as the peaks of the co-eluting penta- and hexamethylated 5- and 6-Me isomers using the old HPLC-MS method. However, this exercise may be complicated by the presence of minor isomers and the presence of a shoulder at the back of the peak of the summed hexamethyl brGDGTs using the old HPLC method, which could be caused by incomplete separation of the 5- and 6-Me isomers, i.e. the point raised by referee #2. To test for this

issue, we have now applied the Pearson et al. calibration (24) to GDGT data from the 25-kyr Challacea sequence (127), which was obtained using the old HPLC-MS method, meaning that peak integrations and fractional abundances represent the coelution of the 5- and 6-Me isomers. In this way, we apply the Pearson et al. calibration to GDGTs in Lake Chala sediments in the most honest way, i.e. with matching HPLC-MS techniques. The result (new Supplementary Fig. 6, reproduced below) is strikingly identical to the MST time series reconstructed from the equivalent DeepCHALLA sequence, particularly when considering that this is a comparison of datasets from two different sediment cores recovered ~650 m apart, not from duplicate analysis on the same samples. Moreover, the small differences in reconstructed MST relate only to the amplitude, not the timing or trend of temperature change (lines S434-452). This clearly demonstrates that we can apply the Pearson et al. temperature calibration to our dataset, despite the potential complications raised by referee #2.

Supplementary Fig. 6: Comparison of MST records for the last 25 kyr from the Challacea and DeepCHALLA sites in Lake Chala, based on GDGT spectra in the Challacea (127) and DeepCHALLA (this study) sediment sequences and both calculated using the Pearson et al. (2011; ref. 24) global lakes calibration.

R27b. The figure from De Jonge et al. (2014; ref. 113) that referee #2 refers to is shown below (Fig. R2), and compares the summed fractional abundance of the individual 5- and 6-Me isomers following application of the new method (y-axis) to the fractional abundances of the hexamethylated brGDGTs as integrated using the old HPLC method with poor separation (x-axis). The caption to that figure as published states that the majority of samples (83%) fall on the 1:1 line, meaning that the fractional abundances are virtually equal, with the points falling above this line representing samples where partial separation meant that a shoulder was excluded. Russell et al. (2018; ref. 115) did the same exercise with the East African lakes dataset (Fig. R3), and here most of the points fall off the 1:1 line, which we surmise is related to the influence of cutting the shoulder off peaks although no mention of the presence/exclusion of a shoulder is given. We do not have information about the level of peak separation in GDGT spectra from the individual samples included in the global lakes dataset of Pearson et al. (2011; ref. 24); for example, whether there was complete overlap of the 5- and 6-Me isomers or whether one of the isomers appeared as a shoulder which then may or may not have been excluded when integrating the peak areas. However, the two studies cited above show that the overall correlation between the fractional abundances as calculated with either method is very strong, and not substantially different from the 1:1 line. Moreover, in case the Pearson et al. (2011; ref. 24) data set mirrors the relationship shown in Russell et al. (2018; ref. 115), any overestimation of brGDGT IIIa is essentially compensated by the underestimation of brGDGT IIa.

Fig. R2. Fig. 4 from De Jonge et al. (2014; ref. 113): “Scatterplot of (*) the fractional abundances of the hexa-methylated brGDGTs as reported in Peterse et al. (2012), versus (**) the summed amounts of the 5- and 6-methyl hexa-methylated brGDGTs calculated in this study. The majority of the samples (83%) plot on the 1/1 line (plotted in black), indicating that it was not possible to separate the 5- and 6 methyl isomers in these soil samples. For samples plotting above the 1/1 line, the separation allowed to cut off a shoulder”.

Fig. R3. Fig. 2. from Russell et al. (2018; ref. 115): “Plot of the fractional abundances of IIIa* and IIa* measured using a Preveil Cyano column (old method, x-axis) that did not separate 5- and 6-methyl brGDGT isomers vs the summed abundances of the 5- and 6-methyl isomers of brGDGTs measured in this study using the new method (y-axis). The majority of the samples fall off the 1:1 line indicating their abundances as measured using the old method were affected by the presence of 6-methyl isomers”.

28. 2. *The East Africa calibrations in Loomis et al. (2012) are excluded because the authors argue that there are not many lakes in the dataset that are similar to Lake Chala (SI Lines 382-392). However, only 16 African lakes are used in Pearson et al. (2011) calibration, which contains 90 lakes in total. Moreover, most of the lakes are way shallower than Lake Chala (Table 1, Pearson et al. (2011)). Therefore, the implication of 6-methyl may not be well constrained in this calibration, given the 6-methyl brGDGTs tend to be produced more in the anoxia deep water (Weber et al., 2018). To date, we’ve known that the 5- and 6-methyl are mainly produced at different depths, as the authors show in Fig. S2a. It is certainly believable that 6-methyl brGDGTs could play a role in reflecting water temperatures. However, we still do not have enough data to quantitatively support this hypothesis. In the meanwhile, the SFS calibration in Russell et al. (2018) seems to work well for the Lake Chala SPM samples (van Bree et al., 2020).*

R28a. To start, based on our detailed studies of the modern system at Lake Chala (26-29,129) we believe that similarities in characteristics such as lake depth and oxycline depth are likely more important than geographical proximity with regard to the ecology of GDGT producers. The fact that the Loomis et al. (2012; ref. 15) calibration consists of East African lake sediments does not on its own make it better suited for Lake Chala. As argued in Supplementary Discussion (lines S420-430), we suspect that lakes from different climate zones are needed to properly express the full range of climate conditions that may be experienced at a single lake across glacial and interglacial periods. It may therefore be significant that in the East African lakes dataset of Loomis et al. (2012; ref. 15) (and of Russell et al., 2018) all lakes from locations with mean annual air temperature (MAAT) below 19 °C are relatively shallow (<33 m) (Fig. R4), hence missing potential analogues for Lake Chala during past

cold climate regimes. By contrast, the Pearson et al. (2011) data set also includes deeper lakes from colder regions (although the published paper only provides ranges of lake depth by region).

R28b. We agree with referee #2 that ideally the link between the 6-Me brGDGTs and temperature would better be studied from their distribution in recently deposited lake sediments, considering that in our record the 6-Me brGDGTs drive the reconstructed cooler temperatures towards the ends of MIS4 and MIS2 (i.e., the late-glacial period) (Fig. 2e). Ideally it should also be investigated in well-dated sediments from the last glacial period at a sizable number of sites worldwide, but this represents a huge logistic and analytical effort. Indeed, if we were able to compare GDGT distributions in glacial sediments to regional glacial temperatures reconstructed from other proxies, a stronger relationship between 6-Me brGDGT methylation and temperature might be observed than what exists today using sediments deposited in a warm interglacial period.

Fig. R4. Mean annual air temperature (MAAT) versus lake depth for the 65 lakes included in the East African lakes calibration of ref. 115. The red dashed line indicates the depth of Lake Chala.

R28c. In line with our approach, we are aware of at least one other lake-based paleoclimate study from eastern Africa (84) where the authors determined that inclusion of the acyclic hexamethylated 6-Me brGDGT (IIIa') in the calibration model was required to credibly estimate Holocene temperature variation. Also, in a set of 17 alkaline Chinese lakes spanning a temperature gradient of 12.7 – 17 °C (128) significant correlations were found between both MBT' and MAT and between MBT'_{6Me} and MAT ($r^2 = 0.64-0.66$) but not between MBT'_{5Me} and MAT, suggesting that methylation of the 6-Me brGDGTs may be a temperature adaptation strategy employed by brGDGT producers living in (alkaline) lakes, in contrast to what is observed in soils where their contribution is linked to pH.

R28d. Lastly we would like to emphasize that the Lake Chala study by van Bree et al. (2020; ref. 27) did not apply the SFS calibration of Russell et al. (2018) to SPM samples, but to multi-year monthly samples of settling particles collected by a sediment trap deployed at a water depth of 35 m, meaning that the brGDGTs they contain came only from the upper 35 m of the water column, not from the anoxic deep-water zone where their production is greatest. Hence, this comparison between estimated and measured MAAT cannot be used to assess the appropriateness of applying the Russell et al. (2018) calibration to the sediment record of Lake Chala.

29. 3. *The records using EAL calibrations (Russell et al., 2018) (Fig. S1c) make sense in a way that the temperature started to increase around the end of LGM which agrees with the onset of Rwenzori*

glacier deglaciation (Jackson et al., 2019). In fact, the ensembled temperatures in Extended Data Figure 3b also show this feature, which contrasts with the Lake Chala MST.

R29a. When applied to the Chala sediment record, the Russell et al. (2018) EAL calibrations infer that the coolest temperatures of the last glacial period occurred around ~52 ka in early MIS3, followed by a prolonged warming trend such that the period of peak glaciation (MIS2) is expressed as no more than a modest and short-lived dip in temperature (Supplementary Fig. 2c). While interesting in isolation, climate-dynamically this result is incongruent with general understanding that globally greater cooling occurred during MIS2 and MIS4, while more modest cooling occurred during MIS3 (e.g., 34,36). Also, the EAL-MBT'_{5Me} calibration infers that highest temperatures of the last 75 ka occurred during the late-glacial period including the Younger Dryas (16-11.5 ka, i.e. before the start of the Holocene), incongruent with general understanding that the current interglacial period has been warmer, on average, than any time during the preceding glacial period; and further both the EAL-SFS and EAL-MBT'_{5Me} calibrations infer that the early Holocene was cooler than the mid- to late Holocene (Supplementary Fig. 2c), incongruent with recent syntheses for the global tropics (38). With regard to the timing of post-glacial warming, in both of our reconstructions based on Russell et al. (2018) calibrations it is inferred to have started around 23 ka, still ~3000 years before the onset of reconstructed warming and glacier retreat in the Rwenzori mountains (~20 ka: 85 and Jackson et al. 2019) and ~5000 years before the start of post-glacial warming as represented in the eastern African ensemble reconstruction (~18 ka; Extended Data Fig. 5b). Moreover, there are logical geographical reasons why the timing of post-glacial warming in a lowland region of easternmost tropical Africa would differ from that at high elevation in the continental interior. As Lake Chala (and the entire Horn of Africa region) is located east of the Congo Air Boundary year-round, we expect to see strong similarity with MIS2 temperature trends reconstructed for the western Indian Ocean and continental southern Africa; and that is indeed the case in our MST record based on the Pearson et al (2011) calibration (Extended Data Fig. 5c-e). The strong coherence between these three reconstructions is particularly comforting, because they are based on three fully independent paleotemperature proxies (respectively lacustrine GDGTs, marine alkenones and fossil pollen assemblages: this study, 52 and 37), AND they cover both the continent and the adjacent ocean.

R29b. In specific reference to Jackson et al. (2019), our results allow an expanded explanation for the onset of Rwenzori deglaciation around 20 ka. From earlier studies we know that along the western margin of the East African Plateau, a region receiving moisture both from the tropical Atlantic Ocean (the African monsoon; Fig. 1) and the Indian Ocean (the Northeasterly and Southeasterly monsoons), precipitation and/or effective moisture during the last glacial period was strongly reduced from at least the start of MIS2 (~30 ka; 14) and probably already from ~37 ka (47). This implies that the expanded Rwenzori glaciers that existed between ~29 ka and 20 ka (Jackson et al. 2019) must have been sustained primarily with snowfall from moist easterly air originating in the Indian Ocean, and indeed our BIT-index data (Fig. 2d) show high effective moisture during most of this period but abruptly declining between 22 ka and 20 ka. Thus, we surmise that glacier retreat in the Rwenzori started ~20 ka due to the snow starvation resulting from strong reduction in moist easterly air flow, in addition to the increasing temperature suggested by Jackson et al. (2019). By ~16-15 ka, post-glacial warming had progressed to the point where even the reinvigorating monsoon circulation in both the tropical Atlantic (e.g., 44) and western Indian Ocean (e.g., 20) could not halt further glacier retreat.

30. 4. *I'm also curious about the temperature rescaling process. It seems like melt-water input from the retreating Mt. Kilimanjaro is a plausible reason for the Lake Chala cooling phase between 20 and 13 ka BP (Sinninghe Damsté et al., 2012), which is also apparently the coolest period in the 75-kyr MST records. How would this potentially affect the rescaling since it's hard to estimate the contribution from the meltwater?*

R30. Sinninghe Damsté et al. (2012; ref. 127) speculated about the possibility that a large input of melt-water from the retreating glacial-era ice cap on Mt. Kilimanjaro may have caused anomalous cooling of Lake Chala (i.e., lake water being substantially colder than ambient air temperature) during

the period 20-13 ka before publication of the pollen-based temperature reconstruction for southern Africa (Chevalier & Chase 2015; ref. 37) stimulated integration of Lake Chala GDGT records with the most pertinent and best-available marine data from the western Indian Ocean (52). While the possible influence of cool meltwater seemed worth mentioning a decade ago, we now deem this explanation implausible because even above treeline on Mt. Kilimanjaro little trace exists of a formerly active river course that may have directed a large amount of ancient meltwater towards Lake Chala, in line with the fact that in this rather dry tropical region (90), mass loss from the modern-day Mt. Kilimanjaro ice cap mostly occurs through sublimation rather than melting (Kaser et al. 2004; McKenzie et al. 2010).

31. *I also have some thoughts on the interpretation of the BIT index. First of all, the representative of BIT is rather vague. It's not very clear to me whether is this a proxy for precipitation amount, P-E, lake level, or other stuff.*

R31. As explained in published previous work (28) and elaborated upon in this paper (lines 111-119, S501-539) the Chala BIT index is a moisture-balance proxy in which the spatial and temporal distribution of GDGT-producing microbiota is linked to effective moisture (i.e., climatic moisture balance or P-E) via climate-driven changes in the depth and water-column structure of Lake Chala. The latter is demonstrated by the high congruence between BIT-index values and changes in lake depth inferred from seismic stratigraphy (Supplementary Fig. 7), and between lake depth and water-column mixing regime as reflected in the preservation or partial erasure of mm-scale lamination (Extended Data Fig. 1a). As explained above (**R14b**) the strong link between the water balance of Lake Chala (as reflected in the BIT index) and hydroclimate is demonstrated by its truthful registration of the three regionally most severe decade-scale drought episodes in the last 200 years (29; Supplementary Fig. 7). We refrain from suggesting that the Chala BIT index reflects the regional balance between precipitation and evaporation (P-E) because of uncertainty in how to quantify this relationship (**R54**). However, this is an issue common to the large majority of hydroclimate proxies in natural climate archives, not only those extracted from lake sediments but also those in speleothems or tree rings. Inspired by comments 5 and 12 of referee #1 and to avoid the impression that we can quantify all water inputs and losses to our study system either in the present or the past, we now complement 'moisture balance' with the more neutral term 'effective moisture'. The latter term is comparable to what would be termed 'available water' or 'water availability' in hydrological literature.

32. *Moreover, the schematic mechanism in Figure S2 is a good hypothesis but needs more testification. For example, Fig. c Past low lake level. Even if the lake level was drastically lower than today, the depths of upper zones should remain the same, while the bottom water might be the only part that became 'shallower'.*

R32. With all respect, considering the volume of our modern system studies (see **R12a**) we are at a loss what additional testification we could do. We agree that when lake level was lower than today, the combined thickness of the upper mixing zones may have stayed more or less the same, while the thickness of the permanently anoxic lower zones was reduced. That is, unless elevated concentrations of dissolved solids in the lower water column created a chemocline strong enough to hamper deep mixing, such that also the more dilute upper zones would have become thinner or 'shallower' (126). The apparent confusion on the part of referee #2 may be due to the fact that whereas panels a and b in the original Fig. S2 represented the two seasonal extremes in relative thickness of different water-column zones in the modern system, panel c (the lake low-stand scenario) represented an annually-integrated situation designed to aid interpretation of the sedimentary GDGTs (of which the distributions are also integrated across seasons, because each analysed 2-cm thick sediment increment integrates ~25 years of sedimentation); hence we placed the oxycline at a position intermediate between the two seasonal extremes. But of course, we envision that today's seasonal extremes also occurred during a past low lake level situation. Understanding that readers may be similarly confused as referee #2, we have expanded the figure (now Supplementary Fig. 3) with two panels (c-d) representing the hypothetical seasonal extremes of a past low lake level scenario, which are then combined into the hypothetical annually-integrated situation (panel e = the original panel c).

33. *In addition, the authors argue that ‘... low BIT index values reflect periods of enhanced mixing and a depressed oxycline due to dry and windy conditions when the ITCZ is further afield.’ (SI Lines 457-458); while this seems to have some inconsistency with the Lake Chala SPM results (Baxter et al., 2021). In Figure 6F in Baxter et al. (2021), the BIT indexes tend to be lower during the upper water-column stratification. Collectively, I believe this manuscript would benefit from a better clarification of interpreting the BIT index.*

R33a. We appreciate that the mechanism linking the Chala BIT index to climatic moisture balance is novel and that the proxy has not been applied as such in other lake systems, justifying the need for detailed explanation of our proxy interpretation. Accordingly we expanded our proxy explanation in Supplementary Discussion (lines S501-561). However, we do feel that much of the confusion regarding the Chala BIT index as moisture balance proxy would be alleviated by attentive reading of our published previous work, and in particular Baxter et al. (2021; ref. 28) which explicitly set out to determine and clarify the mechanism involved, both in the modern system of Lake Chala and in its sediment record on glacial-interglacial time scales.

R33b. As concerns the reference of Referee #2 to Fig. 6 of Baxter et al. (2021), we note again (cf. **R28d**) that those BIT-index data are not derived from SPM but from settling particles trapped at 35 m depth, and thus miss out on most of the brGDGT production in the anoxic zone as well as most of the crenarchaeol production by Group I.1a Thaumarchaeota (concentrated just above the oxycline) during the deep mixing period, when the oxycline is depressed (129). Thus, the trapped settling particles showed increased amounts of crenarchaeol, creating low BIT-index values, only during the stratified season when the oxycline is above the 35-m trap depth. In summary, seasonal BIT-index variation in the sediment-trap data can be opposite to that which can be expected in depth-integrated SPM data (or if the sediment trap had been installed just above the lake bottom). Unfortunately, the 17-month period in 2013-2015 covered by our SPM data series is unusual in that only Group I.1b Thaumarchaeotal gene copies were identified, and Group I.1a Thaumarchaeotal gene copies were not detected (28), dissimilar to SPM profiles from earlier years of lake monitoring when Group I.1a Thaumarchaeota were the dominant species (91,129).

34. *In the manuscript, the authors imply that the inverse relationship happened when the pCO₂ reached a tipping point. I feel that this part needs a more thorough discussion. Some of the hypotheses were provided on Lines 226-244 but none of them seems related to pCO₂.*

R34. When considering climate-change impacts on the global biosphere one naturally must distinguish between the direct effect of pCO₂ on plant photosynthesis and its indirect effect on plant physiology through the warming it causes. However, for the present study the principal effect of elevated atmospheric CO₂ concentrations on continental hydrology is through its warming of the atmosphere, and how this affects both the generation and advection of precipitation and the amount of land evaporation, such that below/above a region-specific temperature threshold either its influence on precipitation or on evaporation predominates (lines 251-262). In this particular region, this appears to be the case at an MST value of ~23 °C, as was consistently approached or exceeded from 11.7 ka onward as atmospheric CO₂ rose above ~250 ppmv (Fig. 2e-f and Fig. 3b; lines 280-285). This part of the paper was revised and bolstered with important contributions by hydrologist Dr. Diego Miralles, justifying his co-authorship on the revised manuscript.

Minor comments:

35. *Lines 53-55 This sentence seems irrelevant at this spot and interrupts the coherence.*

R35. In our opinion it remains useful to juxtapose the specific contributions of weather monitoring, climate modelling and paleoclimate reconstruction in resolving the Eastern African climate paradox, but we rephrased the sentence (lines 64-71) to improve coherence with text before and after.

36. *SI Lines 352-354 Raberg et al. (2021) should not be cited here. It seems true that the 6-methyl within the Meth set is somewhat correlated to MAF (Fig. 7 in Raberg et al. (2021)); however, the*

methylation of 6-me brGDGTs shows very poor correlations (Fig. 3 in Raberg et al. (2021)). It is likely because all those 6-methyl brGDGTs show negative correlations with temperature (Fig. 7 in Raberg et al. (2021)). While for the 5-methyl brGDGTs, the brGDGT-Ia is always correlated to temperature positively while the IIIa has a negative correlation. It also should be noted that the 6-methyl has moderate correlations with a variety of environmental factors, based on the Martínez-Sosa et al. (2021), such as pH, precipitation, and conductivity.

R36. Thank you for bringing this to our attention, indeed the reference to Raberg et al. (2021; ref. 125) is not appropriate at this location. In addition, we now mention other environmental factors to which the 6-Me brGDGTs are moderately correlated (lines S323-324).

37. *Fig 1. The topography does not really provide useful information here. Also, please double-check the elevation data. I think the Tibet plateau shouldn't be that tall.*

R37. We find the topography within Africa in Fig. 1 useful because it shows that the position of the CAB during NH summer (and thus the area of eastern Africa fully dependent on Indian Ocean moisture) is partly controlled by the mountainous western margin of the East African Plateau. We modified the colour scale so that the high-elevation areas in eastern Africa stand out more clearly, and most of the Tibetan Plateau is now coloured beige. We also added an outline of the eastern Greater Horn of Africa region where the results of this study apply, based on the bi-modal rainfall seasonality and total dependence on Indian Ocean moisture characterising this region (11). Moreover, this areal delineation is consistent with a synthesis of spatial patterns in eastern African climate variability within the last millennium (21), emphasizing the role of the Congo Air Boundary as convergence zone between eastern and western moisture flows.

Referee #3

38. *Baxter et al., present a new GDGT biomarker record, interpreted as mean summer temperature and moisture balance, from Lake Challa in eastern Africa over the last 75 kyr. The authors find that prior to the Holocene, temperature and moisture balance were correlated, placing importance on certain mechanisms within the climate system. Then, coinciding with the deglacial increase in global CO₂, the two parameters become anti-correlated. This suggests that the increasing global temperatures due to anthropogenic climate change will drive further aridification in the region, contradicting climate models that predict more extreme rainfall. The new reconstructions are of unprecedented temporal resolution for GDGT biomarkers, and the age model is fantastic in the top third of the record, where much of the conclusions are based.*

R38. We thank referee #3 for these nice words, which concur with our opinion about what makes our results both important and robust.

39. *One thing I really like about this paper is how well-motivated the study is. Quantifying the responses of this region to future warming is of utmost importance, and I think there could be more at the back end of the manuscript circling back to this original motivation.*

R39. We expanded both the first and last paragraphs of the main text to accommodate this suggestion, while remaining alert to the recommended word limit; see also **R3b-c** and **R34**.

However, there are some major concerns relating to some supporting evidence of the conclusions. Please find my major, minor, and technical comments and corrections below.

Major Comments:

40. 1. Literature. *I want to first off mention that I firmly disagree with the concept of journal-imposed reference limits, as I think that this inhibits the documentation of thorough literature review and support of claims.*

R40. We are inclined to agree with referee #3 on this point, as the requested inclusion of primary data on the YTT cryptotephra and detailed explanation of the mechanistic link between the Chala BIT index

and hydroclimate has increased our total number of references (including those in supplementary materials) from 107 to 131. However we also pruned our earlier citations, so that the number of main-text references (including those in main figure captions) has increased only slightly from 68 to 70. We find this number adequate to link our results to the most pertinent literature.

41. *That being said, there are statements within the manuscript such as beginning on line 61 (“only few well-resolved climate records ...”), that I feel go beyond the simple omission of citations, and need to be revised. There has been a whole body of work in eastern Africa (e.g., Foerster et al. 2018) that falls into these categories and should be cited and explored further.*

R41. Referee #3 refers here to climatic and paleoenvironmental inferences from the ~620-kyr sedimentary record of the Chew Bahir basin in southern Ethiopia (e.g., Duesing et al. 2021, Trauth et al. 2021, Foerster et al. 2022), one of the sites drilled in the ICDP ‘Hominin Sites and Paleolakes Drilling Project’ (HSPDP) project. While the results of this project are impressive in their own regard, as stated above (**R6**) the Chew Bahir project represents the geological approach to paleoclimate reconstruction, with ‘deep-time’ climate inferences largely disconnected from the modern-day system for the simple reason that under today’s climate regime the drilled lake-bed is most often dry. The eventful lithostratigraphy of the Chew Bahir sediment sequence, with prevalence of silty-sandy facies (Roberts et al. 2021) and evidence for abrupt changes in sedimentation rate (Foerster et al. 2018) indicates that episodes of desiccation or non-deposition have occurred at multiple timescales (Trauth et al. 2015; Roberts et al. 2021), and that even during past periods when Chew Bahir was a permanent lake, basin-central sedimentation often suffered from shifting sedimentation with redeposition of sediments previously deposited elsewhere (cf. **R6c** and Fig. R1). This highly dynamical depositional environment at Chew Bahir implies that short-term variability in a high-resolution climate proxy, such as XRF-measured mineral elements (or elemental ratios) of terrigenous input, may well be signatures of shifting and interrupted sedimentation (i.e., taphonomical processes) rather than of the short-term climate variability one aims to resolve. Crucial for lake-based paleoclimate reconstruction as we approach it, given our principal aim to directly compare and integrate project results with well-resolved global reference climate records, is that continuity of the climate archive can be demonstrated at a timescale shorter than the analytical resolution of the proxy used to represent past climate variability. The scarcity of existing long African lake records meeting this requirement complicates direct comparison of climate proxy time series between African sites to the extent that we deem such regional data integration feasible only when there is ample writing space to discuss record-specific complexities in proxy formation and preservation, and such effort is clearly outside the scope of this paper. Instead, by targeted selection of our study site (aiming for depositional continuity and taphonomical uniformity of the sediment sequence rather than maximizing the length of time covered) allowed us to exclude several confounding factors that affect climate-proxy interpretation elsewhere.

42. *2. Frequency analysis. With a 75 kyr-long record, the spectral and wavelet analyses and their respective conclusions need to be revised. It is clear in Figure ED 2 that the cone of influence does not even touch the 41 kyr cycle. That is because you need at least 3 cycles, preferably more, within the length of a record to document any significance. It isn’t that the 41 kyr cycle is not there, it is that this particular record (although with wonderful resolution!) is simply not long enough to comment on obliquity forcing. I recommend any mention of cycles above precession be removed from the text. I also recommend updating the figures to not even show these cycles that aren’t resolvable, and perhaps the color scale will change in a way that highlights the half-precession cycle or something else better.*

R42. We accept the critique of referees #1 and #3 on this point, and have now replaced the wavelet spectrum with band pass filtering; see **R16b** and the revised Extended Data Fig. 4.

43. *3. Mechanisms. The mechanisms that are discussed as drivers of the variability and relationships between moisture balance and MST are thorough, yet I feel like 1, the data are over-interpreted and*

2, the way that this section is written is too matter-of-fact. Starting with over-interpretation, there appear to be multiple steps between a highly-resolved coupled summer temperature / BIT index record and major global climate system features. Does the increase of CO₂ drive the change in correlation at 11 ka? Without forcing experiments and a fuller study, I warn against mistaking correlation as causation.

R43a. Most certainly we can all agree that correlation does not imply causation. However, in a research field where proof of causation cannot be established through real-life experimentation; where modelling may remain inconclusive due to defects in the models themselves (see **R3a**); and where trustworthy instrumental data at the relevant time scale are scarce and/or inconclusive, our best option is to propose a mechanistic scenario that explains the observed correlation in the data on the basis of known processes operating at the study site and at the relevant time scale, and to show that alternative scenarios are less plausible in this specific regional and temporal context. We hope that the revised final paragraphs of our Discussion can be considered successful in this regard.

R43b. With regard to whether rising CO₂ drives the change in correlation at 11.7 ka, we refer to **R34**.

44. I do think it is good to discuss all of the potential processes, which are there, but the language needs to be toned down to indicate that these are possibilities based on the literature and what this new dataset supports, rather than a firm statement like in lines 169-177 (which needs to be broken up in multiple sentences) suggesting that the new record is definitive proof of a fairly convoluted series of steps. The new data does not confirm that Indian Ocean SST was unusually low (lines 198-199), but it could be consistent with that finding. Some reworking of the text in the latter half of the paper on the discussion of possible forcing mechanisms needs to be done.

R44. We understand that our statements may have sounded firmer than intended, mainly because we wanted to keep them concise. These parts of the text have now been rewritten and slightly expanded to become a more balanced reflection of what can be concluded from our data; see lines 189-198, 218-220 and **R23-R24**.

45. Further, the study is motivated (very well!) by the issue that CO₂ is rising and we are unsure of how EA will respond. This is mentioned again at the end, and the authors highlight the CO₂ threshold, but then temperature itself is invoked as the driver of the shift in correlation. This is a bit confusing, and goes back to my original point that perhaps a regional record of BIT index and GDGT summer temp over one g-ig cycle with no modeling efforts is not robust enough to make these big, impactful conclusions.

R45a. Referring to **R34**, the shift in correlation was driven by the postglacial rise in CO₂, which warmed the global atmosphere until mean annual temperature in eastern equatorial Africa consistently reached and exceeded ~23 °C at sea level (Fig. 2e), which appears to be a regional threshold temperature above which the positive influence of temperature on continental evaporation dominates over its positive influence on monsoonal precipitation (lines 46-48 and 253-262).

R45b. With respect, we find it somewhat unfair that referee #3 appears to reduce our data to just two observations, namely one glacial and one interglacial. Actually, our conclusion is based on the highly significant contrast in the region's temperature-moisture relationship between the last glacial period (243 observations, MST range 20.3-23.6 °C, BIT-index range 0.24-1.0) and the Holocene (37 observations, MST range 21.7-25.1 °C, BIT-index range 0.36-1.0), with the latter being the interglacial period that anthropogenic warming will likely prolong beyond its normal duration. In an ideal world there would be no limit to the volume of data that can be assembled to address a particular research question. Unfortunately, the number of natural climate archives with the required length and resolution, and containing well-constrained climate proxies, is scarce. Moreover, in terms of orbital insolation drivers, each interglacial period of the late Quaternary is different. Indeed depending on the criteria applied, either MIS9 (~330,000 years ago), MIS11 (~400,000 years ago) or MIS19 (~780,000 years ago) have been suggested by various authors as best analogue to the Holocene in terms of

orbital insolation forcing (Loutre & Berger 2003; Ruddiman 2007; Tzedakis 2010; Yin & Berger 2015), and so the exact CO₂ threshold value driving the shift between moisture and temperature in eastern Africa may be different in each case. Surely, we cannot be expected to postpone making an insightful statement about our data until the paired high-resolution reconstruction of regional temperature and moisture covers multiple glacial cycles (i.e., well beyond the age of the Lake Chala basin).

46. *There is also at times some circular logic, like lines 204-206, that needs to be explained in more detail.*

R46. Sorry, we can't put our finger on why this sentence appears to suffer from circular reasoning.

47. *Two more moderate issues are the 1, the BIT index values of 1 and 0 should be removed from any sort of correlation analysis [...]*

R47. We firmly disagree with referee #3 on this issue. First, BIT-index values never equal 0 throughout our record, and have a value of 1 in only 3 out of 373 analysed sediment horizons (cf. **R13**). Evidently, a regression analysis excluding the latter 3 samples does not change the strength of the correlations within the subsets of glacial and interglacial data points (Fig. R5), nor does it change the timing of the tipping point between these two periods. More fundamentally, the instances where the BIT-index value approaches 1 do not result from defective measurement or missing data: it simply means that crenarchaeol was below detection limit in the GDGT analyses of these sediment horizons. As we discuss in the supplementary materials (lines S540-549), the BIT index does appear to have reduced sensitivity under very wet climate conditions, meaning that it becomes a categorical proxy rather than a (semi-)quantitative proxy (**R13**). In our opinion, as long as the data point is robust, it contains useful information and consequently should be retained in the analysis.

Fig. R5: Correlations between the Chala BIT index and rescaled MST before (blue points) and after (red points) 11.7 ka as in our original submission (left) and the identical regressions but excluding the three sediment horizons with BIT-index value = 1 (right).

48. *[...] and 2, summer temp and moisture balance are not annual temp and rainfall, and so more discussion into how powerful these are for interpreting the climate cycle should be included.*

R48. It is common for brGDGT paleothermometers to be based on correlations with warm-season temperatures or with the average temperature of the months above freezing (MAF: 116, 125), the idea being that the growth rate of brGDGT producers is reduced during the colder seasons. This has been substantiated by a study on a cultured strain of brGDGT-producing Acidobacteria, which exhibited the highest growth rates under elevated temperatures (Halamka et al. 2022). This implies

that any temperature estimate based on down-core brGDGT distributions intrinsically has some bias towards local warm-season temperature, and hence is more appropriately represented as mean summer temperature (MST) than mean annual temperature (MAT). However, considering the subdued temperature seasonality near the equator (7 °C in the Chala region; 90), summer temperature (i.e., maximum mean monthly temperature) is much closer to mean annual temperature in our study region than in higher-latitude regions.

Minor Comments:

49. - *Please refer to the study region as eastern Africa, as East Africa is a political designation and eastern Africa is a geographic one.*

R49. We generally agree, and have consistently replaced 'equatorial East Africa' with 'eastern equatorial Africa' (**R18**).

50. - *I've never heard it be called the east African climate paradox, and this isn't mentioned anywhere else in the paper, so I think it's fine to remove*

R50. A quick search in Web of Science yielded eight publications in reputable climatology and hydrology journals since 2015 which refer to the East African climate paradox in title or abstract (e.g., 11-13; Mölg & Pickler 2022) although in the most recent publications it is referred to as 'eastern African climate-change paradox'. An important objective of this study is to produce paleoclimate data and interpretations contributing to unresolved issues in anthropogenic climate change relevant to the projection and evaluation of future climate scenarios. The referee's comment reinforces our feeling that this kind of discipline-bridging approach is greatly needed.

51. - *More citations for climate dynamics intro line 69-74*

R51. We added mention of the two major monsoon systems in this region (line 83, cf. Fig. 1), but otherwise have aimed for succinctness.

52. - *Please be more up front about the age model, which does have many constraints, but they are 99% concentrated in the top 25 kyr of the record. Make this clear.*

R52. We hope that our responses to the age-model related comments of referee #1 (**R1, R8, R10-11**) alleviate the concerns of referee #3 in this matter.

53. - *Explain use of moisture balance – P-E? how does this relate to BIT? Lines 95-96.*

R53. Correct, we use moisture balance in the sense of P-E but avoided use of the latter term because the exact relationship between the Chala BIT index and P-E has not been quantified, as is commonly the case with hydroclimate proxies extracted from natural climate archives. As stated above (**R31**), we now complement 'moisture balance' with the more neutral term 'effective moisture'. How the Chala BIT index relates to this climate variable is explained on lines 111-119 and S501-539, and recapitulated in the Discussion (lines 264-268).

54. - *Calling temp interpretation MST but moisture-balance BIT-index is perhaps misleading (One is the climate parameter and one is the index). BIT index has been interpreted other ways, so maybe abbreviate to MB?*

R54. We feel that our current designation of the two climate proxies is appropriate. MST is a GDGT-based proxy calibrated to modern-day measurements of summer temperature, whereas the BIT index is a GDGT-based proxy not rooted to any specific modern-day climate variable but represents different processes depending on the setting where it is applied. For example, in the marine realm the BIT index is often used as an indicator of river discharge or more general the influx of terrestrial material, which in turn may be linked semi-quantitatively to the amount or intensity of continental rainfall. In Lake Chala, the BIT index is a semiquantitative representation of climatic moisture balance through changes in lake depth and water column structure (cf. **R31**) but is not calibrated to exact values of P-E (in mm yr⁻¹). Rainfall increase in eastern equatorial Africa during the early Holocene relative to today is

estimated to have been on the order of ~30% (e.g., Bergner et al. 2003), corresponding to ~0.4 BIT units in our data (Fig. 2d). Lake-level changes experienced by East African rift lakes within the historical period are due to sustained rainfall anomalies on the order of ~10% (Nicholson & Yin 2001), and in our data BIT-index values vary within ~0.10-0.15 during the last century (new Supplementary Fig. 7). However, we feel there is not enough solid data available to calibrate the Chala BIT index over its entire range of variation expressed in our 75-kyr record, particularly as it does not reflect precipitation as such but effective moisture.

55. - *Line 116: what climate proxy? Be more specific and perhaps use a different term than “climate-proxy” when meaning climate reconstruction or proxy records?*

R55. The record of hydroclimate variation shown by the BIT index and seismic stratigraphy, i.e. the same two hydroclimate proxies as used in the present study. We now specify that this statement about half-precessional periodicity refers to hydroclimate variation within the last 25 kyr (lines 134-137).

56. - *If GDGT temp is defined and supported as MST, continue calling it MST and not temperature*

R56. Agreed, except when focus is on the source of the GDGTs or on development of the MST proxy as such.

57. - *Line 151 and beyond: drought refers to shorter events with certain impacts – perhaps change the language to extreme aridity*

R57. In climatology and hydrology literature, ‘extreme aridity’ refers to the climate regime with extremely large moisture deficit, such as occurs today in the central Sahara desert. We do not think that such climate regime (with virtually no rainfall at all) ever occurred in the Lake Chala region within the past 75 kyr. The term ‘drought’ refers to an episode of undefined duration when weather conditions are substantially drier than the long-term local mean; in values of absolute rainfall or P-E it means different things in different places. In context of our results, we refer to the latter phenomenon.

58. - *Line 165: it’s tricky to compare proxies for different things from different regions – tone down these conclusions here about forest contraction with a BIT-index record*

R58. The two principal climate variables determining vegetation composition (or biome distribution) in tropical Africa today are mean annual rainfall and the duration of seasonal drought (e.g., Gritti et al. 2010, Philippon et al. 2019), meaning that plants generally respond to variation in effective moisture. Also the paleovegetation literature from West-Africa (45-47) refers to dry climate conditions during the period of forest contraction, supporting the validity of our east-west comparison.

59. - *Line 182: I don’t think it’s necessary to recall this as a mystery interval*

R59. Agreed, sentence rephrased and reference removed.

60. - *Line 241-244: this kind of background info on drought should be introduced earlier*

R60. We now moved some of this information to the BIT-index explanation on lines 111-119, but also keep the full train of thought in the final paragraphs to better highlight the present-day inverse relationship between effective moisture and temperature.

61. - *Line 249-250: this is an important point that needs to be expanded on more and earlier – that this is how it is today, but paleoclimate records x, y, and z show otherwise going further back in time*

R61. We feel that a more general discussion of the relationship between moisture balance and CO₂ through a larger part of Earth history, and in different regions, is outside the scope of this paper.

Technical corrections:

62. - *Line 30: which allows... us? for the?*

R62. OK, sentence rephrased (lines 37-42).

63. - *Last line of abstract: make it clear this is about future climate change*

R63. Together with our explicit mention of “diminishing future water resources” (line 50), we hope it is clear that “continued” (line 49) refers to a prolongation of current trends into the future.

64. - *Line 67: change to *depositionally-continuous and permanently-stratified**

R64. The studied sediment sequence is depositionally continuous (line 79). The lake’s water column is permanently stratified today (meromictic, i.e. not mixing completely every year; Lewis 1983), but in the past 75,000 years it may have mixed completely occasionally, at least during periods when cm-scale banded sediments were deposited. However, such mixing events were likely short-lived even at the seasonal scale (i.e., days rather than weeks; 94) and hence the lower water column most probably remained permanently anoxic throughout the past 75,000 years (lines 92-93).

65. - *Line 75: varve refers to annual no matter the depth scale*

R65. Agreed, but as they have not all been counted (currently two sections of 2609 and 2510 years have been counted; 74 and lines S51-56) we do not want to claim that the mm-scale laminated sediments represent varves throughout, and therefore use the more careful term ‘varve-like’.

66. - *Line 76: not sure what is meant by quiet*

R66. This is now replaced by “turbulence-free” (line 92).

67. - *Line 82-83: grammar seems off here, perhaps rephrase*

R67. Now replaced by “testifying that profundal sedimentation dynamics have been highly stable through time” (lines 98-99).

68. - *Line 90: change to *globally-distributed**

R68. The meaning seems clear to us, and hyphenation is not needed here (line 108).

69. - *Line 104: change to *can be assumed to be stable...**

R69. “...can be assumed stable through time” (now line 124) is stylistically correct, and more concise.

70. - *Line 149: move the word “often”, perhaps before prevailed*

R70. OK, done (line 170).

71. - *Line 153: remove “is known to have”*

R71. OK, done (line 174).

72. - *Line 155: change to *BIT-index* (although other times it’s referred to as *BIT index – be consistent*), but again, it would be best to call this moisture balance or change *MST* to *brGDGT* calibration*

R72. OK, replaced by “BIT-index minima” (line 175). Following common style manuals, we apply hyphenation only when “BIT-index” is a compound adjective.

73. - *Line 248: *conducive of**

R73. The final paragraph has been rewritten, resolving this issue.

74. - *Fig 1: the colors and arrows everywhere is not very effective at introducing the study location. I don’t think it’s necessary to include the Hulu-Dongge cave site. Make sure it is color-blind appropriate.*

R74. The wind-vector and monsoon-system arrows are now set in thinner lines, the warm and cool ocean domains are delineated using more pastel-coloured hues, and the elevational gradient is more pronounced to draw attention to the continents. We feel it is useful to show the Hulu-Dongge site as its record is the principal reference record for past tropical monsoon dynamics but does represent a different monsoon system. High congruence between our data and the Hulu-Dongge record

underscores the strong climate-dynamical links between the western and (far) eastern parts of the Indian Ocean. We have now also indicated the locations of the most pertinent long hydroclimate proxy records in equatorial West Africa (46-47), cited in the text and representing monsoon dynamics in the eastern tropical Atlantic Ocean.

75. - Fig 2: it's very difficult to see the error region of MST – and is it just a simple 2 degrees? Please use error propagation from the respective calibration

R75. The systematic error of the Pearson et al. (2011) calibration is 2 °C, after rescaling of our MST proxy time series this becomes 0.7 °C. This is mentioned in the caption to Fig. 2.

76. - Fig extended 2: I commented on this above – please redo these figures to eliminate periodicities that are not resolved by the dataset. Add (kyr) to the Period labels.

R76. Agreed, see **R16b** and **R42**.

Additional references, used in this rebuttal only:

- Barker, P. A. et al. Seasonality in equatorial climate over the past 25 ky revealed by oxygen isotope records from Mount Kilimanjaro. *Geology* **39**, 1111-1114 (2011).
- Bergner, A.G.N., Trauth, M.H. & Bookhagen, B. Paleoprecipitation estimates for the Lake Naivasha basin (Kenya) during the last 175 k.y. using a lake-balance model. *Glob. Planet. Change* **36**, 117-136 (2003).
- Dang, X.Y., Xue, J.T., Yang, H. & Xie, S.C. Environmental impacts on the distribution of microbial tetraether lipids in Chinese lakes with contrasting pH: Implications for lacustrine paleoenvironmental reconstructions. *Science China Earth Sciences* **59**, 939-950.
- Dearing, J.A. Sedimentary indicators of lake-level changes in the humid temperate zone: A critical review. *J. Paleolimnol.* **18**, 1-24 (1999).
- Duesing, W. et al. Changes in the cyclicity and variability of the eastern African paleoclimate over the last 620 kyrs. *Quat. Sci. Rev.* **273**, 107219 (2021).
- Fiedler, S. et al. Simulated tropical precipitation assessed across three major phases of the coupled model intercomparison project (CMIP). *Mon. Weather Rev.* **148**, 3653-3680 (2020).
- Foerster, V. et al. Towards an understanding of climate proxy formation in the Chew Bahir basin, southern Ethiopian Rift. *Palaeogeogr. Palaeoclimatol. Palaeoecol.* **501**, 111-123 (2018).
- Foerster, V. et al. Pleistocene climate variability in eastern Africa influenced hominin evolution. *Nat. Geosci.* **15**, 805ff. (2022).
- Gritti, E.S. et al. Simulated effects of a seasonal precipitation change on the vegetation in tropical Africa. *Clim. Past* **6**, 169-178 (2010).
- Håkansson, L. & Jansson, M. Principles of lake sedimentology. Berlin: Springer (1982).
- Halamka, T.A. et al. Production of diverse brGDGTs by *Acidobacterium Solibacter* usitatus in response to temperature, pH, and O₂ provides a culturing perspective on brGDGT proxies and biosynthesis. *Geobiology* **21**, 102–118 (2022).
- Hilton, J.A. conceptual framework for predicting the occurrence of sediment focusing and sediment redistribution in small lakes. *Limnol. Oceanogr.* **30**, 1131-1143.
- Jackson, M. S. et al. High-latitude warming initiated the onset of the last deglaciation in the tropics. *Sci. Adv.* **5**, eaaw2610 (2019).
- Jungandreas L., Hohenegger C. & Claussen M. How does the explicit treatment of convection alter the precipitation-soil hydrology interaction in the Holocene African humid period? *EGUsphere preprint*, <https://doi.org/10.5194/egusphere-2022-890> (2022).
- Kaser, G., Hardy, D.R., Molg, T., Bradley, R.S. & Hyera, T.M. Modern glacier retreat on Kilimanjaro as evidence of climate change: Observations and facts. *Internat. J. Climatol.* **24**, 329–339 (2004).
- Larsen, C.P.S. & MacDonald, G.M. Lake morphometry, sediment mixing and the selection of sites for fine resolution palaeoecological studies. *Quat. Sci. Rev.* **12**, 781–792 (1993).
- Lewis Jr., W. M. A revised classification of lakes based on mixing. *Can. J. Fish. Aquat. Sci.* **40**, 1779-1787 (1983).
- Liu, J., Ning, L., Yan, M., Sun, W., Chen, K & Qin, Y. Paleomonsoon modelling within PMIP: recent progress and future directions. *PAGES Magazine* **29**, 96-97 (2021).
- Loutre, M.F. & Berger, A. Marine Isotope Stage 11 as an analogue for the present interglacial. *Glob. Planet. Change* **36**, 209-217 (2003).

- McKenzie, J.M., Mark, B.G., Thompson, L.G., Schotterer, U. & Lin, P.N. A hydrogeochemical survey of Kilimanjaro (Tanzania): Implications for water sources and ages: *Hydrogeol. J.* **18**, 985–995 (2010).
- Mölg, T. & Pickler, C. A Mid-troposphere perspective on the East African climate paradox. *Environ. Res. Lett.* **17**, 084041 (2022).
- Nash, D.J. et al. African hydroclimatic variability during the last 2000 years. *Quat. Sci. Rev.* **154**, 1-22 (2016).
- Nicholson, S. E. & Yin, X. Rainfall conditions in equatorial East Africa during the nineteenth century as inferred from the record of Lake Victoria. *Clim. Change* **48**, 387-398 (2001).
- Peterse, F. et al. Revised calibration of the MBT–CBT paleotemperature proxy based on branched tetraether membrane lipids in surface soils. *Geochim. Cosmochim. Acta* **96**, 215-229 (2012).
- Odongo, V.O. et al. Characterisation of hydroclimatological trends and variability in the Lake Naivasha basin, Kenya. *Hydrol. Process.* **29**, 3276-3293 (2015).
- Olaka, L. A., Odada, E. O., Trauth, M. H. & Olago, D. O. (2010). The sensitivity of East African rift lakes to climate fluctuations. *J. Paleolimnol.* **44**, 629-644 (2010).
- Philippon, N. et al. The light-deficient climates of western Central African evergreen forests. *Env. Res. Lett.* **14**, 034007 (2019).
- Roberts, H.M. et al. Using multiple chronometers to establish a long, directly-dated lacustrine record: Constraining >600,000 years of environmental change at Chew Bahir, Ethiopia. *Quat. Sci. Rev.* **266**, 107025 (2021).
- Rowan, D. J., Kalf, J. & Rasmussen, J. B. Estimating the mud deposition boundary depth in lakes from wave theory. *Can. J. Fish. Aquat. Sci.* **49**, 2490-2497 (1992).
- Ruddiman, W. F. The early anthropogenic hypothesis: Challenges and responses. *Rev. Geophys.* **45**, RG4001 (2007).
- Scholz, C.A. et al. 2007. East African megadroughts between 135 and 75 thousand years ago and bearing on early-modern human origins. *Proc. Natl. Acad. Sci. USA* **104**, 16416-16421.
- Shanahan, T.M. et al. Age models for long lacustrine sediment records using multiple dating approaches - An example from Lake Bosumtwi, Ghana. *Quat. Geochronol.* **15**, 47-60.
- Trauth, M.H. et al. Episodes of environmental stability and instability in Late Cenozoic lake records of eastern Africa. *J. Hum. Evol.* **87**, 21-31.
- Tzedakis, P.C. The MIS 11–MIS 1 analogy, southern European vegetation, atmospheric methane and the "early anthropogenic hypothesis". *Clim. Past* **6**, 131-144 (2010).
- Valero-Garcés, B.L. et al. A multi-dating approach to age-modelling long continental records: The 135 ka El Cañizar de Villarquemado sequence (NE Spain). *Quat. Geochronol.* **54**, 101006.
- Verschuren, D. Sedimentation controls on the preservation and time resolution of climate-proxy records from shallow fluctuating lakes. *Quat. Sci. Rev.* **18**, 821-837 (1999a).
- Verschuren, D. Influence of depth and mixing regime on sedimentation in a small, fluctuating tropical soda lake. *Limnol. Oceanogr.* **44**, 1103-1113 (1999b).
- Verschuren, D. Reconstructing fluctuations of a shallow East African lake during the past 1800 yrs from sediment stratigraphy in a submerged crater basin. *J. Paleolimnol.* **25**, 297-311 (2001).
- Verschuren, D., Edgington, D. N., Kling, H. J. & Johnson, T. C. Silica depletion in Lake Victoria: sedimentary signals at offshore stations. *J. Great Lakes Res.* **24**, 118-130 (1998).
- Yin, Q., & Berger, A. Interglacial analogues of the Holocene and its natural near future. *Quat. Sci. Rev.* **120**, 28-46 (2015).

Reviewer Reports on the First Revision:

Referees' comments:

Referee #1 (Remarks to the Author):

The details of past climate conditions recovered from Lake Chala on the side of Mt. Kilimanjaro are among the highest-fidelity records obtained from the continental tropics. Unquestionably this series of sediment cores from the confined watershed will be an essential reference site for future modeling and reconstruction studies.

The key results from the work highlight the relationship between temperature and hydroclimate, and they show that during the cooler episode of the last glacial these parameters have a positive relationship, which turns negative at the start of the Holocene, when CO₂ values increased significantly, and temperature control of evaporation began to dominate over temperature-enhanced precipitation.

The authors have proceeded to address many of the most pressing issues from the original submittal. It is heartening to now read the details of the YTT chronology, the more detailed explanation of the BIT index as hydroclimate proxy, and enhanced discussion of the mechanisms driving the changes in climate at the start of the Holocene.

The principal conclusions of this manuscript regarding the GI-IG transition are unaffected by the linear age model proposed for the interval 25-73 ka. Indeed I am not concerned that the general assertion of consistent sedimentation rates may hold. However this gap in absolute ages remains an assertion based upon a set of assumptions only. The imagery and seismic data are simply no substitute for additional radiocarbon ages in the interval of concern (or at least over the interval 25-40 ka). The issue is not for the conclusions of this paper, but rather the implications for what will surely be numerous publications that will follow, which will use this age model and material as the default for the foreseeable future.

Despite the very nicely laminated cores illustrated in the extended data, and the well-analyzed reflection seismic record that indeed nicely documents the turbidite events, it is highly unlikely that the sedimentation rates between 25-73 ka are strictly linear. This can be observed in the well-dated interval ~14-25 ka, where sedimentation rates vary by ~30% or more (Extended Data Figure 1).

Moreover although core images 6-10 shown for the depth range 25-55 m are all beautifully laminated, there are 3-5-fold changes in thickness of individual varves between them that imply non-trivial changes in sedimentation rates.

Why are additional absolute (radiocarbon) ages important in the context of this manuscript? It is because of the numerous very interesting events that clearly emerge in the older part of this record. For instance, the hydroclimate results show that the driest interval of the record, other than the LGM, is in their postulated Stage 4, perhaps coincident with H6. But is it really? Absolute ages over this interval will greatly enhance the interpretability of the older part of the record. The correlation shade panels (referencing the timing of marine isotope stages (MIS), Heinrich events (H1- 335 H6), the Last Glacial Maximum (LGM) and Younger Dryas (YD) stadial) imply far greater age model accuracy than is warranted here.

Referee #2 (Remarks to the Author):

I appreciate the authors' thorough response, and I find most of my concerns about the proxy interpretation are explained.

I'm still curious about why the ensemble temperature (Extended Data Figure 5b) started warming right after LGM while the Chala MST shows warming after H1. Maybe I wasn't bringing this up explicitly in my previous comments, but I feel this is interesting given the records only cover one glacial-IG cycle. The authors argue (R29a) the Chala MST is expected to show better similarity with Indian ocean SST and South Africa temperatures (Extended Data Figure 5e&f), and they both, in fact, show the warming trend after LGM, like the ensemble temperature (Extended Data Figure 5b).

The interpretation of BIT for Lake Chala relies on extensive field observations, which are extremely valuable for validating the proxies themselves. Based on the current Supplementary Information and Baxter et al. (2021) paper, I feel we are on the same page that the BIT is more directly pointing to lake dynamic, which hydroclimate variabilities could partly influence. I agree that the BIT in this case could be a semi-quantitative representation of hydroclimate, as the authors state in the Supplementary Information and the rebuttal. Meanwhile, I believe it's worth putting this caveat (only a semi-quantitative proxy, or equivalent description) in the main text. The interpretation of BIT can vary at different study sites, so people should be more cautious about this in my honest opinion.

Referee #3 (Remarks to the Author):

Baxter et al. have done a thorough discussion in response to all three reviewers, which I think is admirable. Many of my original comments have been well-reasoned and edited and/or clarified in the text. I remain still not thoroughly convinced that the correlation between temperature and effective moisture shifts so clearly, and thus, I think the main conclusion, that with continued CO₂, associated increased temps will drive the region to become drier, is so clearcut as presented. Please find more detail about this major comment, along with some minor and technical notes. Thank you to the authors for the immense amount of work that went into replying and reframing the text.

Major Comments:

1. The robustness of the moisture/temp correlation and the abrupt sign change

The authors rebutted my earlier comment on the usage of BIT-index values of 1 by removing the 3 datapoints that technically have a value of 1. However, in figures 2 and 3 it is abundantly clear that there are many samples that have a BIT-Index value of very, very close to 1. This means that when the peaks were integrated, if there was any little peak in the crenarchaeol GDGTs of the index ratio, the value would not technically be 1.

That being said, there are ways to get around this with non-parametric correlations. I highly recommend trying these techniques to deal with the non-normally distributed data that comes from a ratio (BIT Index). And I think that the correlation findings will hold, but the correct statistical

techniques, especially for this section that is so central to the main finding of the paper, need to be applied.

However, even if the correlation does hold for the Holocene ($n=37$), when the time series' of BIT-index and MST are compared, I simply don't see it. I think the termination of the high-BIT excursion coincides with the increase in MST, but because this signal is so strong and that excursion appears to be a transient event, this may be aliasing the significant correlation that is found. After 8 ka, they look largely unrelated, and the more recent time period is the precursor to the future moisture response to increasing temps and CO₂, which also don't appear to be correlated in the last 10 kyr (Fig 2e and 2f). The argument relies on temperature being very strongly linked with CO₂, but this isn't quite what we're seeing from the GDGT record.

This is all to say that I am not convinced from the arguments in the paper and the statistical handling of the datasets that the answer to the climate paradox has been found.

2. Orbital properties

Again, I encourage the authors to limit the mention of obliquity as a reason for the linking with the high northern latitudes. The record is less than twice the length of this periodicity, which is now mentioned in the text, but I still don't find it compelling within the bounds of the rules of spectral analysis. The periodicity that is found is 42.8, which is exactly double the precession-band that is noted, and with the missing precession low during MIS3, this could easily manifest into a double period. Further, an obliquity signal in the tropics does not necessarily mean there is an influence from the northern high latitudes (see O'Mara, 2022, doi:10.1038/s41467-022-31120-x). Also, the discussion of half-precession seems to be sort of a continuation of earlier papers from Chala, whereas it's not strong in this dataset (extended Fig 4a and b).

I don't see an explanation of what the 99% global and 99% local cut offs are for, besides in the Vaughan paper, so please include this in the figure caption. Please also explain why a 3rd order polynomial is used for detrending prior to spectral analysis.

Minor Comments:

- Change the word 'infer' in the title – the contrasting is not actually doing the inferring. Suggests? Demonstrates? Go back to imply? – this also applies to some other places the word infer is used, like in line 49.
- Line 72-73: I still take issue with the lack of records cited for records extending beyond the LGM. While I understand and appreciate the rebuttal to this previous comment in that you need a high-res record that matches the lithology, unlike some XRF records, the way it's framed now is excluding many records. There are many dDwax records of precipitation (and other proxies!) in this region that are continuous, well-resolved, and extend beyond the LGM. I understand what the authors are saying in the rebuttal, but the phrasing must be adjusted, then, to make it not such a broad umbrella. Perhaps it can be changed to Only a few continuous climate records from eastern Africa that are sufficiently resolved and dated to compare to....
- Lines 281-282: I understand that this has patterns of regional climate signal, but it is still based on local terrestrial sources. I think this needs to be toned down.
- Line 287: keep the discussion/cause – focused on CO₂, not temp, which is being used for the

correlation aspect. So instead of modestly cooler, can the authors say lower CO₂? I think there are a few times where T and CO₂, which are correlated on longer timescales (see my comment in the first section) are conflated and then the discussion is a bit lost.

- For the age model, I do appreciate the newly update language surrounding this, but I'm a bit confused about why there are no more 14C ages before ~22 ka.

Technical corrections:

- Generally, there are some very long sentences. I do not want to change the writing of the authors, as these stylistic points are their prerogative. But from a reader's standpoint, I found myself getting a bit lost in the independent phrases. For clarity, it may be best to make some of those long sentences into two.

- Just a suggestion, change whereas in line 35 to something stronger, like despite, and again in line 51 – and I don't think notwithstanding is really the correct word here

- line 108: remove comma after consider

- Line 128: remove 'was expected'

- Line 218-220: this is not an independent clause

- Line 269: use additionally OR also

Figures:

- I still find the map fairly difficult to look at – It looks like there are shadow effects under the arrows, and the scale bars don't seem to be centered, etc. It's busy but actually has a lot of space to work with. Maybe the smaller arrows could be removed?

- Similarly, there's a lot going on in Fig 2. I agree that these plots all need to be shown together, but the amount of background shading, colors, and dashed lines is a lot

B. Author response to referee comments

Referee #1

84. The details of past climate conditions recovered from Lake Chala on the side of Mt. Kilimanjaro are among the highest-fidelity records obtained from the continental tropics. Unquestionably this series of sediment cores from the confined watershed will be an essential reference site for future modeling and reconstruction studies. The key results from the work highlight the relationship between temperature and hydroclimate, and they show that during the cooler episode of the last glacial these parameters have a positive relationship, which turns negative at the start of the Holocene, when CO₂ values increased significantly, and temperature control of evaporation began to dominate over temperature-enhanced precipitation. The authors have proceeded to address many of the most pressing issues from the original submittal. It is heartening to now read the details of the YTT chronology, the more detailed explanation of the BIT index as hydroclimate proxy, and enhanced discussion of the mechanisms driving the changes in climate at the start of the Holocene.

R84. We very much appreciate the referee's reconfirmation of the global importance of the Lake Chala climate archive and the significance of the results presented in this manuscript, as well as their positive assessment of the earlier revisions made.

85. The principal conclusions of this manuscript regarding the GI-IG transition are unaffected by the linear age model proposed for the interval 25-73 ka.

R85. We are delighted that referee #1 concurs with our position on this issue.

86. Indeed I am not concerned that the general assertion of consistent sedimentation rates may hold. However this gap in absolute ages remains an assertion based upon a set of assumptions only.

R86. In our opinion it is exactly the atypical continuity, constancy and tranquillity ('high fidelity') of sediment accumulation in Lake Chala, demonstrated by our detailed analysis of the equally atypically high quality and resolution of the seismic data (11,53), which give sufficient credence to the validity of our assumption of long-term stability in sedimentation rate such that it constitutes a reliable lithostratigraphical constraint on the age-depth relationship in the 25-73.7 ka section. To drive this point home, an analogy can be made with the dating of the glacial section of the Greenland ice-core climate proxy record, which until layer counting in the NGRIP core eventually produced absolute ages, first to 42 ka (Andersson et al. 2006) and then to 60 ka (GICC05 age scale; Svensson et al. 2008), relied for more than a decade on a relatively simple snow compaction and ice flow model combined with a thermodynamically derived estimate of the difference in snow accumulation between the Holocene and the glacial period (SS09sea age scale; Johnsen et al. 2001). This extrapolated age model was generally deemed credible because of i) demonstrated continuity of the ice-core record (at least those drilled near the summit of the ice cap); ii) no long-term trend and only modest long-term variation in snow accumulation within the Holocene, suggesting that this might also be a valid assumption for the glacial period; iii) uniform visual appearance of glacial ice with depth (and long-term average concentrations of various chemical constituents); and iv) obvious lack of a mechanism that might create sustained changes in the rate of snow accumulation. In fact, still today the 60-123 ka section of the Greenland climate proxy record relies on the SS09sea extrapolated age model (E. W. Wolff,

University of Cambridge, *personal communication*), from since it was spliced into the GICC05 age scale of the 0-60 ka section (Wolff et al. 2010). Based on our seismic and lithological data (Extended Data Fig. 1) and arguments presented in the Supplementary Materials (lines S141-211), we believe that the portion of the Lake Chala record studied here deserves to be treated on a similar footing, certainly now that its base is anchored in the YTT cryptotephra (73.7 ± 0.4 ka). Which, as we have noted before, was found almost exactly at the depth where our seismic-based extrapolated age model predicted that it would occur (lines S168-171 and **R8a-b**). Interestingly, layer counting in the Greenland ice-core record to 60 ka (Svensson et al. 2008) necessitated correction of the extrapolated age model by ~ 700 years at that depth (Wolff et al. 2010). For comparison, our prediction of the depth at which the YTT would be found in the DeepCHALLA record was 68 cm off, which is equivalent to ~ 800 years.

87. The imagery and seismic data are simply no substitute for additional radiocarbon ages in the interval of concern (or at least over the interval 25-40 ka).

R87. We respectfully disagree, considering that the unavoidably large analytical, calibration and age-model related uncertainties on ^{14}C -based age determinations in this time range have bedevilled numerous attempts to unambiguously link signatures of millennial-scale climate and vegetation change in long European records to the now (i.e., since 2010; **R86**) well-constrained timing of millennial-scale events in the glacial section of the Greenland ice-core record. See, for example, Moreno et al. (2014) on the results of the community-wide INTIMATE project; these authors explicitly lament the poorly constrained (and to large extent ^{14}C -based) chronologies of pre-LGM records. In our opinion, ^{14}C -derived age markers on the >25 ka DeepCHALLA section would, by themselves, give only a false impression of chronological constraint. Moreover, given their large uncertainty envelopes they would carry little weight in the age model because the age-depth curve would be ‘pulled’ towards the strong attractor formed by the YTT age marker; see also **R91b**.

88. The issue is not for the conclusions of this paper, but rather the implications for what will surely be numerous publications that will follow, which will use this age model and material as the default for the foreseeable future.

R88. Referring to **R78** above, optimal age control on the proxy signatures of millennial-scale climate variability during the glacial period, such as the Heinrich events, is important for resolving questions about global climate dynamics during that cooler glacial period. We certainly do plan to address such paleoclimatology-themed questions in future publications, but these are i) not the focus of the present paper and ii) have lesser immediate implications for future climate change, as this involves warming beyond the already warm natural climate of the current interglacial period during which massive cryosphere-related phenomena such as the Heinrich events have not occurred.

89. Despite the very nicely laminated cores illustrated in the extended data, and the well-analyzed reflection seismic record that indeed nicely documents the turbidite events, it is highly unlikely that the sedimentation rates between 25-73 ka are strictly linear. This can be observed in the well-dated interval ~ 14 -25 ka, where sedimentation rates vary by $\sim 30\%$ or more (Extended Data Figure 1).

R89. As explained at length in the Supplementary Materials (lines S141-211) and in our previous rebuttal (**R8a**), sedimentation rates between 14.5 and 20.5 ka are on average 24% higher because pronounced lower lake level (reduced lake depth) increased sediment focusing towards the centre of the basin (53). In other words, during this particular period there is a mechanism to winnow settling sediment particles towards the depositional centre and thus accelerate the sedimentation rate there. We hope the referee can appreciate that in the confined basin (‘watershed’) of a steep-sided crater lake, such variation in sediment focusing is the only mechanism to produce sustained changes in sediment accumulation. One of us has shown this to be case in three crater lakes in Kenya with varying ratios of lake depth to diameter (Verschuren 1999a, 1999b, 2001; Fig. R6), and it will certainly be so in Lake Chala where near-vertical rocky crater slopes reach to ~ 55 m water depth (Fig. S1). Besides enhancing sediment focusing, the major lake-level drop that occurred 20.5-14.5 ka (we estimate it to have dropped to ~ 40 m below current lake level; 53) also allowed wind-driven or convective turbulence

to occasionally reach the profundal lake floor, enough to partly mix the mm-scale seasonal laminae and change the facies to cm-scale banding. If in the period 73.7-25 ka there was ever an episode of low lake level causing accelerated sedimentation lasting even a few millennia, it would similarly have been reflected in a facies change from mm-scale to cm-scale lamination, but we do not see this (cf. R8a).

Fig. R6. Relationship between lake depth, water-column stability, lithological facies changes and the rate of sediment accumulation in Lake Sonachi, a hydrologically closed and fairly shallow saline crater lake in Kenya. **a**, Temporal variation in lake depth over the past ~200 years (lake-level record based on instrumental gauge measurements transferred to the core sequence using high-resolution ²¹⁰Pb-dating), lithology and sedimentation rate, illustrating the enhanced sediment focusing, and switch to non-laminated sediment facies, during low-stands; from Verschuren (1999a). **b**, Illustration of the pre-dominant influence of hydroclimate variation (here multi-annual trends in rainfall) on lake depth, water-column stability, and the alternation of finely laminated (varve-like) and non-laminated sediment sections; from Verschuren (1999b). The 19th-century low-stands here highlighted in pink are coeval with the two pronounced 19th-century BIT-index minima in the decade-scale hydroclimate reconstruction from Lake Chala 300 km to the south (61; Fig. S6) because they represent the same severe drought episodes known to have affected much of equatorial eastern Africa (137).

90. Moreover although core images 6-10 shown for the depth range 25-55 m are all beautifully laminated, there are 3-5-fold changes in thickness of individual varves between them that imply non-trivial changes in sedimentation rates.

R90a. We are very well aware that the visual appearance of the mm-scale laminations seems to suggest substantial variation of varve thickness at relatively short (interannual to century) time scales. This is also what was found following the counting of two long (2609- and 2510-year) varved sections in the Challacea sequence (56). However, this varve-thickness variation does not imply sustained changes in long-term sedimentation rate that could substantially affect the age-depth curve, for two reasons.

R90b. First, the visually apparent light-dark couplets in this sequence should not be readily assumed to represent varves, the way they would in varved records from temperate-region lakes, or from tropical lakes experiencing one rain season annually (such as Yoa, Bosumtwi or Malawi: Francus et al. 2013, Shanahan et al. 2012, Johnson et al. 2001). Chala is an equatorial lake with too little temperature seasonality to drive the annual cycle of water-column mixing and stratification (94, 126) which controls the seasonal succession of organic and inorganic deposition that in turn creates varved sediments. Here we deal with two stratification periods (during rainy seasons) and two mixing periods (during dry seasons) per year, and how many nutrients are brought to the surface during each of those dry seasons determines the occurrence of diatom blooms which after settling to the bottom create the light

laminae. Many dark laminae are so thin (Fig. R7), and/or ‘contaminated’ with diatoms blooming during the short dry season that varve boundaries can only be determined correctly by microscopic analysis of epoxy-embedded thin sections (lines S55-60). In previous work on the Chala varves (56), supported by several years of monthly sediment-trap data (57), a visually apparent light-dark couplet was often found to represent two varve years. For this reason we conservatively identify the facies of mm-scale lamination as ‘varve-like’, pending opportunity to count them microscopically.

R90c. Second, because under lake high-stand conditions, such as occur today, the variation in varve thickness is ultimately determined by the duration and depth of dry-season mixing (which controls in-lake nutrient cycling and thus the magnitude of diatom blooms). The only way to generate a sustained change in nutrient budget for the phytoplankton would be a major lake-level drop allowing oxygen to reach the lake floor and promote organic decomposition sufficiently to strongly enhance diatom (and other phytoplankton) productivity. But again, if this were to have happened, a facies change would have occurred from mm-scale to cm-scale banding (**R89**).

R90d. Perhaps most pertinent to this discussion, even in a well-constrained ^{14}C -based chronology such as the 25-0 ka section of the Chala record (to our knowledge still the best-dated tropical lake record of that length worldwide), variation in sedimentation rate is mostly determined by the stiffness parameter imposed on the age model. We (52) enjoyed the luxury of being able to adjust that stiffness parameter to the observed variation in varve thickness within the last 3000 years (56; Fig. R7); most often this is done more arbitrarily.

Fig. R7. *Left:* Idealized structure of a Chala varve composed of a light-coloured layer predominantly consisting of diatom frustules and a darker-coloured layer containing calcite crystals and amorphous organic matter mixed with fine-grained clastic mineral particles. *Right:* Varve-thickness variation in the last ~3050 years (age based on ^{210}Pb - and ^{14}C -dating; 52), within which 2609 varves could be delineated with certainty. The light/dark layer distinction (on average 83% and 17% of total varve thickness) is based on how these layers appear in microscopic thin sections, not their visual appearance on the core surface.

91. *Why are additional absolute (radiocarbon) ages important in the context of this manuscript? It is because of the numerous very interesting events that clearly emerge in the older part of this record. For instance, the hydroclimate results show that the driest interval of the record, other than the LGM, is in their postulated Stage 4, perhaps coincident with H6. But is it really? Absolute ages over this interval will greatly enhance the interpretability of the older part of the record. The correlation shade panels (referencing the timing of marine isotope stages (MIS), Heinrich events (H1-H6), the Last Glacial Maximum (LGM) and Younger Dryas (YD) stadial) imply far greater age model accuracy than is warranted here.*

R91a. We must take exception to the referee’s statement with regard to the timing of the YD, HS1 and LGM (even H2), as these millennial-scale events do fall within the 25-ka ^{14}C -dated section. Admittedly, the referee may have been confused by the age markers plotted at the bottom of Fig. 2 as previously

submitted, which due to a production error reflected uncalibrated ^{14}C ages instead of calendar ages. We apologise for this oversight, which has been corrected in the present version of Fig. 2, in line with the data shown in Extended Data Fig. 1 and listed in Supplementary Table 2.

R91b. The Chala BIT-index signature of inferred drought dated to between 12.9 ka and 11.7 ka (and perfectly consistent between the Challacea (28) and DeepCHALLA reconstructions: Fig. S6) is by far the best-dated YD signature of any African lake record. It is mostly because we realise that ^{14}C age determinations older than 25 ka can by themselves never attain comparable precision on the timing of millennial-scale events within the glacial period that we hold off on presenting such data until they can be integrated with the results of other absolute dating methods (which are being developed but will take time to complete), ideally also of microscopic varve counting (which is very labour-intensive, considering the sheer number).

R91c. With respect for the optimism expressed by referee #1, the reality is that the large calendar-age uncertainty on ^{14}C calibrations for the >25-ka period, and unavoidable contamination of >25-ka sediments with small, but increasingly impactful amounts of young carbon during sample processing, aggravated in our system by unknown variability in the magnitude of the old-carbon offset of bulk organic carbon (the ‘reservoir’ age; 52), renders such ^{14}C dating highly uncertain. In paleorecords known (or suspected) to extend beyond 55 ka, the customary method to correct for young-carbon contamination is to estimate the (average) amount of young carbon present in all dated samples from the so-called ‘background’ ^{14}C age of >55-ka material that in principle should be devoid of ^{14}C . But due to among-sample variation in the amount of young-carbon contamination, the scatter in such ‘background’ ^{14}C ages is typically large, such that even a modest difference between the median and average age of a set of ‘background’ ^{14}C ages (i.e., deviation from a Gaussian distribution) can influence the correction applied to >25 ka ^{14}C dates sufficiently to shift their calibrated ages by several millennia. Realistic estimation of the magnitude of each of these multiple sources of uncertainty results in compound error envelopes on those ^{14}C -derived ages that are (much) wider than the duration of the millennial-scale events themselves, and hence the >25 ka ^{14}C dates would carry little weight in the composite age model, particularly in the vicinity of a strong ‘attractor’ (i.e., robust and precise time marker) such as the 73.7 ± 0.4 ka YTT. Of course, several other African climate reconstructions have made use of ^{14}C -dating in the 25-50 ka time frame, but none of these (except the Malawi record; 65) could firmly anchor the record to the well-constrained YTT age. Consequently, also any attempt to evaluate the synchronicity between millennial-scale proxy signatures among long African lake records and millennial-scale high-latitude climate variability such as the Heinrich events would run into the same problem as experienced by the INTIMATE programme. In fact, INTIMATE eventually booked greatest success with U/Th-dated speleothem records (e.g., Adolphi et al. 2018), and for synchronising lake and peat records placed greatest faith in shared tephrochronological links with externally dated reference tephras (e.g., Bronk-Ramsey et al. 2014). To conclude, we think that ^{14}C -derived age determinations beyond 25 ka would only give a false impression of absolute age constraint and hence give way to inappropriate suggestions of long-distance climatic teleconnections. Until the Chala age model in the 25-73.7 ka section can be supported by absolute age markers with appreciably narrower uncertainty envelopes than those associated with ^{14}C dates in that time range, to us the more conservative and justifiable approach is to rely on the strong stratigraphic constraint provided by the seismic and lithological data.

Referee #2

92. *I appreciate the authors’ thorough response, and I find most of my concerns about the proxy interpretation are explained.*

R92. We are heartened by this nice complement, and by reading that the clarifications we provided in our first response have to a large part achieved their objective.

93. *I’m still curious about why the ensemble temperature (Extended Data Figure 5b) started warming right after LGM while the Chala MST shows warming after H1. Maybe I wasn’t bringing this up*

explicitly in my previous comments, but I feel this is interesting given the records only cover one glacial-IG cycle. The authors argue (R29a) the Chala MST is expected to show better similarity with Indian ocean SST and South Africa temperatures (Extended Data Figure 5e&f), and they both, in fact, show the warming trend after LGM, like the ensemble temperature (Extended Data Figure 5b).

R93. Because the ensemble temperature reconstruction uses the published age models of each record at face value (i.e., not taking into account differences between the records in chronological control on their pre-Holocene sections), we don't think it is appropriate to derive from this ensemble the most probable timing of post-glacial warming in eastern Africa. Taking our cue from an approach taken by Loomis et al. (24), we only use the ensemble to obtain average absolute values for ΔT_{LGM} (peak glacial cooling) and ΔT_{HCO} (maximum Holocene warming) relative to today with which to rescale our Chala MST record. To determine the level of synchrony or asynchrony between all these records requires an analysis which takes the dating uncertainty on the start of the warming, or the period of most rapid warming, into account, such as was done by Tierney & deMenocal (81) in a different context, and for which formal procedures exist (e.g., De Cort et al. 2021). However, such effort is outside the scope of this study. As to our suggestion of similarity between the timing of post-glacial warming registered at Lake Chala and at sites 10 and 11, which similarly reflect conditions in the (south-)western Indian Ocean, we can only observe that the timing of most rapid postglacial warming registered at the latter two sites occurs at respectively ~16-15 ka and ~14-13 ka, post-dating the LGM by at least 3000 years and thus presumably recording a signature of HS1-period regional cooling. In fact, this is also the case at two sites included in the ensemble temperature reconstruction: Rutundu (site 3) and Victoria (site 5; Extended Data Fig. 5). At Mahoma (site 4), postglacial warming is inferred to have started ~20 ka (69), i.e., already before the end of the LGM as generally defined.

94. The interpretation of BIT for Lake Chala relies on extensive field observations, which are extremely valuable for validating the proxies themselves. Based on the current Supplementary Information and Baxter et al. (2021) paper, I feel we are on the same page that the BIT is more directly pointing to lake dynamic, which hydroclimate variabilities could partly influence. I agree that the BIT in this case could be a semi-quantitative representation of hydroclimate, as the authors state in the Supplementary Information and the rebuttal. Meanwhile, I believe it's worth putting this caveat (only a semi-quantitative proxy, or equivalent description) in the main text. The interpretation of BIT can vary at different study sites, so people should be more cautious about this in my honest opinion.

R94a. Referee #2 may have noticed that in our Lake Chala publications dealing with calibration and validation of the BIT index as hydroclimate proxy in that system, and as well in Supplementary Discussion of the present manuscript (lines S544-550), we have always stressed that our interpretation of the BIT index and the mechanisms underlying its variation is specifically valid for Lake Chala, emphasizing that environmental control on the sedimentary BIT index may be different at other lake sites. As well have we advised against its uncritical application as hydroclimate proxy at other sites. To avoid any confusion in this matter, we therefore consistently refer to the 'Chala BIT index' in the current work. Similarly, we do nowhere interpret the BIT index as a quantitative proxy for effective moisture, but only discuss BIT index results in terms of more/less moisture availability. Referring to **R54** (and see also **R98b**), we think that it would be fair to consider the Chala BIT Index an at least semi-quantitative hydroclimate proxy, but there is not enough solid data available to calibrate the index over its entire range of variation expressed in our 75-ka record, particularly as it does not reflect precipitation as such but effective moisture.

R94b. At the same time, we hope that our modern system studies of this deep crater lake (18, 60, 85, 123) and reconstructions in the recent historical past (61, 18) have by now demonstrated that regional hydroclimate variability has a strong, not to say dominant, influence on the BIT index of Lake Chala sediments at the time scales relevant to paleoclimate reconstruction. And of course, the robustness of such a reconstruction is also contingent upon long-term stability of the local depositional environment, as demonstrated by the seismic and lithological evidence (**R89**). These (undoubtedly fairly unique) boundary conditions support our confidence that we can reliably interpret BIT-index variations in the

Lake Chala sediment record as past changes in the region's hydroclimate, at least in the 75-ka section studied here where we think that our modern-system understanding is applicable (lines S547-550). Whether the BIT index can also be applied as such to the sediment records of more dynamic African (and other tropical) lake systems, such as lakes with large catchments, or with a distinct chemical gradient in their water column, should, with all respect, not be our concern at this point; cf. **R81**.

Referee #3

95. *Baxter et al. have done a thorough discussion in response to all three reviewers, which I think is admirable. Many of my original comments have been well-reasoned and edited and/or clarified in the text. I remain still not thoroughly convinced that the correlation between temperature and effective moisture shifts so clearly, and thus, I think the main conclusion, that with continued CO₂, associated increased temps will drive the region to become drier, is so clearcut as presented. Please find more detail about this major comment, along with some minor and technical notes. Thank you to the authors for the immense amount of work that went into replying and reframing the text.*

R95. We are thankful for the *unisono* positive remarks made by all three reviewers on our efforts to accommodate their comments either in the revised manuscript or in our previous response.

Major Comments:

96. *The robustness of the moisture/temp correlation and the abrupt sign change. The authors rebutted my earlier comment on the usage of BIT-index values of 1 by removing the 3 datapoints that technically have a value of 1. However, in figures 2 and 3 it is abundantly clear that there are many samples that have a BIT-Index value of very, very close to 1. This means that when the peaks were integrated, if there was any little peak in the crenarchaeol GDGTs of the index ratio, the value would not technically be 1. That being said, there are ways to get around this with non-parametric correlations. I highly recommend trying these techniques to deal with the non-normally distributed data that comes from a ratio (BIT Index). And I think that the correlation findings will hold, but the correct statistical techniques, especially for this section that is so central to the main finding of the paper, need to be applied.*

R96. Thank you for this valuable specific recommendation. We have now recalculated the regressions shown in Fig. 3 using non-parametric correlation, and indeed the results still hold with high statistical significance (lines 173 and 186).

97. *However, even if the correlation does hold for the Holocene (n =37), when the time series' of BIT-index and MST are compared, I simply don't see it. I think the termination of the high-BIT excursion coincides with the increase in MST, but because this signal is so strong and that excursion appears to be a transient event, this may be aliasing the significant correlation that is found. After 8 ka, they look largely unrelated, and the more recent time period is the precursor to the future moisture response to increasing temps and CO₂, which also don't appear to be correlated in the last 10 kyr (Fig 2e and 2f). The argument relies on temperature being very strongly linked with CO₂, but this isn't quite what we're seeing from the GDGT record.*

R97. We would like to point out that we nowhere claim proportionality of either temperature (MST) or effective moisture (BIT index) in the Horn of Africa with the absolute concentration of atmospheric CO₂, as the referee appears to suggest. We are only stating that above a certain region-specific threshold temperature (here ~24 °C at sea level) associated with the crossing of a certain CO₂ concentration (here ~250 ppm during the postglacial rise in atmospheric CO₂; Fig. 2f), the positive (but highly non-linear) effect of temperature on land evaporation starts dominating over the positive (and equally non-linear) effect of temperature on Horn of Africa monsoon dynamics. This is no different than the non-linear land and vegetation feedbacks which are being invoked to explain abrupt mid-Holocene drying of the Sahara (Claussen et al. 1999), which appeared to be set in motion when local summer insolation dropped below a threshold value of 470 W/m² (deMenocal et al. 2000). Or it is no different than the disproportionate effect of atmospheric CO₂ variation on global temperature

between glacial and interglacial periods, because when temperature dropped persistently below 0 °C in high-latitude regions, growth of large continental ice sheets strongly enhanced the ice-albedo feedback of Greenland, Antarctica and Arctic sea ice.

98. This is all to say that I am not convinced from the arguments in the paper and the statistical handling of the datasets that the answer to the climate paradox has been found.

R98a. This comment presents a strong challenge to the conclusions reached in our paper, and thus warranted additional analyses of new and existing data to increase confidence that our results are both trustworthy and representative for the study region. The results of two of these analyses have been incorporated in the main text and supplementary materials, those of two others in this rebuttal.

R98b. Using our unmodified climate-proxy time series, the inverse correlation between BIT index and MST values over the last 9 ka (the period after retreat from very high BIT-index values) is less strong (Spearman $\rho = -0.29$, $n = 50$; Fig. R8) than for the Holocene as a whole ($\rho = -0.57$, $n = 62$; Fig. 3a), but still statistically significant ($p = 0.042$), as expected given clearly opposite trends from the mid-Holocene to the late Holocene (Fig. 2d-e). The lesser significance of the BIT-MST correlation within the last 9000 years can be attributed to the higher influence of background variability in the proxy values when the range of variation is smaller, besides secondary climate-dynamical and hydrological phenomena influencing mean annual temperature and/or effective moisture registered at Lake Chala. The correlation and its statistical significance can be improved substantially (to $\rho = -0.61$; $p < 0.001$) by smoothing our BIT-index and MST proxy time series with a 5-point rolling mean (Fig. R8).

Fig. R8. **a**, Chala BIT-index variation in the last 9 ka, with red open circles connected by a thin red line indicating the 50 individual data points and the thick black line representing a 5-point rolling mean of these values. **b**, Regression of BIT-index versus rescaled MST values using the 50 individual data points, with Spearman's rank coefficient (ρ) and associated p value reflecting statistical significance at the $p < 0.05$ level. **c**, Variation in rescaled MST in the last 9 ka, formatted as in panel (a). **d**, Regression of BIT-index versus rescaled MST values using the 5-point smoothed data series ($n = 48$), with ρ and associated p value reflecting statistical significance at the $p < 0.001$ level. In all panels, small grey x symbols indicate sampled intervals partly consisting of turbidite material and therefore omitted from time series and correlation analysis. Here, time-series smoothing mimics the smoothing influences on proxy time series from other sites due to low sedimentation rates or bioturbation, see **R98b**.

This is because each datapoint in our unsmoothed BIT-index and MST time series is based on sampling core increments of 2 cm (integrating ~25 years of sedimentation) at 16-cm (~200-year) intervals, and thus the time series retains a high degree of the scatter that would be expressed in a proxy record where each datapoint integrates over ~25 years of sedimentation and is sampled contiguously (i.e., with ~25-year temporal resolution). See for example the high degree of scatter in the decade-scale resolution BIT-index time series of Buckles et al. (61; Fig. S6), contiguously sampled with each 1-cm core increment integrating 8.4 years of sedimentation on average. When the portion of that time series covering the last 1000 years ($n = 120$) is integrated over contiguous 4-cm (~33-year) intervals, the range of BIT-index values is reduced by half (0.42-0.86 to 0.47-0.70; see ref. 61). However, the high scatter in this decadal-resolved record is not white noise, as indeed the three most extreme BIT anomalies recorded within the last 200 years coincide with historically documented episodes of severe drought (137; cf. Fig. R6). In climate-proxy records from non-laminated (bioturbated or physically mixed) sediments at other sites this short-term variability is erased (i.e., the proxy time series are already smoothed *in situ*), and when laminated sediments are sampled such that each data point integrates over, say, ~100 years of sedimentation (because of lower sedimentation rates, or greater analytical sample needs), smoothing is imposed at the sampling stage. We prefer to adhere to the unsmoothed proxy time series in the main text and figures of this manuscript, considering that nobody will dispute that the early Holocene was the wettest period of the entire Holocene throughout eastern Africa (Tierney et al. 2011) and, therefore, we see no reason why BIT-index values approaching 1 during the period 11.7-9 ka should be excluded from the correlation representing current interglacial conditions.

R98c. To anchor our reconstructions of temperature (T, represented by MST) and effective moisture (P-E, represented by the Chala BIT index) in the context of recent historical (i.e., anthropogenic) climate change, we used available observational datasets of mean annual P, E and T over the past 42 years (1980-2021) to map the spatial distribution of recent historical correlation between P-E and T across the African continent and adjacent southwestern Asia (Fig. 1a). This map constitutes immediate visual evidence that most areas within the Greater Horn of Africa (i.e., including large areas of the East African Plateau outside of the Horn of Africa as defined climatologically, on account of their receipt of moist westerly air flow during boreal Summer) have experienced a progressive drying trend, while in large swathes of northern and western Africa recent historical warming has been accompanied by greater effective moisture. Together with maps of projected changes in T, P and P-E by the end of the 21st century (Fig. 1b; CMIP6 ensemble model using the SSP5-8.5 emission scenario, ref. 48), we hope that our revised Fig. 1 map and text added on lines 53-58 now better drives home the essence of the Eastern African Climate Paradox, namely that the Horn of Africa is suffering progressive drying while climate models project that the region should be receiving greater rainfall and/or at enjoy stable effective moisture. Besides the map's effectiveness in visualising the present-day reality of the Greater Horn of Africa entering its sixth consecutive failed rainy season (WMO 2022, UNHCR 2023), it has not escaped our attention that many of the dry (sub-) tropical regions which in recent years have suffered the opposite climate hazard of unusually intense rainfall and flooding (northern Nigeria, Sudan, South Sudan, even Pakistan) are coloured blue on this map. More generally, we hope that the revised Fig. 1 helps emphasize both the significance and timeliness of our findings.

R98d. To assess whether i) the contrasting trend in P-E versus T between the Horn of Africa and western/central tropical Africa (i.e., areas east and west of the Congo Air Boundary) observed in the instrumental data is also expressed at long time scales, and whether ii) our paleoclimate reconstruction from Lake Chala can be considered representative for long-term climate history in the Horn of Africa, we determined how moisture has been related to temperature throughout the Holocene (the last 11.7 ka) and during the last glacial period (i.e., prior to 11.7 ka) at five sites in eastern Africa besides Chala from where paired temperature and hydroclimate reconstructions of sufficient temporal reach and resolution are available (Methods and Supplementary Table 4). This showed that during the glacial period continental hydroclimate variation was positively related to temperature at all six registering sites across eastern Africa, and that this relationship switched to negative during the Holocene only at the three Horn of Africa sites (Extended Data Fig. 5). Thus, notwithstanding caveats related to the fact

that the paleoclimate reconstructions from these six sites are based on diverse temperature and hydroclimate proxies, all with their own caveats (lines S552-598; **R102b**), the spatial pattern of Holocene moisture-temperature relationships among the available eastern African sites is fully consistent with the spatial pattern of recent historical trends in the observational record (lines 197-206), and as well our Lake Chala data can be considered representative for long-term climate history in the Horn of Africa.

99. (2). *Orbital properties*. Again, I encourage the authors to limit the mention of obliquity as a reason for the linking with the high northern latitudes. The record is less than twice the length of this periodicity, which is now mentioned in the text, but I still don't find it compelling within the bounds of the rules of spectral analysis. The periodicity that is found is 42.8, which is exactly double the precession-band that is noted, and with the missing precession low during MIS3, this could easily manifest into a double period. Further, an obliquity signal in the tropics does not necessarily mean there is an influence from the northern high latitudes (see O'Mara, 2022, doi:10.1038/s41467-022-31120-x). Also, the discussion of half-precession seems to be sort of a continuation of earlier papers from Chala, whereas it's not strong in this dataset (extended Fig 4a and b).

R99. We agree with referee #3 that our discussion of orbital insolation signatures in the presented climate proxy data was somewhat out on a limb. Recognising that this discussion is not essential to the main topic of this paper we have now omitted all reference to orbital insolation variations from the main text and supplementary materials (including the original Extended Data Fig. 4 and Fig. S5). We are now only noting congruence of the Chala BIT and MST records with the succession of marine isotope stages MIS4-MIS1 as reflected in the Global Marine oxygen-isotope Stack (lines 107-108 and Fig. 2a).

100. I don't see an explanation of what the 99% global and 99% local cut offs are for, besides in the Vaughan paper, so please include this in the figure caption. Please also explain why a 3rd order polynomial is used for detrending prior to spectral analysis.

R100. Extended Data Fig. 4 has been omitted from the revised manuscript, cf. above.

Minor Comments:

101. - Change the word 'infer' in the title – the contrasting is not actually doing the inferring. Suggests? Demonstrates? Go back to imply? – this also applies to some other places the word infer is used, like in line 49.

R101. We are aware that 'infer' is a somewhat problematic word choice. Feeling that 'suggests' is too weak and 'demonstrates' too strong, we indeed opted to bring back 'imply' but are perfectly willing to follow the editor's advice in this matter, as well as about the general phrasing of the title (cf. **R77**).

102. - Line 72-73: I still take issue with the lack of records cited for records extending beyond the LGM. While I understand and appreciate the rebuttal to this previous comment in that you need a high-res record that matches the lithology, unlike some XRF records, the way it's framed now is excluding many records. There are many *dDwax* records of precipitation (and other proxies!) in this region that are continuous, well-resolved, and extend beyond the LGM. I understand what the authors are saying in the rebuttal, but the phrasing must be adjusted, then, to make it not such a broad umbrella. Perhaps it can be changed to Only a few continuous climate records from eastern Africa that are sufficiently resolved and dated to compare to....

R102a. Conscious that any selective citation of individual records could be criticized, and finding it more crucial to cite the climate-dynamical and hydrological literature pertinent to developing our argument towards the main conclusion, we rephrased the starting sentence of the second paragraph (lines 68-71) so that it now specifically refers to the relatively low number of published records from eastern Africa with *paired* temperature and hydroclimate reconstructions extending far enough into the last glacial period to help resolve the research question addressed in this study.

R102b. We found only five published records meeting the above criteria besides Chala and used them to further strengthen the general conclusion of our work (see **R98d**). However, it should be realized that, in contrast to our Chala data, these records typically do not record temperature and hydroclimate reconstructions directly from the same systems (e.g., plant waxes are transported over long distance and may derive from geographically changing regions over time). Further, one of these (Victoria, site 5) does not reach the LGM (70), one (Gulf of Aden, site 8) is a marine record where terrestrial hydroclimate is compared with SST rather than air-temperature variation (80-81), and a third (Malawi, site 7) uses not δD_{wax} as moisture proxy but $\delta^{13}C_{\text{wax}}$ (79). The variety of proxies used (Supplementary Table 4) raises the issue of whether the climate records are directly intercomparable: each proxy may record different aspects of hydroclimate (in case of δD_{wax} and $\delta^{13}C_{\text{wax}}$) or temperature (air temperature or SST, and based on either branched GDGTs, isoprenoid GDGTs or alkenones that are produced by different organisms at different depths in the water column). Moreover, some studies consider δD_{wax} to be a precipitation proxy (e.g., 69-70, 76, 125-126), others see a significant imprint of relative humidity (127-128), moisture source (129-130) or effective moisture (131). Consequently, the results of our regional analysis, while visually persuasive (Extended Data Fig. 5), do come with a number of caveats, which we feel compelled to mention in the Supplementary Materials (lines S552-598). In any event, it should be clear that the 75-ka paired moisture-temperature reconstruction from Lake Chala presented in our manuscript is the longest, and (by far) highest-resolution record from eastern Africa currently available, and moreover it uses a locally validated interpretation of the hydroclimate proxy.

103. - Lines 281-282: *I understand that this has patterns of regional climate signal, but it is still based on local terrestrial sources. I think this needs to be toned down.*

R103. Considering the results of our additional analyses (**R98**), we feel that we can consider our Lake Chala data as being broadly representative for the Horn of Africa region, but we rephrased the pertaining sentence to avoid the impression of extrapolating our results too widely (lines 232-235).

104. - Line 287: *keep the discussion/cause focused on CO₂, not temp, which is being used for the correlation aspect. So instead of modestly cooler, can the authors say lower CO₂? I think there are a few times where T and CO₂, which are correlated on longer timescales (see my comment in the first section) are conflated and then the discussion is a bit lost.*

R104. Whether P-E is positively or negatively related to temperature clearly depends on a certain absolute threshold temperature (shown by our data) which likely varies with the ratio between P and E and is therefore region-specific (shown by the observational record, Fig.1a). It just happens that in the Horn of Africa this threshold appears to be crossed at a CO₂ concentration of ~250 ppm. Considering the non-linear thermodynamic processes involved (**R97**), we prefer not to overemphasize the direct CO₂ forcing of the drying, except to make the logical deduction that further warming associated with elevated future CO₂ levels is unlikely to flip the Horn of Africa back into a wetter 'glacial' mode.

105. - *For the age model, I do appreciate the newly update language surrounding this, but I'm a bit confused about why there are no more 14C ages before ~22 ka.*

R105. It appears that also referee #3 has been confused by the erroneous position of the ¹⁴C age markers plotted at the bottom of Fig. 2, which reflected uncalibrated ¹⁴C ages instead of the calendar ages they ought to reflect. Sorry for this, cf. **R91a**.

Technical corrections:

106. - *Generally, there are some very long sentences. I do not want to change the writing of the authors, as these stylistic points are their prerogative. But from a reader's standpoint, I found myself getting a bit lost in the independent phrases. For clarity, it may be best to make some of those long sentences into two.*

R106. We have tried to be conscious of this comment during the rewriting, and are open to further specific recommendations.

107. - Just a suggestion, change whereas in line 35 to something stronger, like despite, and again in line 51 – and I don't think notwithstanding is really the correct word here

R107. We feel that using 'despite' already in the starting context sentence (lines 33-34) would point too strongly towards a certain outcome, so we decided to keep 'whereas'. The concluding sentence (lines 44-47) has been rephrased so that it also highlights our secondary conclusion about the need for more attention to thermodynamic influences on effective moisture, and this avoids use of 'notwithstanding'.

108. - line 108: remove comma after consider

R108. Adopted.

109. - Line 128: remove 'was expected'

R109. In the present revision, the entire section on orbital insolation signatures in our climate proxy data has been deleted (**R99**).

110. - Line 218-220: this is not an independent clause

R110. Sorry, to us this sentence is constructed well, and referee #2 has previously commented on the importance of this statement.

111.- Line 269: use additionally OR also

R111. Adopted.

Figures:

112. - I still find the map fairly difficult to look at – It looks like there are shadow effects under the arrows, and the scale bars don't seem to be centred, etc. It's busy but actually has a lot of space to work with. Maybe the smaller arrows could be removed?

R112. Fig. 1 has been completely overhauled to accommodate the observational data, and in the process we followed up on these suggestions.

113. - Similarly, there's a lot going on in Fig 2. I agree that these plots all need to be shown together, but the amount of background shading, colors, and dashed lines is a lot

R113. Following up on the editor's advice prompted by comment 91 by referee #1, background shading for the MIS has been removed, and that for H2-H6 has been given a lighter grey hue, so we hope its general appearance now falls a bit lighter on the eye (**R80**).

Additional references, used in this rebuttal only:

- Adolphi, F. et al. Connecting the Greenland ice-core and U/Th timescales via cosmogenic radionuclides: testing the synchronicity of Dansgaard–Oeschger events. *Clim. Past* **14**, 1755-1781 (2018).
- Andersen, K. K. et al. The Greenland Ice Core Chronology 2005, 15-42 kyr. Part I: Constructing the time scale. *Quat. Sci. Rev.* **25**, 3246-3257 (2006).
- Bronk Ramsey, C. et al. Integrating timescales with time-transfer functions: a practical approach for an INTIMATE database. *Quat. Sci. Rev.* **106**, 67-80 (2014).
- Claussen, M. et al. Simulation of an abrupt change in Saharan vegetation in the mid-Holocene. *Geophys. Res. Lett.* **26**, 2037–2040 (1999).
- De Cort, G., Chevalier, M., Burrough, S.L., Chen, C. Y., Harrison, S.P. (2021). An uncertainty-focused database approach to extract spatiotemporal trends from qualitative and discontinuous lake-status histories. *Quat. Sci. Rev.* **258**, 106870 (2021).

- deMenocal, P., Ortiz, J., Guilderson, T., Adkins, J., Sarnthein, M., Baker, L., & Yarusinsky, M. (2000). Abrupt onset and termination of the African Humid Period: Rapid climate responses to gradual insolation forcing. *Quat. Sci. Rev.* **19**, 347–361 (2000).
- Francus, F. et al. Varved sediments of Lake Yoa (Ounianga Kebir, Chad) reveal progressive drying of the Sahara during the last 6100 years. *Sedimentology* **60**, 911-934 (2013).
- Johnsen, S.J. et al. Oxygen-isotope and palaeotemperature records from six Greenland ice-core stations: camp Century, Dye-3, GRIP, GISP2, Renland and NorthGRIP. *J. Quat. Sci.* **16**, 299–307 (2001)
- Johnson, T.C., Barry, S. L., Chan, Y., Wilkinson, P. Decadal record of climate variability spanning the past 700 yr in the southern tropics of East Africa. *Geology* **29**, 83-86 (2001).
- Moreno, A. et al. A compilation of Western European terrestrial records 60-8 ka BP: towards an understanding of latitudinal climatic gradients. *Quat. Sci. Rev.* **106**, 167-185 (2014).
- Shanahan, T.M. Late Quaternary sedimentological and climate changes at Lake Bosumtwi Ghana: new constraints from laminae analysis and radiocarbon age modeling. *Palaeogeogr. Palaeoclimatol. Palaeoecol.* **361-362**, 49-60 (2012).
- Svensson, A. et al. A 60,000 year Greenland stratigraphic ice core chronology, *Clim. Past* **4**, 47-57 (2008).
- Tierney, J.E., Lewis, S.E., Cook, B.I., LeGrande, A.N., Schmidt, G.A. Model, proxy and isotopic perspectives on the East African Humid Period. *Earth Planet. Sci. Lett.* **307**, 103-112 (2011).
- UNHCR. <https://www.unhcr.org/news/briefing/2023/2/63fdbcee4/horn-africa-drought-enters-sixth-failed-rainy-season-unhcr-calls-urgent.html> (2023).
- Verschuren, D. Sedimentation controls on the preservation and time resolution of climate-proxy records from shallow fluctuating lakes. *Quat. Sci. Rev.* **18**, 821-837 (1999a).
- Verschuren, D. Influence of depth and mixing regime on sedimentation in a small, fluctuating tropical soda lake. *Limnol. Oceanogr.* **44**, 1103-1113 (1999b).
- Verschuren, D. Reconstructing fluctuations of a shallow East African lake during the past 1800 yrs from sediment stratigraphy in a submerged crater basin. *J. Paleolimnol.* **25**, 297-311 (2001).
- WMO. <https://public.wmo.int/en/media/news/greater-horn-of-africa-faces-5th-failed-rainy-season> (2022).
- Wolff, E. W., Chapellaz, J., Blunier, T., Rasmussen, S. O., Svensson, A. Millennial-scale variability during the last glacial: the ice core record. *Quat. Sci. Rev.* **29**, 2828-2838 (2010).

Reviewer Reports on the Second Revision:

Referees' comments:

Referee #1 (Remarks to the Author):

I appreciate the extensive remarks by the authors in response to my comments and to those of the other reviewers. These will lead to a much better outcome.

Minor comments:

Line 66. Something was dropped here: "This allowsto determine whether the region's semi-arid tropical climate regime..."

Line 70-71. "... only a handful comprise well-resolved reconstructions of both temperature and hydroclimate (Methods, Extended Data Fig. 5)." This requires citations.

Lines 258-259. Provide citation for illustration of the ITCZ and CAB boundaries.....there are many interpretations in the literature.

Age model discussion: The Editor provided the option of 1) adding additional radiocarbon results or 2) adding extensive caveats to Figure 2 and relevant parts of the text. Authors chose option 2.

Lines 95-101. Explicit statements are required HERE regarding the greater age uncertainty of the 25-73 ka interval. These seem not to be included in this round of edits.

Lines 130-132 Add additional statement about relative age model uncertainty on the 64-57 ka drought. Also on lines 158-159.

The Editor-recommended statement "varve-like sediments with higher age with higher uncertainties" still needs to be incorporated into the actual figure (not just a caption modification). Lines 289-290. The 'caveat' statement in this caption is clearly inadequate: "Considering limited age control in the 25–73.7-ka section of our climate-proxy time series, the timing of Heinrich events H2–H6 is indicative only.". This should clearly state that the "...the timing of Heinrich events H2–H6 and MIS 2-3 and MIS 3-4 boundaries are only approximations." Moreover I strongly recommend changing the pale H2-H6 bar boundaries to make the edges diffuse, to graphically convey the higher age uncertainties of this part of the record.

R86 & R87. That the Greenland Ice Core record does not have a robust age model prior to 30 ka, and accordingly one is not needed from Lake Chala is a specious argument. Radiocarbon dating is not an option in ice cores, whereas it is most certainly a reasonable approach and highly achievable in tropical lake systems. There are many valid examples of robust radiocarbon dates from lake basins older than 25 ka, which are not fundamentally flawed nor colored by high uncertainties, as implied in this rebuttal. There is no "25ka cutoff" between robust (as shown by the exquisite younger Chala record) and inadequate radiocarbon dating. Rather this appears to be an arbitrary operational cutoff, relying only on the wonderful 2005 Challacea record and not on any new dates. Additional ¹⁴C constraints are easily achieved, to at least 40 ka. The assumption of constant sediment rates in the older interval continues to raise issues. For instance, the authors document a change in sedimentation rates during the interpreted interval of low water levels in the late glacial, but there is no corresponding change in lamination character/sedimentation rates during the reported interval 50-65 ka, when BIT values are similarly diminished.

R87-91. We appreciate the detailed responses and consideration of radiocarbon challenges due to

old carbon/reservoir impacts, young carbon contamination, older section calibration issues, as well as the impact of age modeling considerations (such as the stiffness parameter). Age modeling approaches continue to evolve however, and for the Chala record to withstand the test of time it will need additional absolute age constraints (sooner rather than later). Given that the Chala team continues to pursue the many pathways for further constraining the chronology, we urge the authors not to box themselves in by publishing a record that overstates the confidence of older record chronology.

Referee #3 (Remarks to the Author):

Baxter, Verschuren et al. have once again provided extremely thorough responses to each of my and other reviewers' comments. A huge amount of effort on their part has paid off, and I believe the study is very clearly presented and reasoned. I really like their new framing of the energy-limited versus moisture-limited regimes, as I think it simplifies things and allows the story to be more directly related back to the original motivation, as the structure and writing lays out nicely. I, and potentially other readers, am still not 100% convinced by the anticorrelation in the Holocene when I look at the time series, even the smoothed version, although the statistics don't lie. Thus, this finding, in the context of the ample support for the conclusions, demonstrates that this work is of high quality with larger implications for global climate and future climate change than the typical regional paleoclimate reconstruction. Nice work, team. I have some very minor comments, corrections, and notes on the figures below.

Minor Comments:

- I suggest acknowledging regional heterogeneity – even though this is a regional record, it is still including mostly arid, but also some of the Ethiopian Highlands – in abstract last sentence, can add 'on average'
- Lines 80-81: the ITCZ is different, although related, so either leave this in and say 'traditionally associated with the ITCZ', or just take out
- Line 205: what aspect of the climate regime?

Technical corrections:

- first paragraph, add citation for model projections that's in the caption already
- Line 66: 'allows to determine' sounds a little off
- Line 70: change 'both' to 'coupled'
- Line 97: change ka to kyr or change the last to since – also line 102, 120, 150, 157, remove ago line 193, 205 – this is a continuing (very small) problem throughout. Ka means 'thousand years ago' and ky means 'thousand years'.
- Remove comma at end of line 186
- Lines 210-216: this is all one sentence that should be broken up for clarity
- Line 238: use another word besides 'data' as it was just used

Figures:

- Fig. 1: color scaling should reflect the significance – 0.4 correlation is still pretty red

- Fig. 2 caption: 75 ka temp... sounds like it's at a specific time – see note above
- Fig. 3: maybe add these new terms moisture-limited and energy-limited to panel b; also, consider plotting BIT index on a nonlinear scale – log? – to reflect the non-parametric relationship

Author Rebuttals to Second Revision:

Manuscript 2022-08-12602C

'Contrasting glacial to interglacial temperature-moisture relationships imply further Horn of Africa drying' submitted by A. Baxter, D. Verschuren and 10 co-authors for publication in *Nature*.

Author response to new round of comments by referees #1 and #3

Referee #1 (Remarks to the Author):

I appreciate the extensive remarks by the authors in response to my comments and to those of the other reviewers. These will lead to a much better outcome.

R146. We are grateful for the opportunity to further discuss pertinent issues, and concur that our paper has much improved in the process.

Minor comments:

Line 66. Something was dropped here: "This allowsto determine whether the region's semi-arid tropical climate regime..."

R147. We now combined this and the preceding sentence for smoother reading (lines 62-66).

Line 70-71. "... only a handful comprise well-resolved reconstructions of both temperature and hydroclimate (Methods, Extended Data Fig. 5)." This requires citations.

R148. All published paired hydroclimate-temperature records from East Africa of sufficient temporal coverage are now incorporated in this work through our regional assessment (lines 218-230, Fig.6 & ED Table 1), but as in most cases the temperature and hydroclimate reconstructions were published separately, citing them here would compromise citation of other pertinent literature.

Lines 258-259. Provide citation for illustration of the ITCZ and CAB boundaries.....there are many interpretations in the literature.

R149. In ref. 17, expert Sharon Nicholson provides both a traditional schematic representation of ITCZ and CAB trajectories similar to ours, and extensive discussion about what these terms stand for in an African context, and which we have briefly incorporated in our text (lines 79-81).

Age model discussion: The Editor provided the option of 1) adding additional radiocarbon results or 2) adding extensive caveats to Figure 2 and relevant parts of the text. Authors chose option 2.

Lines 95-101. Explicit statements are required HERE regarding the greater age uncertainty of the 25-73 ka interval. These seem not to be included in this round of edits.

Lines 130-132 Add additional statement about relative age model uncertainty on the 64-57 ka drought. Also on lines 158-159.

The Editor-recommended statement "varve-like sediments with higher age with higher uncertainties" still needs to be incorporated into the actual figure (not just a caption modification).

Lines 289-290. The 'caveat' statement in this caption is clearly inadequate: "Considering limited age control in the 25–73.7-ka section of our climate-proxy time series, the timing of Heinrich events H2–H6 is indicative only.". This should clearly state that the "...the timing of Heinrich events H2–H6 and MIS 2-3 and MIS 3-4 boundaries are only approximations." Moreover I strongly recommend changing the pale H2-H6 bar boundaries to make the edges diffuse, to graphically convey the higher age uncertainties of this part of the record.

R150. Considering that the central topic of this paper is not about those glacial-era climate events, we are perfectly willing to make it clear to the reader that their timing as presented in Fig. 4 (as well ED Figs. 4 and 6; note the new numbering) is approximative. However, to avoid a multitude of caveat statements in text, figures and figure captions, we respectfully opted for the alternative solution to state upfront in the main text (lines 111-119) that the timing of these events as presented is based on

linear interpolation across the 73.7-25.2 kyr interval, and consequently that all cited age ranges in this interval should be viewed as approximate:

“The age model supporting our climate-proxy time series is anchored in 170 radiometric age markers (^{14}C and ^{210}Pb) covering the last 25.2 kyr (Supplementary Table 1), Younger Toba Tuff (YTT) glass shards from the 73.7-kyr Toba super-eruption in Indonesia²⁵ discovered near the base of the studied sequence (Fig. 3, Extended Data Fig. 3), and linear interpolation across the section of predominantly varve-like sediments constituting the 73.7-25.2 kyr interval (Fig. 2). Notwithstanding lithostratigraphic constraints on long-term variation in the rate of profundal sediment accumulation in this lake system (Methods), the age ranges cited below of millennial-scale climate events within this age interval should be viewed as approximative.”

We think the reader will appreciate that this statement applies to all instances in the text where age ranges are preceded by the tilde (~) sign to indicate the approximate nature of the reported values.

These text changes are further complemented by the following statement ending the Fig.4 caption (lines 477-479), and by the light grey vertical bars representing H2-H6 now having diffuse edges:

“Considering linear age interpolation in the 25.2–73.7-kyr section of our proxy time series (Methods), the timing of Heinrich events H2–H6 should be viewed as approximative (as indicated by graded light grey shading).”

Finally at the bottom of Fig. 4 we opted for the matter-of-fact specification “Varve-like sediments, interpolated age”.

We sincerely hope that in combination these changes alleviate the referee’s remaining concerns about our presentation of the age model.

R86 & R87. That the Greenland Ice Core record does not have a robust age model prior to 30 ka, and accordingly one is not needed from Lake Chala is a specious argument. Radiocarbon dating is not an option in ice cores, whereas it is most certainly a reasonable approach and highly achievable in tropical lake systems. There are many valid examples of robust radiocarbon dates from lake basins older than 25 ka, which are not fundamentally flawed nor colored by high uncertainties, as implied in this rebuttal. There is no “25ka cutoff” between robust (as shown by the exquisite younger Chala record) and inadequate radiocarbon dating. Rather this appears to be an arbitrary operational cutoff, relying only on the wonderful 2005 Challa record and not on any new dates. Additional ^{14}C constraints are easily achieved, to at least 40 ka. The assumption of constant sediment rates in the older interval continues to raise issues. For instance, the authors document a change in sedimentation rates during the interpreted interval of low water levels in the late glacial, but there is no corresponding change in lamination character/sedimentation rates during the reported interval 50-65 ka, when BIT values are similarly diminished.

R151. Appreciating the continued discussion of this matter, we can easily explain the lack of a change in sediment facies and sedimentation rate in the interval ~65-50 kyr by the observation that ~65-50 kyr ago still about 45-30 m less sediment had accumulated on the bottom of Lake Chala than by the time of the low-stand dated to 20.5-14.5 kyr ago, and consequently a much greater lake-level drop was required to create a comparable level of bottom turbulence and sediment focusing. In fact, the 0.85-m thick section of mixed mm/cm-scale lamination at 49.81-50.66 mfd (dated to ~58 kyr ago: Fig. 2b-c) is coeval both with the brief seismic low-stand at 50-51.5 mblf (Fig. 2a) and with lowest BIT values in that part of the sequence (Fig. 4d), meaning that at the height of the BIT-inferred drought, water-column mixing did occasionally reach the bottom. However, it is unlikely to have caused enhanced sediment focusing, given that comparable deep-mixing events in the Holocene (Fig. 2b) did not. In other words, further back in time the lithology of profundal Lake Chala sediments becomes gradually less sensitive to changes in lake surface level (and thus to changes in regional hydroclimate). The non-stationary relationship over long time scales between lithology and sedimentary GDGT distributions as hydroclimate proxies is a strong argument against combining them in a multivariate analysis to extract

the main trends of past hydroclimate variation, as we have argued before (**R14a**). Importantly, in the context of the age model presented here, what the deep-time lithology is losing in terms of climate-proxy sensitivity, it is winning in long-term stability of the rate of sediment accumulation.

R87-91. We appreciate the detailed responses and consideration of radiocarbon challenges due to old carbon/reservoir impacts, young carbon contamination, older section calibration issues, as well as the impact of age modeling considerations (such as the stiffness parameter). Age modeling approaches continue to evolve however, and for the Chala record to withstand the test of time it will need additional absolute age constraints (sooner rather than later). Given that the Chala team continues to pursue the many pathways for further constraining the chronology, we urge the authors not to box themselves in by publishing a record that overstates the confidence of older record chronology.

R152. We are aware of the high expectations raised by the exquisite ^{14}C -dating of the 25-kyr Challacea record (refs. 34,69). This is exactly why we opt to publish the results of absolute dating when multiple dating approaches will have converged into a robust final chronology. But again, this will be critical when discussing possible global teleconnections in millennial-scale climate events during the glacial period, a topic more suitable for a geoscience audience. We firmly believe that the conclusions and implications of the present study deserve a broader audience, and at the same time are, and will remain, robust even with an eventually adjusted age model.

Referee #3 (Remarks to the Author):

Baxter, Verschuren et al. have once again provided extremely thorough responses to each of my and other reviewers' comments. A huge amount of effort on their part has paid off, and I believe the study is very clearly presented and reasoned. I really like their new framing of the energy-limited versus moisture-limited regimes, as I think it simplifies things and allows the story to be more directly related back to the original motivation, as the structure and writing lays out nicely. I, and potentially other readers, am still not 100% convinced by the anticorrelation in the Holocene when I look at the time series, even the smoothed version, although the statistics don't lie. Thus, this finding, in the context of the ample support for the conclusions, demonstrates that this work is of high quality with larger implications for global climate and future climate change than the typical regional paleoclimate reconstruction. Nice work, team. I have some very minor comments, corrections, and notes on the figures below.

R153. We thank referee#3 (Rachel Lupien) for the complement and her renewed expression of appreciation of our work. As regards the anti-correlation in the Holocene: following up on our rebuttal (**R97b**) to the referee's earlier comment on this issue, the revised Fig. 5 now displays the non-parametric regressions based on slightly smoothed proxy time series (using a 5-point running mean) of both the glacial and Holocene sections of the 75-kyr record to accentuate low-frequency variability. The R and Rho values obtained when using the unsmoothed proxy time series are added to the figure caption (lines 487-490). Besides resulting in a visually more persuasive Fig. 5, our decision to calculate the Lake Chala regressions using smoothed proxy time series was mostly prompted by the realisation that it allows more correct comparison with regression results from the five other paired African proxy records available for our regional analysis (ED Table 1; Fig. 6). Indeed the smoothing applied to the Chala time series for this purpose mimics the greater time integration of individual data points in the other records, considering that these derive either from bioturbated or physically mixed sediments (Gulf of Aden, Rutundu, Victoria) in which short-term variability is erased *in situ*, or from sites with lower sedimentation rates (Tanganyika, Malawi) such that averaging of short-term variation is imposed at the sampling stage (Methods, lines 954-963).

Minor Comments:

- I suggest acknowledging regional heterogeneity – even though this is a regional record, it is still including mostly arid, but also some of the Ethiopian Highlands – in abstract last sentence, can add ‘on average’

R154. We agree that annual rainfall will remain higher at higher elevations within the Horn of Africa region than in the arid lowlands. However as indicated by the historical observational data (Fig. 1a) also those highlands situated in Ethiopia, at least the easternmost areas receiving moisture exclusively from the Indian Ocean, are likely to become drier. So we don’t feel the need to qualify our concluding statement beyond ‘likely’ (lines 44-46).

- Lines 80-81: the ITCZ is different, although related, so either leave this in and say ‘traditionally associated with the ITCZ’, or just take out

R155. We chose the first suggested option, and now write “Twice-annual passage of the tropical rain belt traditionally associated with the Intertropical Convergence Zone (ITCZ)”, referring to ref. 17 for details (lines 79-81).

- Line 205: what aspect of the climate regime?

R156. Now specified: “hydroclimate regime” (line 231).

Technical corrections:

- first paragraph, add citation for model projections that’s in the caption already

R157. Ref. 5 cited in the first paragraph is intended to cover a comprehensive set of different climate-model simulations, whereas ref. 51 in Fig. 1 is one representative example selected for visualisation.

- Line 66: ‘allows to determine’ sounds a little off

R158. We now combined this and the preceding sentence for smoother reading (lines 62-66).

- Line 70: change ‘both’ to ‘coupled’

R159. The temperature and hydroclimate reconstructions are not really ‘coupled’, as in fact it is desirable that the respective proxies are as independent as possible. We write ‘both’ (line 70) because for our analysis the temperature and hydroclimate reconstructions must derive from one and the same geological archive, so that its age model applies to both proxy time series. For optimal comparison the two proxy time series ideally involve the same set of samples, in which case ‘paired’ (line 70) seems to us the most suitable specifier.

- Line 97: change ka to kyr or change the last to since – also line 102, 120, 150, 157, remove ago line 193, 205 – this is a continuing (very small) problem throughout. Ka means ‘thousand years ago’ and ky means ‘thousand years’.

R160. We have followed the Nature format, which prescribes use of ‘kyr ago’.

- Remove comma at end of line 186

R161. OK, removed (line 207).

- Lines 210-216: this is all one sentence that should be broken up for clarity

R162. This chain of cause and effect proved difficult to split, but we rewrote it more succinctly (lines 236-241).

- Line 238: use another word besides ‘data’ as it was just used

R163. OK, replaced by ‘results’.

Figures:

- *Fig. 1: color scaling should reflect the significance – 0.4 correlation is still pretty red*

R164. OK, adjusted so only values above R values of ± 0.6 becomes bright red and blue.

- *Fig. 2 caption: 75 ka temp... sounds like it's at a specific time – see note above*

R165. This is now replaced by “Paired temperature and hydroclimate reconstruction for easternmost Africa spanning the last *ca* 75 kyr...”, although as the base of the record is close to the well-defined 73.7 \pm 0.4 kyr YTT age marker, we do not think the total time coverage of the proxy records is uncertain.

- *Fig. 3: maybe add these new terms moisture-limited and energy-limited to panel b; also, consider plotting BIT index on a nonlinear scale – log? – to reflect the non-parametric relationship*

R166. We added ‘energy-limited’ and ‘moisture-limited’ labels to the positive and negative (i.e., inverse) correlations between moisture and temperature in Fig. 5a. Referring to our rebuttals (**R96-R97**) to related previous comments by referee #3, we do not see the advantage of plotting BIT on a non-linear scale. However we now display both Pearson’s and Spearman’s rank correlation coefficients in Fig. 5a to indicate how non-parametric regression affects the result.